# Deficiency of gluconeogenic enzyme PCK1 promotes metabolic-associated fatty liver disease through PI3K/AKT/PDGF axis activation in male mice

Qian Ye [1,6], Yi Liu[1,6], Guiji Zhang[1,6], Haijun Deng [1,6], Xiaojun Wang[2,6], Lin Tuo[3,6], Chang Chen[4], Xuanming Pan[1], Kang Wu[1], Jiangao Fan[5], Qin Pan[5], Kai Wang [1 ✉], Ailong Huang [1 ✉] & Ni Tang [1 ✉]

Metabolic associated fatty liver disease (MAFLD) encompasses a broad spectrum of hepatic disorders, including steatosis, nonalcoholic steatohepatitis (NASH) and fibrosis. We demonstrated that phosphoenolpyruvate carboxykinase 1 (PCK1) plays a central role in MAFLD progression. Male mice with liver *Pck1* deficiency fed a normal diet displayed hepatic lipid disorder and liver injury, whereas fibrosis and inflammation were aggravated in mice fed a high-fat diet with drinking water containing fructose and glucose (HFCD-HF/G). Forced expression of hepatic PCK1 by adeno-associated virus ameliorated MAFLD in male mice. PCK1 deficiency stimulated lipogenic gene expression and lipid synthesis. Moreover, loss of hepatic PCK1 activated the RhoA/PI3K/AKT pathway by increasing intracellular GTP levels, increasing secretion of platelet-derived growth factor-AA (PDGF-AA), and promoting hepatic stellate cell activation. Treatment with RhoA and AKT inhibitors or gene silencing of RhoA or AKT1 alleviated MAFLD progression in vivo. Hepatic PCK1 deficiency may be important in hepatic steatosis and fibrosis development through paracrine secretion of PDGF-AA in male mice, highlighting a potential therapeutic strategy for MAFLD.

Metabolic dysfunction-associated fatty liver disease or 'MAFLD' (formerly known as 'NAFLD') is the most common chronic liver disease worldwide, affecting nearly 25% of US and European adults[1]. MAFLD is defined by the presence of steatosis in >5% of hepatocytes, regardless of alcohol consumption or other concomitant liver diseases, especially the presence of obesity and T2DM[2]. It entails a wide spectrum of hepatic clinical conditions, spanning from uncomplicated steatosis to nonalcoholic steatohepatitis (NASH), a more serious form of liver damage hallmarked by irreversible pathological changes such as inflammation, varying degrees of fibrosis, and hepatocellular damage, which is more likely to develop into cirrhosis and hepatocellular carcinoma[3]. Although multiple parallel insults, including oxidative damage, endoplasmic reticulum stress, and hepatic stellate cell (HSC) activation, have been

[1]Key Laboratory of Molecular Biology for Infectious Diseases (Ministry of Education), Institute for Viral Hepatitis, Department of Infectious Diseases, The Second Affiliated Hospital, Chongqing Medical University, Chongqing, China. [2]Institute of Hepatobiliary Surgery, Southwest Hospital, Third Military Medical University (Army Medical University), Chongqing, China. [3]Department of Infectious Disease, Hospital of the University of Electronic Science and Technology of China and Sichuan Provincial People's Hospital, Chengdu, China. [4]Institute of Life Sciences, Chongqing Medical University, Chongqing, China. [5]Department of Gastroenterology, Xin Hua Hospital, School of Medicine, Shanghai Jiao Tong University, Shanghai, China. [6]These authors contributed equally: Qian Ye, Yi Liu, Guiji Zhang, Haijun Deng, Xiaojun Wang, Lin Tuo. ✉e-mail: wangkai@cqmu.edu.cn; ahuang@cqmu.edu.cn; nitang@cqmu.edu.cn

proposed to explain the pathogenesis of MAFLD, the underlying mechanisms remain unclear[4].

In gluconeogenesis, glucose is generated from non-carbohydrate substrates, such as glycerol, lactate, pyruvate, and glucogenic amino acids, mainly in the liver, to maintain glucose levels and energy homeostasis. Phosphoenolpyruvate carboxykinase 1 (PCK1) is the first rate-limiting enzyme in gluconeogenesis and converts oxaloacetate to phosphoenolpyruvate in the cytoplasm[5]. Our previous studies showed that PCK1 deficiency promotes hepatocellular carcinoma progression by enhancing the hexosamine-biosynthesis pathway[6]. However, PCK1 regulates not only glucose homeostasis but also lipogenesis by activating sterol-regulatory element-binding proteins[7]. Patients lacking PCK1 function present diffuse hepatic macrosteatosis concomitant with hypoglycemia and hyperlactacidemia[8]. Similarly, mice with reduced *Pck1* expression develop insulin resistance, hypoglycemia, and hepatic steatosis, indicating the important role of PCK1 in regulating both glucose homeostasis and lipid metabolism[9,10]. However, the precise role of PCK1 in MAFLD progression is not well-understood.

The phosphoinositide 3-kinase/protein kinase B (PI3K/ATK) pathway plays a critical role in regulating cell growth and metabolism. This pathway is activated in response to insulin, growth factors, energy, and cytokines and, in turn, regulates key metabolic processes such as glucose and lipid metabolism and protein synthesis[11]. AKT promotes de novo lipogenesis primarily by activating sterol regulatory element-binding protein[12]. PI3K/AKT dysregulation leads to numerous pathological metabolic conditions, including obesity and type 2 diabetes[13]. MAFLD is characterized by dysregulated glucose and lipid metabolism in the liver. Although the PI3K/AKT pathway is a key regulator for sensing metabolic stress, its exact role in MAFLD progression is unclear[14,15].

In this study, we explored the role of *Pck1* in a mouse model. We determined the molecular mechanisms underlying disordered lipid metabolism, inflammation, and fibrosis induced by *Pck1* depletion. We also delineated the functional importance of the PI3K/AKT pathway and paracrine secretion of PDGF-AA as its effectors in steatohepatitis, providing a potential therapeutic strategy for treating MAFLD.

## Results
### PCK1 is downregulated in patients with NASH and mouse models of MAFLD
To determine whether PCK1 is involved in MAFLD, we first examined hepatic gene expression in a published transcriptome dataset [Gene Expression Omnibus (GEO): GSE126848] containing samples from 14 healthy participants, 12 patients with obesity, 15 patients with NAFLD, and 16 patients with NASH[16]. Bioinformatics analysis showed that 32 genes were markedly changed in obesity, NAFLD, and NASH; 12 genes were considerably downregulated and 20 genes were upregulated (Supplementary Fig. 1a–c). Notably, *PCK1* was gradually reduced in patients with obesity, NAFLD, and NASH (Fig. 1a, b). Downregulation of *PCK1* mRNA was also observed in a similar dataset (GSE89632) (Fig. 1b). Moreover, immunohistochemistry (IHC) assays showed that hepatic PCK1 protein levels were significantly lower in patients with NASH than in healthy participants (Fig. 1c). Similarly, *PCK1* mRNA and protein levels decreased in the liver of mice fed a high-fat diet with drinking water containing fructose and glucose (HFCD-HF/G) for 24 weeks (Fig. 1d, e).

Next, palmitic acid (PA) was used to mimic the liver steatosis of patients with MAFLD in vitro[17]. Cell growth was assessed in a CCK8 assay after treatment with different concentrations of PA (Supplementary Fig. 1d). Interestingly, the PCK1 mRNA and protein levels were downregulated in a dose-dependent manner during 24 h PA stimulation (Fig. 1f, g), suggesting that the transcription of PCK1 was inhibited in response to lipid overload. We screened several known regulators of *PCK1* (Supplementary Fig. 1e, f) and found that activating transcription factor 3 (*ATF3*), a transcriptional repressor of *PCK1*[18], was upregulated

upon PA stimulation (Fig. 1h). Similarly, ATF3 expression was remarkably upregulated in liver samples derived from patients with NASH and MAFLD model mice (Fig. 1i, j, Supplementary Fig. 1g, h). Chromatin immunoprecipitation assays revealed that the binding of ATF3 to the *PCK1* promoter increased following PA administration (Fig. 1k). ATF3 knockdown considerably enhanced *PCK1* promoter activity and restored PCK1 expression in human hepatocytes treated with PA (Fig. 1l), while overexpression of ATF3 played a opposite effects on *PCK1* promoter activity (Supplementary Fig. 1i, j). Furthermore, correlation analysis revealed that *ATF3* was negatively correlated with the *PCK1* mRNA level based on the GEO database (GEO: GSE135251) (Fig. 1m). These results indicate that increased lipid intake led to the upregulation of the repressor ATF3, impairing *PCK1* transcription in patients with NASH and mouse models.

### L-KO mice exhibit a distinct hepatic steatosis phenotype
To explore the role of *Pck1* in fatty liver disease, wild-type (WT) and liver-specific *Pck1*-knockout mice (L-KO) mice were fed a chow diet for 24 weeks (Fig. 2a). Hepatic-specific depletion of PCK1 was confirmed by performing immunoblotting (Fig. 2b). Starting at 16 weeks, L-KO mice showed an increased body weight compared with WT mice, however, the results of the glucose tolerance test (GTT) and insulin tolerance test (ITT) did not significantly differ (Fig. 2c). Moreover, increased liver weight was observed in L-KO mice (Fig. 2d). Alanine transaminase (ALT) and aspartate transaminase (AST) levels were higher in L-KO mice, indicating liver injury (Fig. 2e). In addition, total triglyceride (TG), total cholesterol (TC), and free fatty acids (FFAs) in the liver tissues and serum were elevated in L-KO mice compared to those in WT mice (Fig. 2f, g). Histochemistry and enzyme-linked immunosorbent assay (ELISA) showed that L-KO mice had prominent hepatic steatosis, increased inflammatory infiltration, and high levels of TNF-α, whereas hepatic fibrosis was not observed (Fig. 2h–j). Additionally, PCK1 deficiency significantly increased the mRNA levels of genes related to fatty acid transport and inflammation, whereas there were no significant changes in fibrosis-related genes in L-KO mice fed the chow diet (Fig. 2k). These data suggest that L-KO mice exhibited a distinct hepatic steatosis phenotype and liver injury even when fed normal chow.

### Hepatic loss of *Pck1* promotes inflammation and fibrogenesis in MAFLD mice
To explore whether an unhealthy diet could exacerbate pathologic changes in L-KO mice, WT and L-KO mice were fed HFCD-HF/G (Fig. 3a)[19,20]. Starting at 4 weeks, L-KO mice showed significant weight gain (Fig. 3b). The GTT and ITT showed that L-KO mice developed a more severe form of glucose intolerance and insulin resistance compared with those in WT mice (Supplementary Fig. 2a, b). L-KO mice had heavier livers compared with WT mice (Fig. 3c), although there was no significant difference in the liver weight ratio (Supplementary Fig. 2c). Insulin, AST, ALT, TC, TG, and FFAs increased in the serum and liver homogenates of L-KO mice, suggesting more serious liver injury and lipid metabolism disorder (Fig. 3d, e, Supplementary Fig. 2d, e). Analyses of L-KO liver sections revealed increased fat droplets, more severe fibrosis, and greater macrophage infiltration (Fig. 3f, Supplementary Fig. 2f). Furthermore, L-KO mice exhibited higher NAFLD activity scores (NAS score) and higher TNF-α and IL-6 levels (Fig. 3g, h). In addition, the expression of inflammatory factors, lipogenic enzymes, and fibrogenesis-associated genes was upregulated in L-KO mice (Supplementary Fig. 2g). In summary, mice lacking hepatic *Pck1* showed substantial liver inflammation and fibrosis after being fed the HFCD-HF/G.

Besides, we used a genetic model with hepatic deficiency in phosphatase and tensin homolog (PTEN) to induce MAFLD. We generated a mouse model with biallelic deletion of *Pck1* and *Pten* in the liver (cPten^(f/f)Pck1^(f/f)) (Supplementary Fig. 3a, b). At 6 months,

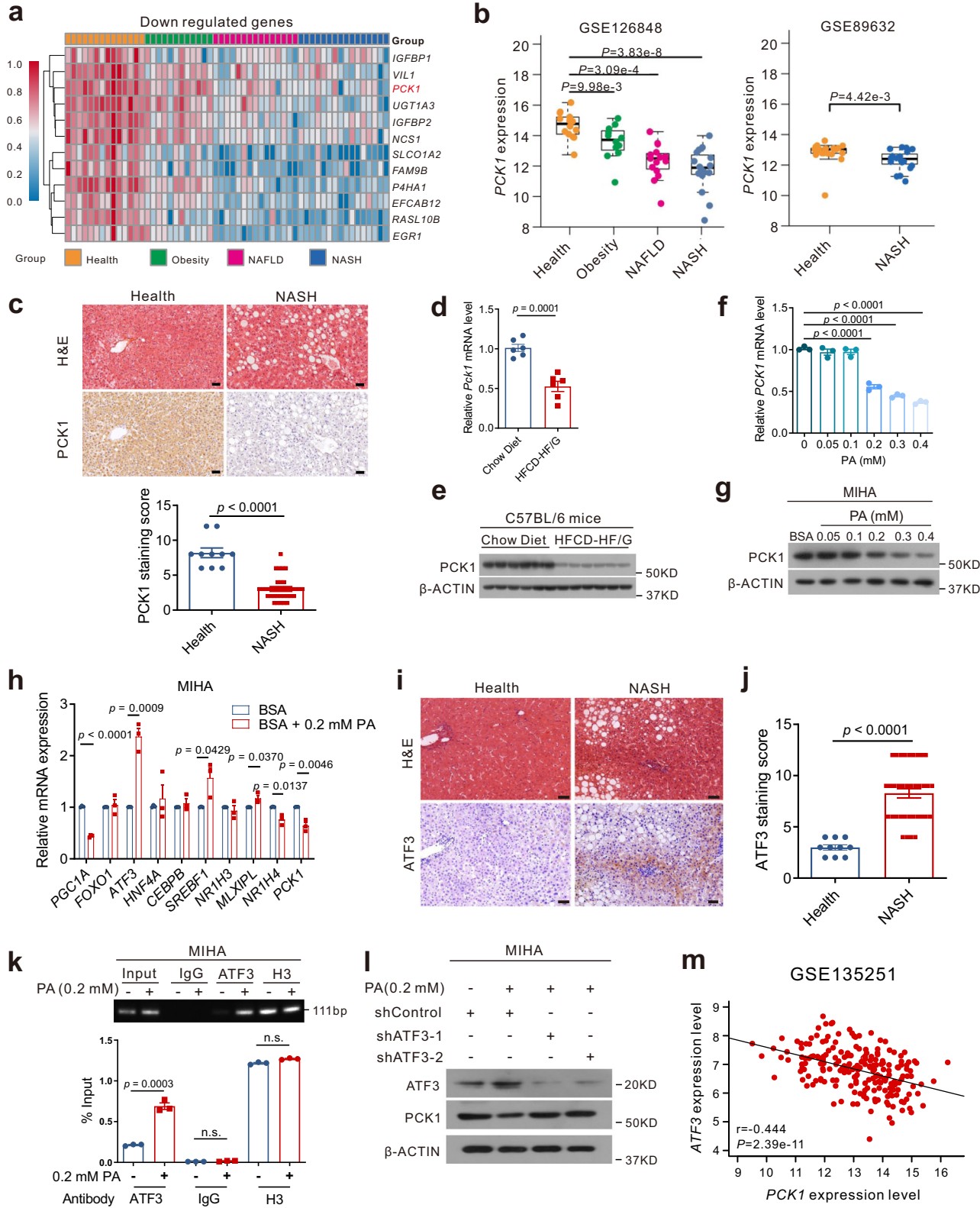

cPten[f/f]Pck1[f/f] mice had higher liver/body weight ratios compared with cPten[f/f] mice (Supplementary Fig. 3c). ALT, AST, IL-6, TNF-a, TC, TG, and FFAs levels significantly increased in the serum and liver tissues of cPten[f/f]Pck1[f/f] mice, suggesting more serious liver injury and lipid metabolism disorder in cPten[f/f]Pck1[f/f] mice (Supplementary Fig. 3d, e). Histological analysis of cPten[f/f]Pck1[f/f] liver sections revealed increased fat droplets, more severe fibrosis, and greater macrophage infiltration (Supplementary Fig. 3f, g). In agreement with increased inflammation and fibrosis, elevated expression levels of genes related to inflammation and fibrosis were observed in cPten[f/f]Pck1[f/f] mice (Supplementary Fig. 3h). In summary, hepatic Pck1 depletion showed substantial liver steatosis, inflammation and fibrosis in PTEN-null livers.

**Fig. 1 | PCK1 is downregulated in patients with NASH and mouse models of MAFLD. a** Genes downregulated in patients with health ($n = 14$), obesity ($n = 12$), NAFLD ($n = 15$), and NASH ($n = 16$) from GSE126848 dataset. **b** Relative *PCK1* mRNA levels of health ($n = 14$), obesity ($n = 12$), NAFLD ($n = 15$), and NASH ($n = 16$) in GSE126848 and of health ($n = 19$), and NASH ($n = 24$) in GSE89632 datasets. The box plots show the medians (middle line) and the first and third quartiles (boxes), whereas the whiskers show 1.5× the IQR above and below the box. Unpaired, two-sided Mann–Whitney U test *P* values are depicted in the plots, and the significant *P* value cutoff was set at 0.05. **c** PCK1 expression in normal individuals and patients with NASH and semi-quantitative analyses of immunohistochemistry (IHC) data (health, $n = 10$; NASH, n = 36). Scale bars: 50 μm. **d, e** mRNA (**d**) and protein (**e**) levels of PCK1 in the livers of WT mice fed with chow diet or HFCD-HF/G ($n = 6$). n was the number of biologically independent mice. The samples were derived from the same experiment and the blots were processed in parallel. **f, g** PCK1 mRNA (**f**) and protein

(**g**) levels in MIHA cells treated with BSA or PA-BSA. **h** Relative levels of indicated genes in MIHA cells treated with 0.2 mM PA. **i, j** Representative ATF3 expression in normal individuals and patients with NASH (**i**) and semiquantitative analyses of IHC data (**j**) (health, $n = 10$; NASH, $n = 36$). Scale bars: 50 μm. **k** Chromatin immuno-precipitation assays were performed in MIHA cells with or without PA treatment using an antibody against ATF3, IgG, or H3. **l** Protein levels of PCK1 in MIHA cells infected with either shControl or shATF3 treated with 0.2 mM PA. The samples were derived from the same experiment and the blots were processed in parallel. **m** Correlation analysis of *ATF3* mRNA level with *PCK1* in human NAFLD/NASH liver samples (GSE135251, $n = 206$). For **f** and **h**, $n = 3$. Data are expressed as the mean ± SEM; n.s., not significant. *p* values obtained via two-tailed unpaired Student's *t* tests, one-way analysis of variance with Tukey's post hoc test, or non-parametric Spearman's test. Source data are provided as a Source Data file.

## AAV-mediated restoration of hepatic PCK1 alleviates the MAFLD phenotype in *Pck1*-deficient mice

We then investigated whether adeno-associated virus (AAV)-based *Pck1* replacement therapy could reverse ongoing liver derangement, which is typically observed in patients with MAFLD. After 10 weeks of chow diet or HFCD-HF/G feeding, WT and L-KO mice were injected through the tail vein with AAV serotype 8 (AAV8) vector expressing *Pck1* under the control of a liver-specific promoter (thyroxine-binding globulin, TBG), AAV8-TBG-*Pck1* or AAV8-TBG-*control* (Fig. 3i). The expression of *Pck1* mRNA and PCK1 protein levels was identified (Supplementary Fig. 4a, b). Interestingly, mice with PCK1 re-expression showed a lower liver weight, body weight, serum liver enzymes, and serum lipid contents (Fig. 3j, Supplementary Fig. 4c, d). Moreover, lipid deposition, inflammation, and fibrosis significantly improved in L-KO mice injected with AAV8-TBG-*Pck1* (Fig. 3k-l). Hepatic gene expression analyses indicated that the expression of genes involved in inflammation and liver fibrosis was greatly attenuated by PCK1 restoration in L-KO mice (Supplementary Fig. 4e). Overall, these data support that forced PCK1 expression in the liver protects against MAFLD in mice.

## Transcriptomic and metabolomics analyses confirmed that the loss of *Pck1* promotes hepatic lipid accumulation

To comprehensively investigate the role of *Pck1* deficiency in MAFLD development, we performed RNA-seq analysis of liver samples from L-KO and WT mice fed the normal chow diet or HFCD-HF/G for 24 weeks. Gene Ontology analysis indicated that lipid metabolic processes were remarkably upregulated in L-KO mice fed the HFCD-HF/G (Fig. 4a). The volcano plot showed that genes involved in fatty acid uptake, such as solute carrier family 27 member 1 (*Slc27a1*) and fatty acid translocase (*Cd36*), and lipid droplet synthesis, such as cell death-inducing DFFA like effector C (*Cidec*) and cell death-inducing DFFA like effector A (*Cidea*), were upregulated in response to the HFCD-HF/G (Fig. 4b). Gene Set Enrichment Analysis (GSEA) revealed that the PPAR signaling pathway was prominently upregulated in L-KO mice fed either diet (Fig. 4c, Supplementary Fig. 5a, b). Several genes selected from the dataset were independently validated by quantitative polymerase chain reaction (qPCR) and immunoblotting and found to be significantly overexpressed in L-KO mice (Fig. 4d, e). Furthermore, genes involved in the glycerol 3-phosphate (G3P) pathway were upregulated in L-KO mice (Fig. 4f). Metabolomics analysis showed that compared with WT mice fed the HFCD-HF/G, L-KO mice had significantly higher G3P and PA levels (Fig. 4g, h). As G3P is a substrate for TG synthesis and PA is a key intermediate metabolite in de novo lipogenesis, *Pck1* ablation may promote substrate accumulation for lipid synthesis.

To further examine the function of PCK1 in steatosis in vitro, we overexpressed (*PCK1*-OE) using the AdEasy adenoviral vector system and knocked out PCK1 (*PCK1*-KO) using the CRISPR-Cas9 system in human hepatocytes (Supplementary Fig. 5c, d). We found that *PCK1*-OE attenuated the accumulation of lipid droplets, whereas *PCK1*-KO

facilitated lipid accumulation (Supplementary Fig. 5e, f). Collectively, these results suggest that hepatic *Pck1* deficiency leads to lipid accumulation by promoting the expression of lipogenic genes and accumulation of substrates related to lipid synthesis (Supplementary Fig. 5g).

## Hepatic *Pck1* deficiency leads to HSC activation via PI3K/AKT pathway

RNA-seq analysis indicated that the PI3K/AKT pathway was specifically activated in L-KO mice fed the HFCD-HF/G (Fig. 5a, b). Immunoblotting revealed p-AKT (S473) and p-AKT (T308), which are two activated forms of AKT, and downstream c-MYC were significantly upregulated in both the livers and primary hepatocytes of L-KO mice fed the HFCD-HF/G (Fig. 5c, d). qPCR confirmed that genes related to the PI3K/AKT pathway were highly expressed in L-KO mice (Supplementary Fig. 6a). Similarly, p-AKT (S473 and T308) significantly decreased in human *PCK1*-OE cells but increased in *PCK1*-KO cells after 0.2 mM PA treatment (Fig. 5e, f).

To clarify the role of PI3K/AKT pathway activation, transcriptome data were further analyzed. Interestingly, *Col1a1*, *Col3a1*, and *Lama2*, which are primary components of the extracellular matrix (ECM), were upregulated, as shown in the heat map of the PI3K/AKT pathway (Supplementary Fig. 6b). Moreover, GSEA analysis revealed that ECM-receptor interaction was upregulated in L-KO mice (Supplementary Fig. 6c). Because ECM deposition is typically considered as the key event underlying liver fibrosis, we predicted that the activation of the PI3K/AKT pathway promotes fibrosis in L-KO mice. HSCs are major ECM secretors; thus, we performed co-culture assays with human hepatocyte (MIHA) and human hepatic stellate cell lines (LX-2) (Fig. 5g). Interestingly, the mRNA levels of *ACTA2* (α-SMA, an HSC activation marker), *COL1A1*, and *COL3A1* increased in LX-2 cells co-cultured with *PCK1*-KO cells but decreased in LX-2 cells co-cultured with *PCK1*-OE cells (Fig. 5h, i). Similarly, COL1A1, COL3A1, and α-SMA expression increased in the liver tissues and primary HSCs of L-KO mice (Fig. 5j, k), which was confirmed in IHC analysis of COL3A1 (Supplementary Fig. 6d). However, these increases were partially reversed by MK2206, an AKT inhibitor (Fig. 5l, m). Collectively, these data suggest that the loss of PCK1 in hepatocytes induces HSC activation and ECM formation by activating the PI3K/AKT pathway.

## Paracrine PDGF-AA from hepatocytes promotes HSC activation

Hepatocytes elicit several fibrogenic actions in a paracrine manner to promote HSC activation[21]. Thus, PCK1-mediated hepatic fibrosis may be involved in paracrine disorders. To test this hypothesis, several pro-fibrotic factors were screened; *Pdgfa* was found to be significantly elevated in the liver tissues of L-KO mice (Fig. 6a). Bioinformatics analysis confirmed that *PDGFA* was significantly increased in patients with NAFLD and NASH (Fig. 6b). *Pdgfa* encodes a dimer disulfide-linked polypeptide (PDGF-AA), and the chronic elevation of PDGF-AA in the mouse liver induces fibrosis[22]. Immunoblotting and ELISA revealed

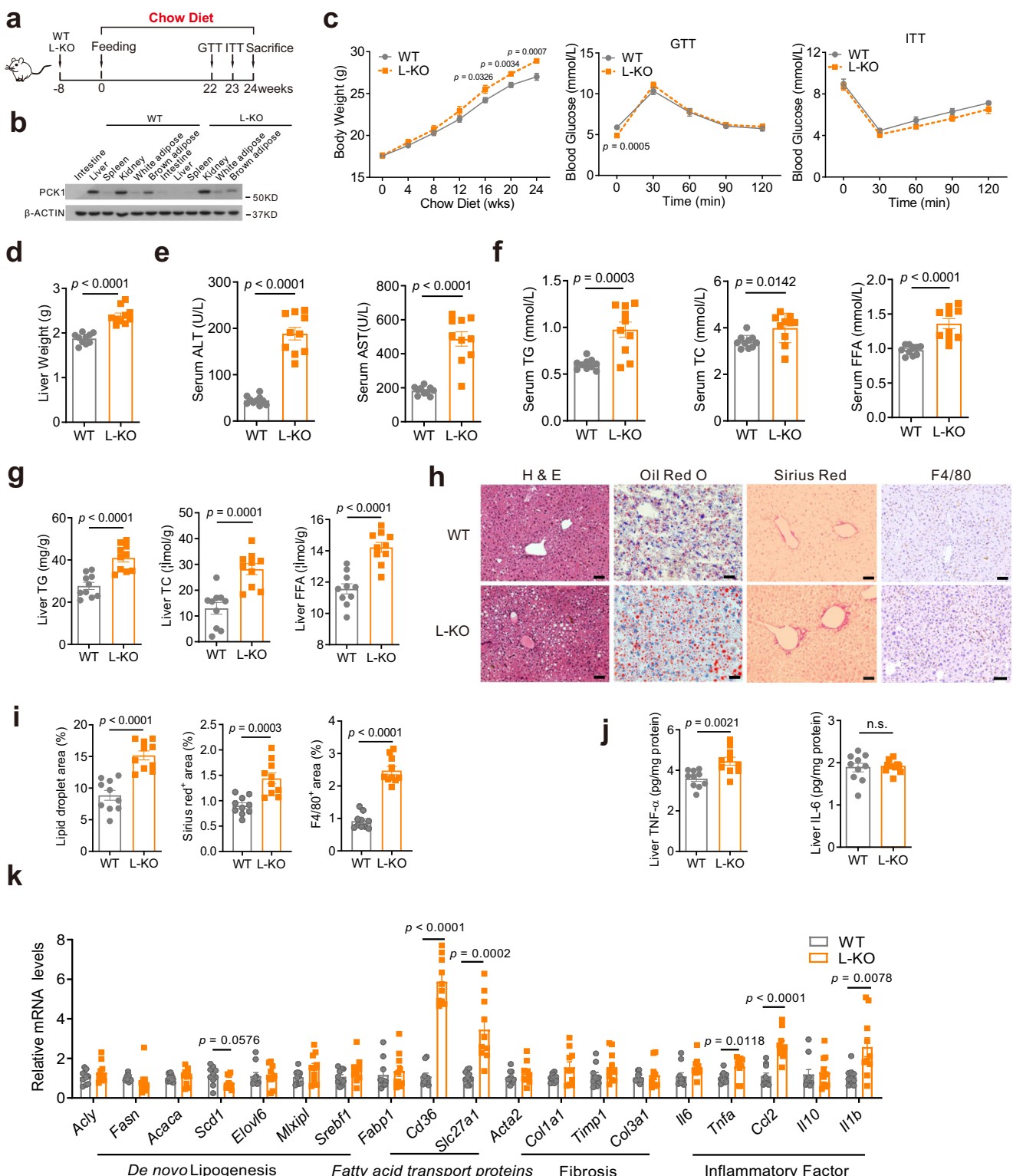

**Fig. 2 | L-KO mice fed chow diet exhibited a distinct hepatic steatosis phenotype. a** Schematic diagram of the mouse model fed the chow diet, $n = 10$/group. **b** PCK1 protein expression in WT and L-KO mouse intestine, liver, spleen, kidney, white adipose, and brown adipose confirmed by immunoblotting. The samples were derived from the same experiment and the blots were processed in parallel. This experiment was repeated for three times with similar results. **c** Body weight, glucose tolerance test (GTT), and insulin tolerance test (ITT) were measured in WT and L-KO mice ($n = 10$). **d** Liver weight of WT and L-KO mice ($n = 10$). **e–g** Determination of alanine aminotransferase (ALT), aspartate aminotransferase (AST), total triglycerides (TG), total cholesterol (TC), and free fatty acid (FFA) levels in the

serum or liver tissues ($n = 10$). **h** Paraffin-embedded liver sections were stained with hematoxylin and eosin (H&E), Sirius red, and F4/80. Frozen sections stained with Oil Red O. Scale bars: 50 μm. **i** Quantification of liver sections of WT and L-KO mice fed the chow diet ($n = 10$). **j** Levels of TNF-α and IL-6 in the liver tissues ($n = 10$). **k** Quantitative PCR analysis of liver mRNA expression ($n = 10$). For immunoblotting, the samples were derived from the same experiment and the blots were processed in parallel. n was the number of biologically independent mice. Data are expressed as the mean ± SEM; n.s., not significant. $p$ values obtained via two-tailed unpaired Student's $t$ tests. Source data are provided as a Source Data file.

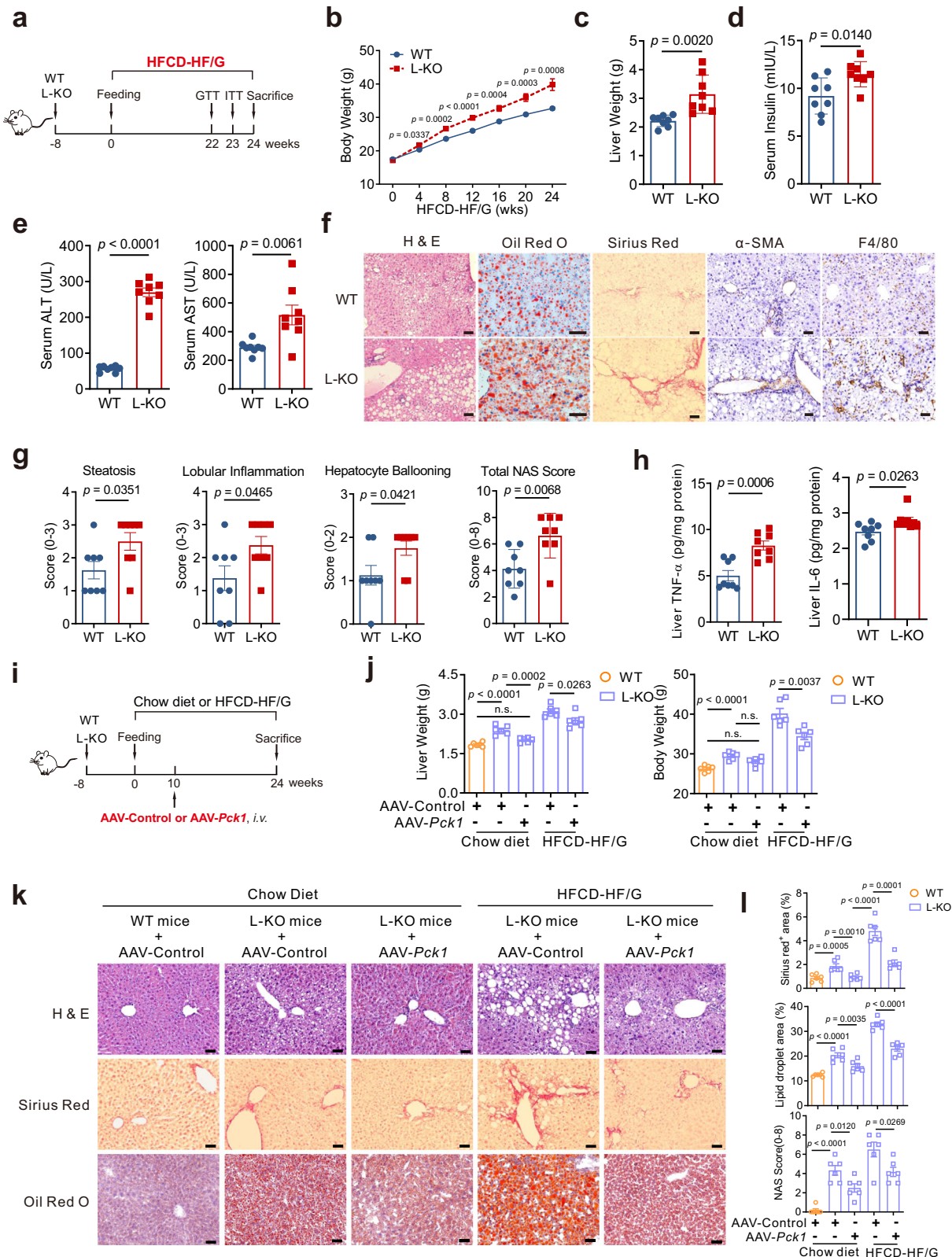

increased PDGF-AA expression in the liver tissues, primary hepatocytes, and plasma of L-KO mice (Fig. 6c–f). Moreover, the PDGF-AA concentration markedly increased in the culture medium of *PCK1*-KO cells but decreased in that of *PCK1*-OE cells treated with 0.2 mM PA (Fig. 6g, h). Correspondingly, platelet-derived growth factor receptor alpha (*PDGFRA*), which encodes the PDGF-AA receptor, increased in LX-2 cells co-cultured with *PCK1*-KO cells but decreased in LX-2 cells co-cultured

with *PCK1*-OE cells (Fig. 6i, j). To determine whether the pro-fibrogenic effect was mediated by PDGF-AA secretion, we treated the cells with a neutralizing antibody against PDGF-AA. As expected, the increases in α-SMA, COL1A1, and COL3A1 in LX-2 cells co-cultured with *PCK1*-KO cells were reversed by anti-PDGF-AA treatment (Fig. 6k).

A review of the transcriptome data showed that *Pdgfa* appeared in the heat map of the PI3K/AKT pathway (Supplementary Fig. 6b). The

**Fig. 3 | PCK1 ablation accelerates inflammation and fibrogenesis in MAFLD model. a** Schematic diagram of mouse model fed the HFCD-HF/G. **b, c** Body weight ($n = 11$ per group) and liver weight ($n = 8$ per group) were measured in WT and L-KO mice. **d, e** Serum levels of insulin, alanine aminotransferase (ALT), and aspartate aminotransferase (AST) were measured ($n = 8$). **f** Paraffin-embedded liver sections were stained with hematoxylin and eosin (H&E), Sirius red, α-SMA immunostaining, and F4/80 immunostaining. Frozen sections stained with Oil Red O. Scale bars: 50 μm. **g** NAFLD activity scores (NAS) for each group ($n = 8$). **h** Levels

of TNF-α and IL-6 in the liver tissues detected using enzyme-linked immunosorbent assay (ELISA) ($n = 8$). **i** Schematic showing the administration protocol for AAV8-TBG-*control* or AAV8-TBG-*Pck1* in WT and L-KO mice for experiments shown in **j–l**, $n = 6$/group. **j** Liver weight and body weight. **k** Paraffin-embedded liver sections were stained with H&E, Sirius red staining and Oil Red O staining. Scale bars: 50 μm. **l** Quantifications of Sirius red staining, Oil Red O staining, and NAS. Data are expressed as the mean ± SEM; n.s., not significant. *p* values obtained via two-tailed unpaired Student's *t* tests. Source data are provided as a Source Data file.

IHC results showed that p-AKT (S473) was positively correlated with PDGF-AA (Fig. 6l). The AKT inhibitor MK2206 significantly blocked the increase in *PDGFA* levels in the supernatants or cells lysates of *PCK1*-KO cells (Fig. 6m–o). Taken together, these data confirm that PCK1 deficiency promoted PDGF-AA expression through the PI3K/AKT pathway and activated HSCs through hepatocyte-HSC crosstalk.

## PCK1 deficiency promotes the activation of the PI3K/AKT/PDGF-AA axis by activating RhoA signaling in hepatocytes

Rho GTPases, which cycle between active GTP-bound and inactive GDP-bound conformations, activate the PI3K/AKT pathway[23–25]. Considering that PCK1 catalyzes the conversion of oxaloacetate to phosphoenolpyruvate, consuming GTP to generate GDP, we predicted that PCK1 deficiency alters intracellular GTP homeostasis. To test this hypothesis, high-performance liquid chromatography (HPLC) analysis was conducted to detect the intracellular levels of GTP. Interestingly, intracellular GTP levels decreased in *PCK1*-OE cells (Supplementary Fig. 7a) but increased in *PCK1*-KO cells (Supplementary Fig. 7b). Considering that Rho GTPases are activated when combined with GTP[26], we examined the proteins levels of several Rho GTPases in the mouse liver tissues and found that GTP-bound RhoA significantly increased and inactivated RhoA, p-RhoA (S188), decreased in L-KO mice (Fig. 7a–c). Similar results were observed in the primary hepatocytes of mice fed the HFCD-HF/G (Supplementary Fig. 7c). Consistently, after PA treatment, the levels of GTP-bound RhoA decreased and p-RhoA (S188) expression increased in *PCK1*-OE cells, whereas the opposite results were observed in *PCK1*-KO cells (Fig. 7d–g). Moreover, we found that the addition of 5′-GTP, 2Na⁺ (salt of guanosine triphosphate) activated the RhoA/PI3K/AKT pathway in primary hepatocytes (Supplementary Fig. 7d). However, inhibition of de novo guanine nucleotide synthesis by mycophenolic acid (MPA) robustly inhibited the RhoA/PI3K/AKT pathway by decreasing the extent of phosphorylated AKT (Supplementary Fig. 7e). Next, RhoA inhibitor (Rhosin) or shRhoA was used to determine whether PI3K/AKT activation depends on RhoA. Immunoblotting, ELISA, and qPCR assays showed that both Rhosin and shRhoA blocked the increase in the activated forms of AKT and PDGF-AA in the *PCK1*-KO cell lysate and supernatant, as well as *ACTA2*, *COL1A1*, and *COL3A1* expression in LX-2 cells co-cultured with *PCK1*-KO hepatocytes (Fig. 7h–j, Supplementary Fig. 7f–h). To further evaluate the involvement of RhoA in HSC activation, we isolated primary hepatocytes from HFCD-HF/G-fed WT or L-KO mice; treated the cells with MK2206, Rhosin, or dimethyl sulfoxide vehicle; and then co-cultured the cells with primary HSCs from chow-fed WT mice. The activation of co-cultured HSCs was partially eliminated by the inhibition of AKT or RhoA in primary hepatocytes (Supplementary Fig. 7i). Moreover, PCK1 and p-RhoA (S188) were downregulated in samples from patients with NASH, whereas p-AKT (S473) and PDGF-AA levels were upregulated (Fig. 7k, l). These data indicate that PCK1 ablation stimulated the PI3K/AKT/PDGF-AA axis by activating RhoA.

## Genetic or pharmacological disruption of RhoA and AKT1 reduced progressive liver fibrosis in vivo

To explore whether blocking RhoA/PI3K/AKT could rescue the MAFLD phenotype in L-KO mice, the genetic and pharmacological disruption of RhoA and AKT1 was performed in vivo (Fig. 8a). Pharmacological

inhibition of AKT1 or RhoA led to improved glucose intolerance (Supplementary Fig. 8a) and insulin resistance (Supplementary Fig. 8b). The increase in liver weight was also prevented (Fig. 8b), whereas the body weight decreased only in the MK2206 treatment group (Supplementary Fig. 8c). Additionally, Rhosin or MK2206 administration attenuated the levels of inflammatory factors (e.g. AST, ALT, IL-6, and TNF-α) as well as TG and FFA levels, in the serum and liver tissues (Fig. 8c, Supplementary Fig. 8d, e). Similarly, histochemistry revealed reduced liver steatosis, inflammation, and fibrosis in Rhosin- or MK2206-treated mice (Fig. 8d, Supplementary Fig. 8f). Additionally, α-SMA, COL1A1, COL3A1, PDGF-AA, and p-AKT (S473, T308) expression and GTP-bound RhoA levels decreased, whereas p-RhoA (S188) expression increased in the treatment group (Supplementary Fig. 8g, h). MK2206 or Rhosin treatment also reduced the expression of genes related to inflammation and fibrosis (Supplementary Fig. 8i). These results indicate that the pharmacological inhibition of AKT or RhoA can alleviate the clinical phenotypes of MAFLD.

Next, we used pSECC, a lentiviral-based system that combines the CRISPR system and Cre recombinase, to silence RhoA or AKT1 in L-KO mice[27]. In agreement with the results of the experiments described above, AKT or RhoA depletion in vivo partially prevented and liver weight and body weight gain (Fig. 8e). Moreover, serum AST, ALT, TC, TG, and FFA levels significantly decreased in sgAkt1- or sgRhoA-treated mice compared with those in sgCtrl-treated L-KO mice (Fig. 8f, Supplementary Fig. 9a). Histological analysis revealed that the NAS score, hepatic fibrosis, and F4/80⁺ macrophage counts decreased in sgAkt1 and sgRhoA mice, indicating the attenuation of the MAFLD phenotypes (Fig. 8g, Supplementary Fig. 9b, c). Consistently, genes or proteins involved in hepatic inflammation and fibrosis were significantly downregulated in the livers of sgAkt1 and sgRhoA mice, accompanied by the reduced expression of PDGF-AA (Supplementary Fig. 9d, e). These results indicate that the genetic inhibition of AKT or RhoA can alleviate the clinical phenotypes of MAFLD.

## Discussion

We found that the hepatic gluconeogenic enzyme PCK1 plays an important role in MAFLD progression. The expression of PCK1 was diminished in the livers of patients or mice with MAFLD. Moreover, the deletion of PCK1 significantly exacerbated hepatic steatosis, fibrosis, and inflammation in mouse models fed the HFCD-HF/G. Mechanistically, loss of PCK1 not only promotes steatosis by enhancing lipid deposition, but also induces fibrosis through HSC activation via paracrine secretion of PDGF-AA, thus promoting MAFLD progression (Fig. 8h).

Abnormal lipid metabolism is characteristic of MAFLD. Previous studies predicted that altered lipid homeostasis was caused by abnormal expression of genes related to lipid metabolism[28]. However, recent studies demonstrated that disruption of gluconeogenesis also leads to abnormal lipid metabolism. A deficiency of fructose-1,6-bisphosphatase 1 and glucose-6-phosphatase catalytic subunit, which are key enzymes in gluconeogenesis, results in severe hepatic steatosis and hypoglycemia, indicating that the suppression of gluconeogenesis also disrupts lipid homeostasis[17,29]. As the first rate-limiting enzyme in gluconeogenesis, it is currently unclear whether PCK1 plays a critical

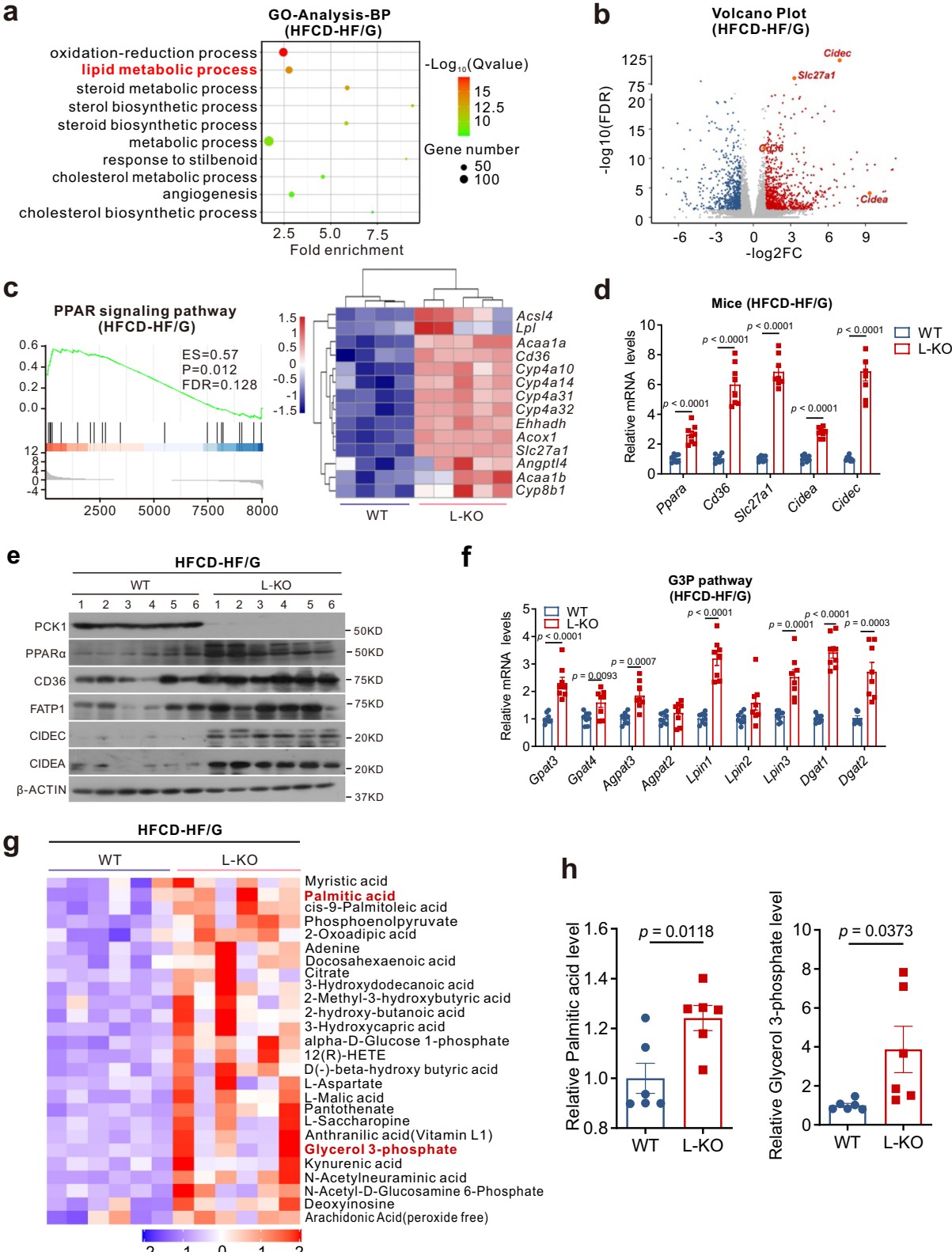

**Fig. 4 | Loss of PCK1 promotes lipid accumulation according to transcriptome and metabolome analyses.** RNA sequencing was performed on the livers of WT ($n = 4$) and L-KO ($n = 5$) mice fed the HFCD-HF/G. **a** Gene Ontology analysis of all significantly changed genes in top 10 biological processes. **b** Volcano plot representation of significantly up- and downregulated genes. **c** Gene Set Enrichment Analysis plot (left) of enrichment in "PPAR signaling pathway" signature; heatmap (right) of significantly upregulated PPAR target genes. **d, e** Quantitative PCR (**d**) and immunoblot (**e**) analysis of indicated genes or protein expression in mouse liver tissues. **f** Relative mRNA expression of key genes in G3P pathway ($n = 8$). **g** Upregulated metabolites detected by untargeted metabolomics ($n = 6$). **h** Relative level of G3P and PA in mouse liver tissues ($n = 6$). Data are expressed as the mean ± SEM. $p$ values obtained via two-tailed unpaired Student's $t$ tests. Source data are provided as a Source Data file.

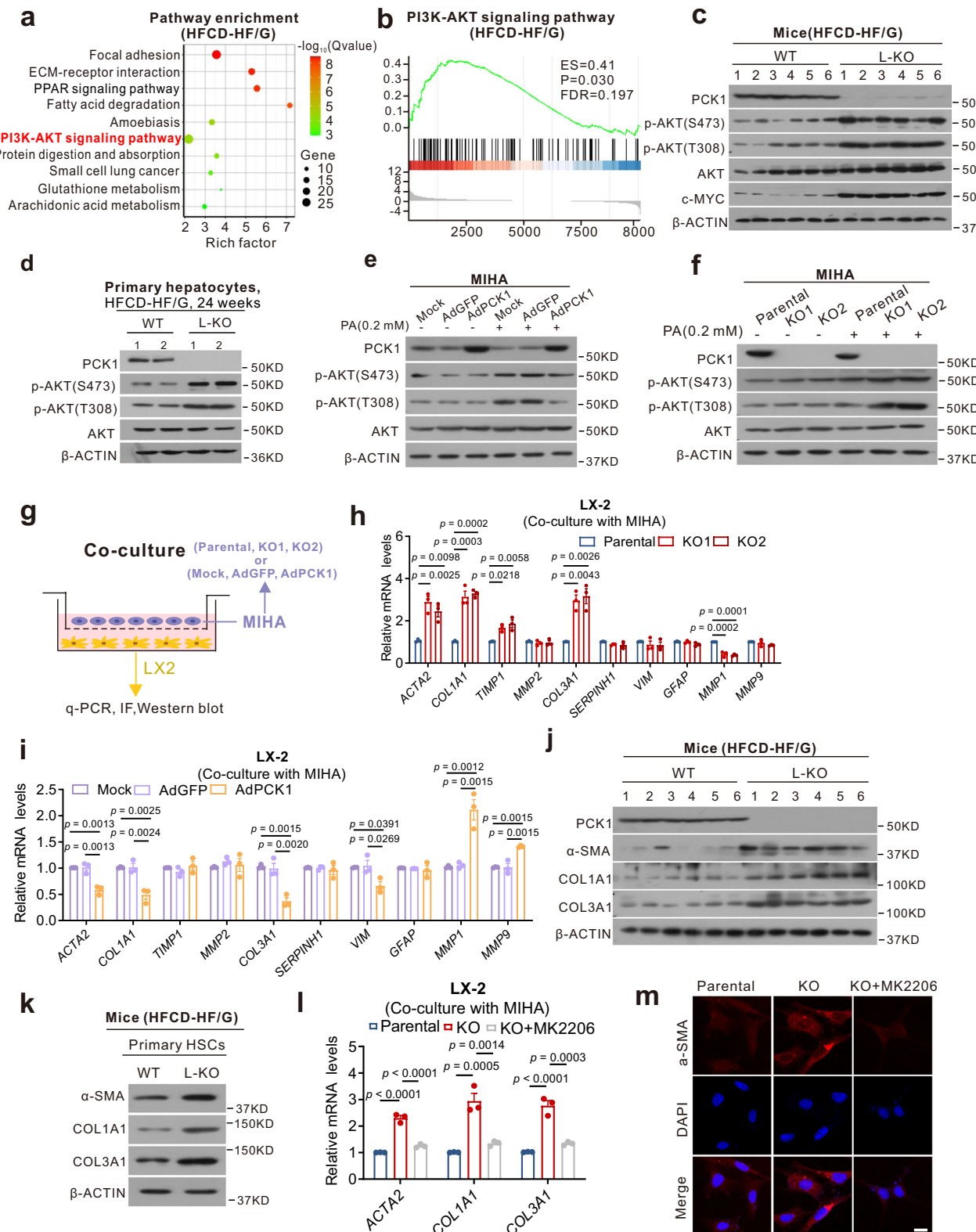

role in MAFLD development. We identified a robust decrease in PCK1 expression in the livers of MAFLD mice and patients with NAFLD/NASH, causing severe hepatic steatosis and confirming that disordered hepatic gluconeogenesis affects lipid homeostasis.

Previous reports showed that PCK1 expression is increased in several obesity/diabetes mouse models, such as ZDF rats and *ob/ob* and *db/db* mice, and the disease progression of MAFLD is positively

correlated with obesity and type 2 diabetes mellitus[30–32]. Interestingly, we found that PCK1 expression was downregulated in a diet-induced murine model. This discrepancy may be related to differences in the animal models used in different studies. Widely used rodent models for genetic forms of obesity and diabetes, such as *ob/ob* and *db/db* mice, exhibit increased plasma glucocorticoids, which may drive PCK1 expression[32,33]. Another explanation is that the high-fat diet

**Fig. 5 | Hepatic PCK1 deficiency leads to HSC activation via PI3K/AKT pathway.**
**a** Pathway enrichment analysis of significantly upregulated genes in L-KO mice.
**b** Gene Set Enrichment Analysis (GSEA) plot of enrichment in PI3K/AKT pathway.
**c**−**f** Immunoblot analysis of AKT and p-AKT (S473 or T308) in mouse liver tissues
(**c**), primary hepatocytes from HFCD-HF/G feeding mice (**d**), *PCK1*-OE (**e**), and *PCK1*-
KO (**f**) MIHA cells with or without 0.2 mM palmitic acid (PA) treatment. The samples
were derived from the same experiment and the blots were processed in parallel.
**g** Schematic flow chart of co-culture models. **h, i** Quantitative PCR analysis of

fibrosis-related genes in LX-2 cells co-cultured with *PCK1*-KO (**h**) or *PCK1*-OE (**i**)
MIHA cells. **j, k** Western blotting of fibrosis-related protein in liver tissues (**j**) or
primary HSCs from HFCD-HF/G feeding mice (**k**) (*n* = 3) **l, m** Relative mRNA
expression (**l**) and immunofluorescence images (**m**) of *ACTA2*/α-SMA, *COL1A1*, and
*COL3A1* in LX-2 cells co-cultured with *PCK1*-KO MIHA cells treated with AKT inhi-
bitor MK2206 (10 µM). Scale bars: 25 µm. Data are expressed as the mean ± SEM.
*p* values obtained via two-tailed unpaired Student's *t* tests or one-way ANOVA with
Tukey's post hoc test. Source data are provided as a Source Data file.

supplemented with high fructose/glucose in drinking water might
suppress PCK1 expression[34]. Under a high-fat diet, PA decreased PCK1
expression via SIRT3 inhibition[35]. Additionally, acetylation, ubiquiti-
nation, and phosphorylation modulate PCK1 expression[36]. A high glu-
cose level was reported to destabilize PCK1 by stimulating its
acetylation, thus promoting its ubiquitination and subsequent
degradation[37]. In this study, we found that ATF3, a member of the basic
leucine zipper family of transcription factors[38], transcriptionally
repressed *Pck1* upon PA overload in vitro and in the mouse model. This
result agrees with those of previous studies suggesting that ATF3 is
upregulated in patients with NAFLD and murine models and inhibits
the expression of PCK1 in alcoholic fatty liver disease[17,39,40]. Therefore,
in this study, we found that PCK1 markedly decreased in MAFLD, and
PA inhibited *PCK1* transcription by upregulating ATF3.

Numerous studies using PCK1 agonists or whole-body *Pck1*
knockdown mice have verified that PCK1 can affect lipid
metabolism[41,42]. In the present study, liver-specific *Pck1* knockout
induced significant hepatic steatosis even under normal feeding con-
ditions. This is very important because it is uncommon for single-gene
ablation to cause spontaneous steatosis unless a high-fat diet is used.
Moreover, mice with liver *Pck1* deficiency present aggravated inflam-
mation when fed a high-fat high-fructose diet, contrasting with the
results of a previous study showing that whole-body *Pck1* knockdown
prevents hepatic inflammation[43]. This discrepancy may be related to
differences between diets and animal models, as whole-body *Pck1*
knockdown may have unexpected effects on glucolipid metabolism.
The macronutrient composition of the diet, the duration of feeding,
and the genetic background of the animals are all important variables
in determining disease severity in preclinical MAFLD models[44]. To
establish an ideal MAFLD model, several attempts have been made to
use HFD in combination with fructose and sucrose-enriched drinking
water[45,46]. In this study, we have constructed the MAFLD model using
the mouse strain derived from the C57BL/6 J and 129S6/SvEvTac, which
may serve as a suboptimal model to investigate advanced stages of
MAFLD. In light of the fact that MAFLD prevalence is higher in men
than in women at all ages[47], we initially used male mice, but not female
mice, in the present studies; therefore, the generalizability of the
findings to female mice is not guaranteed.

Lipid accumulation is characteristic of steatosis. Emerging evi-
dence indicates that increased fatty acid uptake is associated with lipid
accumulation[48,49]. Previous studies demonstrated that the loss of PCK1
in the liver disturbed hepatic cataplerosis and led to the accumulation
of TCA cycle intermediates. The slowed TCA cycle impaired fatty acid
oxidation, resulting in fat accumulation in the liver[50]. Additionally,
elevated plasma TGs and FFAs could contribute to body weight gain in
L-KO mice. Notably, HFCD-HF/G-induced glucose intolerance and
insulin resistance in L-KO mice play essential roles in obesity
and whole-body metabolism. In this study, genes involved in fatty acid
uptake such as *Cd36* and *Slc27a1* were highly expressed in L-KO mice.
In addition, a lipid droplet-associated protein Cidec was increased by
both the chow and HFCD-HF/G diet, and was shown to be upregulated
in patients with NAFLD and L-KO mice, suggesting that PCK1 ablation
promotes lipid droplet formation[51,52]. Abnormal levels of metabolites
also contribute to TG accumulation in the liver, with the G3P pathway
contributing to over 90% of TG synthesis[53]. As our metabolomics data

showed that G3P and PA were significantly upregulated in L-KO mice,
PCK1 deficiency may promote hepatic lipid accumulation by enhan-
cing the expression of Cd36, Slc27a1, and Cidec and the levels of
metabolic substrates such as G3P and PA. However, the precise
mechanism by which PCK1 regulates the G3P pathway and expression
levels of *Cd36* and *Slc27a1* must be further analyzed.

Fibrosis is another characteristic of MAFLD and drives the tran-
sition from simple steatosis to NASH. Activation of HSCs through the
secretion of profibrotic cytokines, such as TGF-β and PDGF, is a key
event in liver fibrosis[54]. A recent study identified high mobility group
protein B1, secreted by fructose-1,6-bisphosphatase 1-deficient hepa-
tocytes, as the main mediator activating HSCs, revealing important
crosstalk between hepatocytes and HSCs via paracrine signaling.
Herein, PDGF-AA was secreted by PCK1-deficient hepatocytes and
acted in a paracrine manner to activate HSCs. Increased deposition of
ECM and activation of HSCs were observed in PDGFA-transgenic mice;
however, the mechanism mediating PDGF-AA upregulation in fibrosis
remains unclear[22]. Here, we demonstrated that PCK1 deficiency pro-
moted PDGF-AA secretion by activating the RhoA/PI3K/AKT pathway.
Mechanistically, PCK1 deletion may increase intracellular GTP levels,
thus promoting the activation of RhoA and further activating the PI3K/
AKT pathway.

Most Rho GTPases cycle between an active GTP-bound and an
inactive GDP-bound form, a process that is regulated by guanine
nucleotide exchange factors, GTPase-activating proteins, and guanine
nucleotide dissociation inhibitors. Guanine nucleotide exchange fac-
tors can activate Rho GTPases by catalyzing the exchange of GDP for
GTP when the intracellular concentration of GTP is high[55]. Several
members of the Rho-GTPase family, such as Rac1, RhoA, and RhoC, can
be activated by increased concentrations of intracellular GTP[56−59]. We
found that PCK1 deficiency activated RhoA by increasing the levels of
intracellular GTP, therefore activating the downstream PI3K/AKT
pathway. Moreover, the genetic and pharmacological disruption of
RhoA and AKT1 can effectively mitigate MAFLD phenotypes in L-KO
mice. In addition, RhoA and AKT inhibitors can reportedly inhibit the
progression of MAFLD through other pathways, such as the NF-κB[60],
Hippo[61,62], and Notch[63] signaling pathways. However, off-target effects
of these inhibitors cannot be completely ruled out. RhoA and AKT
inhibitors are currently only in Phase 3 trials or preclinical studies for
the treatment of liver fibrosis or clinical tumors, with therapeutic
potential for MAFLD[64−66].

In conclusion, hepatic PCK1 deficiency promoted lipid deposition
and fibrosis in a murine MAFLD model. Moreover, hepatic PCK1 loss
activated the RhoA/PI3K/AKT pathway, which increased PDGF-AA
secretion and promoted HSC activation in male mice. AKT/RhoA
inhibitors reduced progressive liver fibrosis, providing a potential
therapeutic strategy for MAFLD treatment.

## Methods
### Animal models
Animal experiments were approved by the Animal Experimentation
Ethics Committees of Chongqing Medical University and performed in
accordance with the Guide for the Care and Use of Laboratory Animals.
*Pck1^f/f^* mice on a 129S6/SvEv background were purchased from the
Mutant Mouse Resource & Research Center (MMRRC: 011950-UNC;

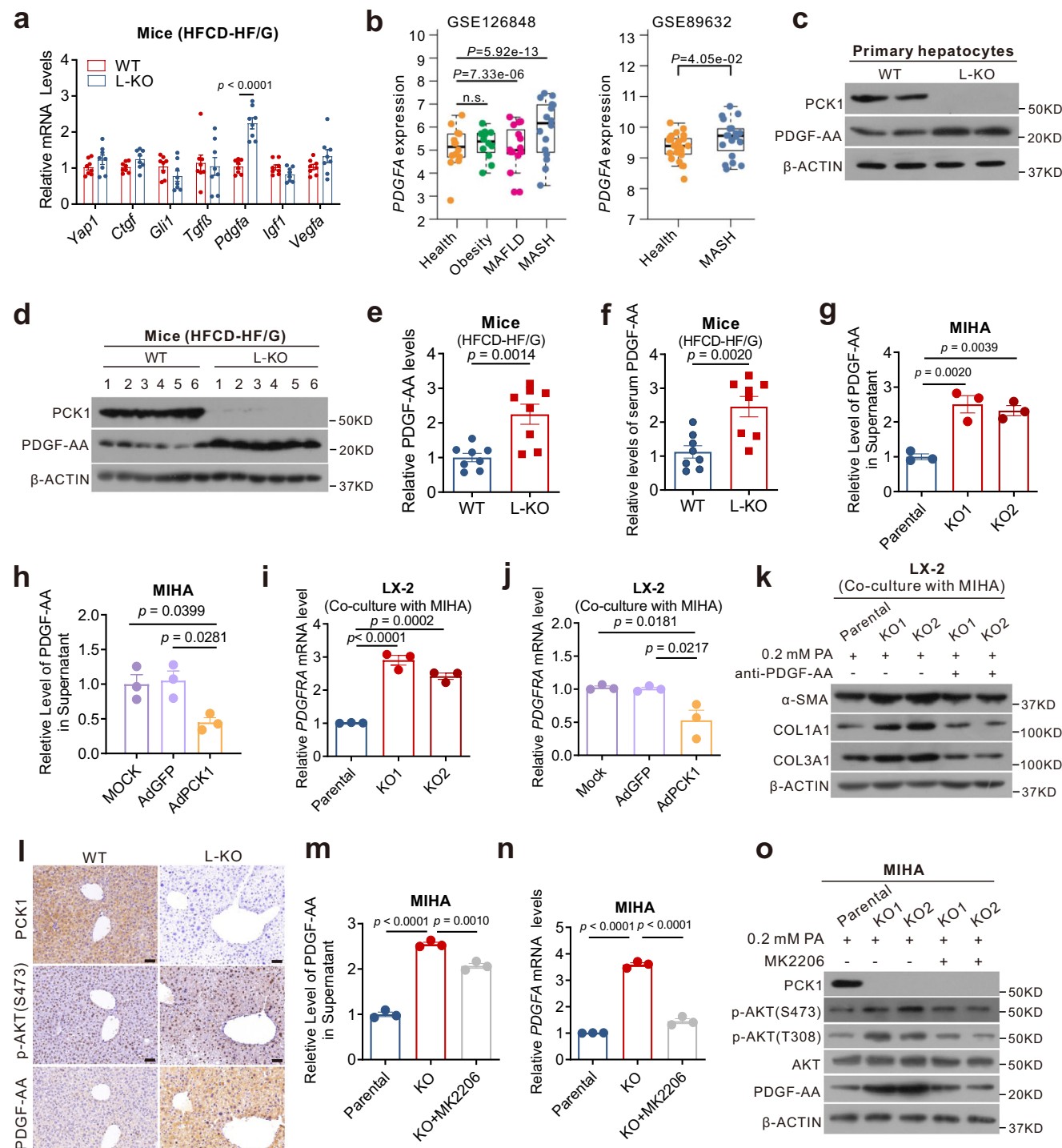

**Fig. 6 | Paracrine PDGF-AA from hepatocytes promotes hepatic stellate cell (HSC) activation. a** Expression levels of genes related to fibrogenesis (*n* = 8 for each group). **b** Relative *PDGFA* mRNA levels of health (*n* = 14), obesity (*n* = 12), NAFLD (*n* = 15), and NASH (*n* = 16) in GSE126848 and of health (*n* = 19), and NASH (*n* = 24) in GSE89632 datasets. The box plots show the medians (middle line) and the first and third quartiles (boxes), whereas the whiskers show 1.5× the IQR above and below the box. Unpaired, two-sided Mann–Whitney U test *P* values are depicted in the plots, and the significant *P* value cutoff was set at 0.05. **c, d** PDGF-AA protein levels in primary hepatocytes (**c**) or liver tissues (**d**) detected by western blotting. The samples were derived from the same experiment and the blots were processed in parallel. **e, f** PDGF-AA levels in liver tissues (**e**) or serum (**f**) were detected using enzyme-linked immunosorbent assay (ELISA) (*n* = 8). **g, h** Secreted PDGF-AA levels in conditioned medium with *PCK1*-KO (**g**) or *PCK1*-OE (**h**) MIHA cells treated with

0.2 mM palmitic acid (PA). **i, j** mRNA levels of *PDGFRA* in cell lysate of LX-2 cells co-cultured with *PCK1*-KO (**i**) or *PCK1*-OE (**j**) MIHA cells treated with PA. **k** Protein level in LX-2 cells co-cultured with *PCK1*-KO MIHA cells containing nonspecific rabbit IgG or a PDGF-AA blocking antibody. **l** Immunohistochemistry (IHC) analysis of PCK1, p-AKT (S473), and PDGF-AA in mouse liver sections (from serial sections). Scale bars: 50 μm. **m, n** Levels of PDGF-AA (**m**) or *PDGFA* (**n**) in conditioned medium or cell lysate of *PCK1*-KO MIHA cells treated with AKT inhibitor MK2206 (10 μM). **o** Protein levels in *PCK1*-KO MIHA cells treated with AKT inhibitor MK2206 (10 μM). The samples were derived from the same experiment and the blots were processed in parallel. For **g, h, i, j, m** and **n**, *n* = 3. Data are expressed as the mean ± SEM; n.s., not significant. *p* values obtained via two-tailed unpaired Student's *t* tests or one-way ANOVA with Tukey's post hoc test. Source data are provided as a Source Data file.

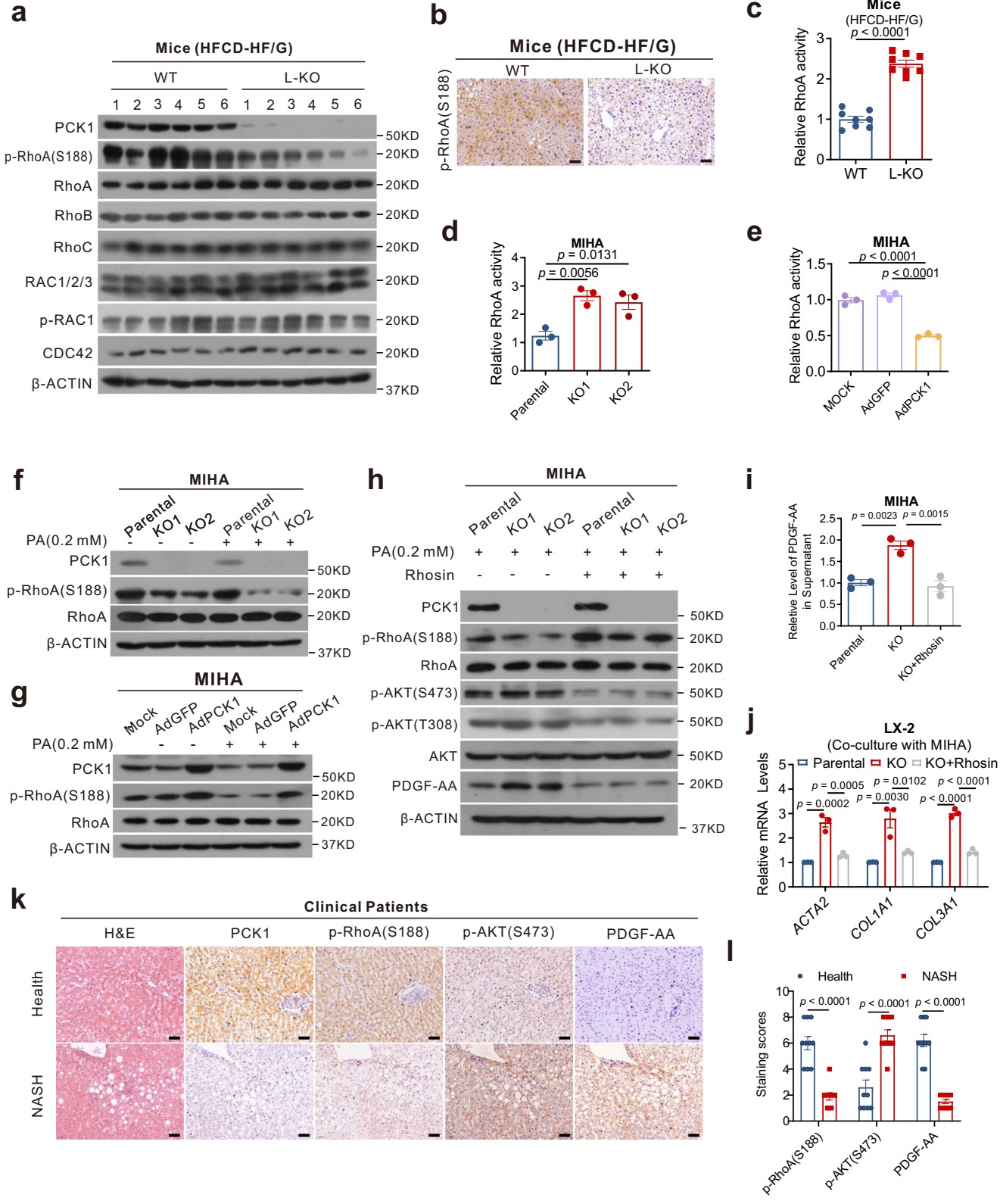

Bar Harbor, ME, USA) and *Alb-Cre* mice on a C57BL/6 background were purchased from Model Animal Research Center of Nanjing University (Nanjing, China). To generate liver-specific *Pck1*-knockout mice (L-KO), *Alb-Cre* mice were crossed with *Pck1*^f/f mice. *Pck1*^f/f mice from the same breeding step were used as controls (wild-type, WT). Male WT and L-KO mice at 7–9 weeks old were fed the HFCD-HF/G (D12492: 60% Kcal fat, with drinking water containing 23.1 g/L fructose and 18.9 g/L

glucose; Research Diets, New Brunswick, NJ, USA) (n = 11 per group) or control chow diet (D12450J: 10% Kcal fat, with tap water; Research Diets) (n = 10 per group) for 24 weeks. Food and drinking water were provided *ad libitum*. Alb-Cre; *Pten*^f/f (c*Pten*^f/f) mice (kindly provided by Prof. Yujun Shi, Sichuan University, Chengdu, China) were on a C57BL/6 background and genotyping of the mice was performed as previously described[67]. L-KO mice were crossed with c*Pten*^f/f mice to

**Fig. 7 | PCK1 deficiency promotes activation of PI3K/AKT/PDGF-AA axis by activating RhoA in hepatocytes. a** Immunoblotting analysis of indicated proteins in mouse liver tissues. The samples were derived from the same experiment and the blots were processed in parallel. **b** Immunohistochemistry (IHC) analysis of p-RhoA (S188) in mouse liver tissues. Scale bars: 50 μm. **c–e** Relative levels of active RhoA were measured using G-LISA colorimetric RhoA activation assay in mouse liver tissues (**c**) ($n = 8$), *PCK1*-KO (**d**) and *PCK1*-OE (**e**) MIHA cells treated with 0.2 mM palmitic acid (PA). **f, g** Immunoblots of p-RhoA (S188) and RhoA in *PCK1*-KO (**f**) and *PCK1*-OE (**g**) MIHA cells with or without 0.2 mM PA treatment. The samples were derived from the same experiment and the blots were processed in parallel. **h** Expression of indicated proteins in *PCK1*-KO MIHA cells after addition of Rhosin (30 μM). **i** Levels of PDGF-AA in the supernatant of *PCK1*-KO MIHA cells treated with Rhosin (30 μM). **j** Relative mRNA expression of *ACTA2*, *COL1A1*, and *COL3A1* in LX-2 cells co-cultured with *PCK1*-KO MIHA cells treated with Rhosin (30 μM). **k** IHC analysis of PCK1, p-RhoA (S188), p-AKT (S473), and PDGF-AA in normal individuals and patients with NASH (from serial sections). Scale bars: 50 μm. **l** Semi-quantitative analyses of immunohistochemistry data of health and NASH human tissues for indicated proteins (health, $n = 10$, NASH, $n = 10$). The cell culture experiments were repeated for three times independently with similar results. For **d**, **e**, **i** and **j**, $n = 3$. Data are expressed as the mean ± SEM. *p* values obtained via one-way ANOVA with Tukey's post hoc test. Source data are provided as a Source Data file.

generate Alb-Cre; *Pten*^f/+^*Pck1*^f/+^ (c*Pten*^f/+^*Pck1*^f/+^). c*Pten*^f/+^*Pck1*^f/+^ mice were crossed to breed Alb-Cre; *Pten*^f/f^*Pck1*^f/f^ (c*Pten*^f/f^*Pck1*^f/f^) mice. Littermates that were negative for the Cre transgene were used as WT controls. Primers used for the *Pck1*, *Pten* and *Cre* have been provided in Supplementary Table 2. Male mice were used in all experiments. All mice were housed in temperature-controlled (23 °C) pathogen-free facilities with a 12 h light-dark cycle and humidity (50% ± 10%) conditions.

For AAV8 transduction, AAV8-TBG-*control* and AAV8-TBG-*Pck1* were purchased from the Shanghai Genechem Co., Ltd. (Shanghai, China) and injected via the tail vein following 10 weeks of HFCD-HF/G feeding ($2 \times 10^{11}$ genome copies/mouse). The efficiency of virus infection and expression of PCK1 in mouse hepatocytes were confirmed by western blot analysis. After a total of 24 weeks of HFCD-HF/G feeding, the mice were sacrificed for analysis.

Genetic and pharmacological inhibition of AKT1 or RhoA were performed in vivo. Genetic depletion of mouse RhoA or AKT1 in vivo was conducted using pSECC (#60820; Addgene, Watertown, MA, USA), a lentiviral-based system that combined both the CRISPR system and Cre recombinase. After HFCD-HF/G feeding for 16 weeks, male L-KO mice were injected with pSECC-sgCtrl, pSECC-sgAkt1, or pSECC-sgRhoA through the tail vein at $1 \times 10^9$ genome copies per mouse, with two booster injections at 7 day intervals. The pharmacological inhibition of AKT1 or RhoA was conducted using MK2206 or Rhosin. After HFCD-HF/G feeding for 16 weeks, the mice were divided into 3 groups and intraperitoneally injected with vehicle solution ($n = 6$), MK2206 (AKT inhibitor, 50 mg/kg, every 3 days) ($n = 5$), or Rhosin (RhoA inhibitor, 20 mg/kg, every 3 days) ($n = 6$) for 8 weeks. All mice were sacrificed for further study after HFCD-HF/G feeding for 24 weeks.

### Liver tissues from patients with NASH
Liver tissue collection was approved by the Institutional Ethics Committees of Chongqing Medical University and Xin Hua Hospital (project license number: XHEC-C-2012-023). Informed consent was obtained from all participants. Paraffin-embedded normal ($n = 10$) and NASH human liver samples ($n = 36$) were kindly provided by Dr. Jiangao Fan, Dr. Xiaojun Wang, and Dr. Yalan Wang. Human liver samples with NASH were obtained either by liver biopsy for diagnostic purpose or from surgical liver resections. All liver specimens were evaluated independently by three experienced pathologists, who are blinded to clinical data, according to the NAFLD activity score (NAS), defined as the sum of steatosis, inflammation and hepatocyte ballooning. Patients with a NAS score ≥5 were considered likely to have NASH[68]. Normal control samples were recruited from samples obtained for the exclusion of liver malignancy during major oncological surgery. Exclusion criteria were the presence of other causes of liver disease, including alcoholic fatty liver disease (>30 g/day for men, >20 g/day for women), chronic infection with hepatitis B and/or C virus, primary biliary cirrhosis, haemochromatosis, autoimmune hepatitis, and Wilson's disease, as well as the use of anti-obesity, glucose-lowering, and/or lipid-lowering

pharmacological treatments. The general characteristics of the NASH human liver samples are listed in Supplementary Table 1.

### Cell culture and treatment
All cell lines were grown in Dulbecco's modified eagle medium (DMEM) supplemented with 10% fetal bovine serum, 100 μg/mL of streptomycin, and 100 U/mL of penicillin at 37 °C in 5% $CO_2$. All cells were negative for mycoplasma. Short tandem repeat tests were performed to ensure the authenticity of the cells. Bovine serum albumin (BSA, 10%), different concentrations of palmitic acid (PA), 10 μM MK2206 (AKT inhibitor), 40 μM Rhosin (RhoA inhibitor), or blocking antibody against PDGF-AA (2 μg/mL) was added to the medium. For in vitro co-culture assays, the human hepatic stellate cell (HSC) line LX-2 and human hepatocyte line MIHA cells (provided by Dr Ben C.B. Ko, The Hong Kong Polytechnic University, Hongkong, China) were used. LX-2 was pre-cultured in the lower chamber for 12 hours, and PCK1-OE or PCK1-KO cells (with or without indicated treatment) were seeded in Transwell inserts (#3401, Corning, NY, USA) that were subsequently loaded into the LX2-containing wells. The cells and supernatants were harvested for further analysis after 48 h of co-culture.

### Isolation of primary mouse hepatocytes
Primary hepatocytes were isolated and cultured from the livers of WT and L-KO mice fed the HFCD-HF/G as described previously[69]. Briefly, following anesthesia, the inferior vena cava was cannulated and the liver was perfused in situ with 40 mL pre-warmed EGTA solution and 40 mL solution containing 0.35 mg/mL pronase (no. P5147, Sigma-Aldrich, St. Louis, MO, USA), followed by 40 mL solution containing 0.55 mg/mL collagenase (no. V900893, Sigma-Aldrich). After perfusion, the liver was crushed, and hepatocytes were released into the DMEM. The cell suspension was filtered through a 100 μm cell strainer and centrifuged at 50 ×g at 25 °C for 3 min. After washing three times, the cells were suspended in DMEM supplemented with 10 mM glucose, 10% fetal bovine serum, 100 nM insulin (P3376, Beyotime Biotechnology, Shanghai, China), and 100 nM dexamethasone (D8040, Solarbio Life Sciences, Beijing, China), and then plated on 60 mm diameter plastic plates[69]. After cell attachment, the medium was replaced with serum-free media, and the cells were used for experiments on the following day.

### Isolation of primary mouse hepatocytes and primary HSCs
HSCs were isolated from the mice as described previously[70]. Briefly, after perfusion with solutions containing protease and collagenase, the crushed liver was digested with a solution containing 1% DNase (10104159001, Roche Diagnostics GmbH, Mannheim, Germany), 0.5 mg/mL protease, and 0.55 mg/mL collagenase for 25 min. The cell suspension was filtered through a 70 μm cell strainer, centrifuged at 580 × g for 10 min at 4 °C, and washed twice with Gey's balanced salt solution (GBSS). The cells were subjected to gradient centrifugation on a 9.7% Nycodenz (1002424, Axis-Shield, Oslo, Norway) to isolate HSCs, which were then plated onto collagen-coated plates. The cells were

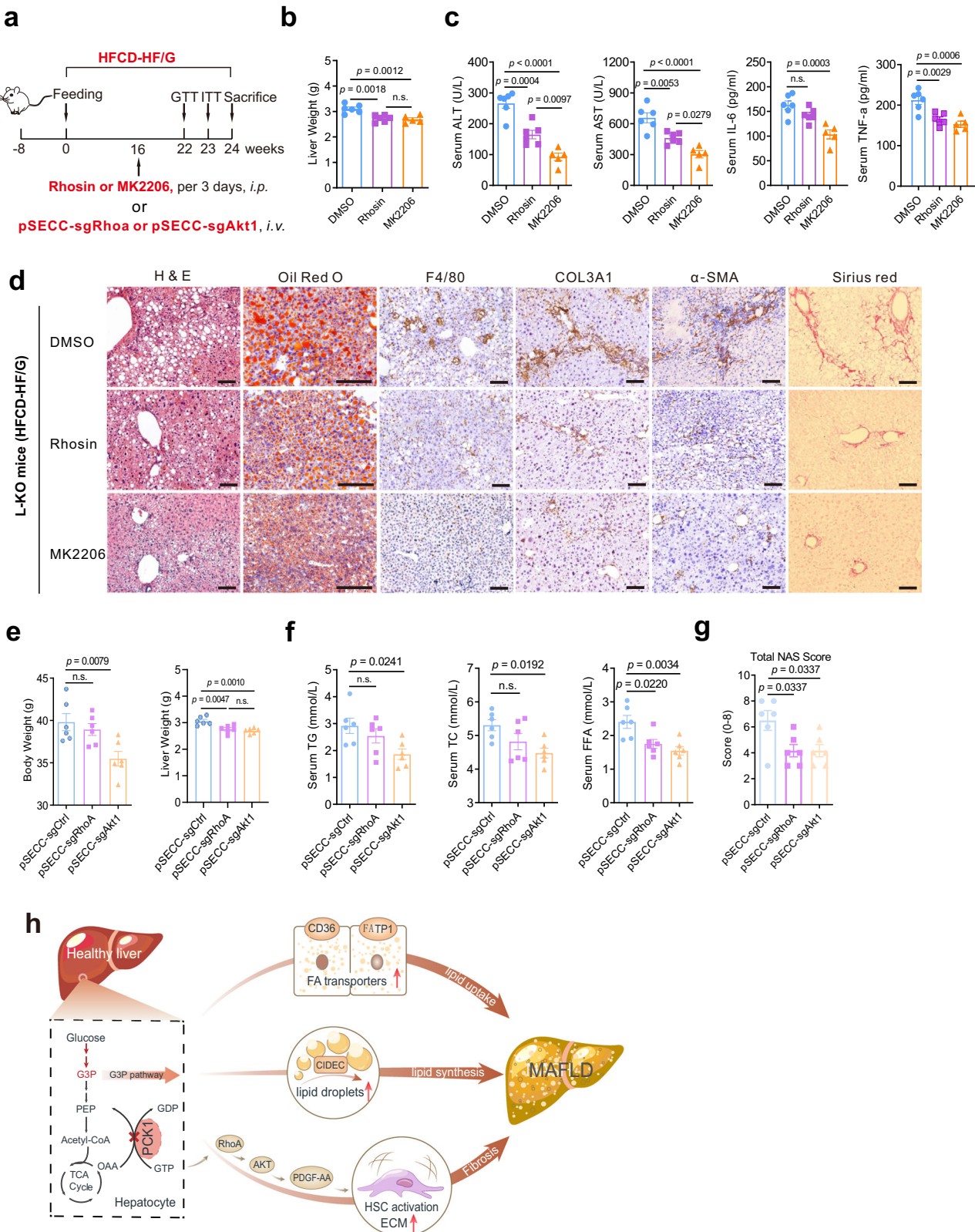

cultured in DMEM containing 10% (vol/vol) fetal bovine serum and 1% penicillin-streptomycin[71].

### Construction of adenovirus, lentivirus, and stable cell lines

AdGFP and AdPCK1 adenoviruses were generated using the AdEasy system as described previously[5]. The *PCK1* knockout (*PCK1*-KO) MIHA cell line was constructed using the CRISPR-Cas9 system (from Prof. Ding Xue, the School of Life Sciences, Tsinghua University, Beijing, China), as described previously[5]. To knock down *ATF3* expression in MIHA cells, three pairs of oligonucleotides encoding short hairpin RNAs (shRNAs) targeting *ATF3* or negative control shRNA were cloned into the pLL3.7 vector (from Prof. Bing Sun, Shanghai Institute of Biochemistry and Cell Biology, Chinese Academy of Sciences, China). The lentiviruses were obtained by transient transfection of the psPAX2

**Fig. 8 | Pharmacological inhibition or genetic silencing of AKT1 or RhoA alleviates MAFLD development in vivo.** L-KO mice were fed the HFCD-HF/G for 24 weeks, and therapeutic treatments were initiated at different times. **a** Schematic diagram of in vivo pharmacological inhibition (**b**–**d**) and pSECC lentivirus-mediated silencing (**e**–**g**) of AKT1 or RhoA. **b, c** Liver weight (**b**), and serum alanine aminotransferase (ALT), aspartate aminotransferase (AST), TNF-α, and IL-6 (**c**) (DMSO, $n = 6$; MK2206, $n = 6$; Rhosin, $n = 5$). **d** Paraffin-embedded liver sections were stained with hematoxylin and eosin, Sirius Red, or immunostained for F4/80, COL3A1, and α-SMA. Frozen sections were stained with Oil Red O. Scale bars: 50 μm.

**e** Quantification of body weight and liver weight in pSECC-sgAkt1 and pSECC-sgRhoA L-KO mice ($n = 6$). **f** Plasma levels of total triglycerides (TG), total cholesterol (TC), and free fatty acids (FFA) ($n = 6$). **g** NAFLD activity scores of liver sections ($n = 6$). **h** Model depicting the critical role of PCK1 in controlling MAFLD progression. n was the number of biologically independent mice. Data are expressed as the mean ± SEM; n.s., not significant.
$p$ values obtained via one-way ANOVA with Tukey's post hoc test. Source data are provided as a Source Data file.

packaging plasmid and pMD2.G envelope plasmid in HEK293T cells by using Lipofectamine 3000 Transfection Reagent (Thermo Fisher Scientific, Waltham, MA, USA). Transfection efficiency was validated by western blotting. Information on the reagents is listed in Supplementary Table 2. The pSECC lentiviral vector cloning and packaging strategy have been described previously[27].

### RNA extraction and real-time PCR
Total RNA was extracted from the liver tissues or cell lines using Trizol reagent (15596018, Invitrogen, Carlsbad CA, USA) according to the manufacturer's instructions. RNA was reverse-transcribed using a PrimeScript RT reagent Kit with gDNA Eraser (Cat: RR047A, TAKARA, Shiga, Japan), and qPCR was performed on a Bio-Rad CFX96 machine (Hercules, CA, USA). The mRNA levels of selected genes were calculated after normalization to β-actin by using the $2^{(-\Delta\Delta C(T))}$ method. Primer sequences are provided in Supplementary Table 3.

### Immunoblotting
Mouse liver tissues and cells were homogenized and lysed in Protein Extraction Reagent (P0013, Beyotime Biotechnology) supplemented with protease inhibitor cocktail (04693159001, Roche, Basel, Switzerland). Equal amounts of protein lysates were separated by sodium dodecyl sulfate-polyacrylamide gel electrophoresis and transferred onto polyvinylidene fluoride membranes (Millipore, Billerica, MA, USA). The antibodies are listed in Supplementary Table 4. β-ACTIN protein was used as a loading control.

### BODIPY staining
The cells were fixed in 4% formaldehyde for 30 min at room temperature, permeabilized with 0.1% Triton X-100 for 5 min at room temperature, and stained with BODYPY (D3823, Invitrogen) for 60 min. Stained sections were analyzed using a Leica confocal microscope (Leica TCS SP8, Wetzlar, Germany).

### CCK8 assay
The cells were seeded at $5 \times 10^3$ cells /well in 96-well plates and treated with different concentrations of PA for 24 h. 10 μL CCK-8 solution (#C0005, Topscience, Shanghai, China) was added to each well and incubated at 37 °C for 1 h. The absorbance was measured at 450 nm.

### ELISA and G-LISA
Serum insulin (P3376, Beyotime Biotechnology), TNFα (PT512, Beyotime Biotechnology), IL-6 (PI326, Beyotime Biotechnology), and PDGF-AA (SEA523Mu, Cloud-Clone Corp., Wuhan, China) concentrations in the liver tissue and plasma were quantified using mouse ELISA kits according to the manufacturer's instructions. The levels of secreted PDGF-AA in the cell culture supernatants were determined using a human ELISA kit (SEA523Hu, Cloud-Clone Corp). Active RhoA was detected in a colorimetric RhoA activation assay (G-LISA) (BK124, Cytoskeleton, Denver, CO, USA).

### Immunofluorescence
LX-2 cells were fixed in 4% formaldehyde for 25 min and then incubated in 10% normal goat serum for 1 h. The cells were incubated with primary α-SMA antibodies. Specific signals were visualized using

secondary antibodies (ZF-0316, Zsbio, Beijing, China). For nuclear staining, the cells were treated with 1 μg/mL DAPI (10236276001, Roche Diagnostics). The samples were detected with a laser-scanning confocal microscope (Leica TCS SP8).

### Chromatin immunoprecipitation (ChIP) assay
Chromatin immunoprecipitation (ChIP) and ChIP-quantitative real-time PCR (ChIP-qPCR) assays for MIHA cells were performed as described previously[72]. Briefly, sonicated chromatin was used for the immunoprecipitation assay. The pre-cleared supernatants were incubated with a monoclonal antibody against ATF3 overnight at 4 °C, followed by a 4 h of incubation with protein A/G agarose beads (LOT: 3460992, Millipore). The purified DNA fragments bound by ATF3 were analyzed using qPCR. IgG and histone H3 were used as negative and positive controls, respectively. The primers used for real-time PCR analysis of PCK1 are listed in Supplementary Table 3.

### Glucose and insulin tolerance tests
The glucose tolerance test (GTT) and insulin tolerance test (ITT) were performed at 2 or 1 weeks prior to sacrifice. The animals were fasted for 16 or 6 h, and then glucose solution (2 g/kg body weight) or insulin (0.75 U/kg body weight) was administered via an intraperitoneal injection, respectively. Venous tail blood samples were collected at 0, 30, 60, 90, and 120 min post-administration to assess blood glucose levels using a glucose meter (HGM-114, OMRON, Kyoto, Japan).

### Luciferase reporter assay
MIHA cells were treated with 0.2 mM palmitic acid (PA) in 12-well plates for 24 h and then co-transfected with 0.5 μg of pReceiver-M02-ATF3 or shControl or shATF3 plasmid, 0.5 μg of luciferase reporter plasmids pGL3-basic or pGL3-PCK1, and 25 ng of pRL-TK-Renilla (as transfection control) for 48 h. Then, cells were assayed for luciferase activity via the Dual Luciferase Assay Kit (Promega, Madison, WI, USA). All experiments were performed thrice and presented as mean ± standard deviation (SD).

### Biochemical analysis
The serum levels of aspartate transaminase (AST), alanine transaminase (ALT), total triglyceride (TG), total cholesterol (TC), and free fatty acids (FFA) were determined with an automated biochemical analyzer (Hitachi 7600, Tokyo, Japan). TG, TC, and FFA levels in the mouse liver tissues were measured with commercial kits according to the manufacturer's protocol (TG: cat. no. BC0625; TC: cat. no. BC1985; FFA: cat. no. BC0595, Solarbio Life Sciences).

### Histological analysis
Paraffin blocks were sectioned into 4 μm slices and used for hematoxylin and eosin (HE) staining, Sirius red staining, and immunohistochemistry (IHC) assay according to standard protocols. Frozen liver tissue sections were stained with Oil Red O (G1260, Solarbio). For pathological grading, all liver specimens were scored by two experienced pathologists according to the NAFLD activity score (NAS), defined as the sum of steatosis (0–3), inflammation (0–3), and hepatocyte ballooning (0–2). An NAS score ≥5 was considered to indicate

NASH. Samples were scanned using a slide scanner (Pannoramic DESK, 3D Histech kft, Hungary). For quantitative analysis, the areas of lipid droplets and Sirius red staining were quantified using ImageJ software (version 1.6.0; NIH, Bethesda, MD, USA). Immunohistochemical staining was semi-quantitatively analyzed using the immunoreactive scoring system[73]. The percentage of positive cells was graded on a scale of 0−4: (0: negative, 1: 0−25%, 2: 26−50%, 3: 51−75%, 4: 76−100%). The signal intensity was scored on a scale of 0−3: 0 = negative; 1 = weak; 2 = moderate; and 3 = strong. Thus, the final immunoreactive score = (score of staining intensity) × (score of percentage of positive cells).

### Transcriptomic analyses
Using Trizol reagent, total RNA was isolated from the liver tissues of WT and L-KO mice fed a chow diet or HFCD-HF/G. The RNA quality was checked with a Bioanalyzer 2200 (Agilent Technologies, Santa Clara, CA, USA). cDNA libraries were prepared using an NEBNext® Ultra™ Directional RNA Library Prep Kit, NEBNext® Poly (A) mRNA Magnetic Isolation Module, NEBNext® Multiplex Oligos according to the manufacturer's instructions (New England Biolabs, Ipswich, MA, USA). Genes with fold-change >2.0 or <0.5 and false discovery rate <0.05 were considered to be significantly differentially expressed. The volcano, heat, and bubble maps were generated using the 'ggplot2' or 'ggpubr' packages in R (version 3.6.3; The R Project for Statistical Computing, Vienna, Austria). Gene set enrichment analysis was performed using 'enrichplot' packages[74]. The RNA-seq data files have been deposited to the Gene Expression Omnibus database (www.ncbi.nlm.nih.gov/geo/) under accession number GSE162211.

### Untargeted metabolomics
Following sacrifice, the livers were snap-frozen in liquid nitrogen and stored at −80 °C until analysis. Untargeted metabolomics was performed using an ultra-high performance liquid chromatography apparatus (Agilent 1290 Infinity LC, Agilent Technologies) coupled to a quadrupole time-of-flight (TripleTOF 6600, AB SCIEX, Framingham, MA, USA) at Shanghai Applied Protein Technology Co., Ltd. (Shanghai, China). Metabolites with a variable importance in projection value >1 were evaluated using Student's $t$-test. $p < 0.05$ was considered to indicate statistically significant results.

### HPLC analysis of cellular nucleotides
Cellular nucleotides were extracted according to published procedures[75]. Briefly, $1 \times 10^6$ cells were washed with phosphate-buffered saline and quenched with liquid nitrogen, and then vigorously mixed with methanol and acetonitrile (1:1, v:v). After incubation on ice for 15 min, the samples were centrifuged at 12,000× $g$ at 4 °C for 10 min. Cellular nucleotides were separated and quantified using a C18 column (Agilent Eclipse XDB-C18, 4.6 × 250 mm, average particle size 5 µm) assembled on the Waters Alliance e2695 Separations Module (Milford, MA, USA). Acetonitrile (5%) and 50 mM $KH_2PO_4$ (pH 6.5) containing 10 mM tetrabutylammonium bromide were used as mobile phase A, and acetonitrile was used as mobile phase B. All samples were separated in the mobile phase at a flow rate of 1 mL/min for 30 min at 22 °C. No degradation of individual nucleotides or changes in the ratios of nucleotide mixtures was detected during the experiments. The GTP concentration in the samples was calculated based on the slope of the calibration curves generated using pooled authentic samples (to mimic the matrix), and guanosine-5′-triphosphoric acid disodium salt was used as a standard.

### GEO database mining
Raw data in the GSE126848, GSE89632, and GSE135251 datasets were downloaded from the GEO database (https://www.ncbi.nlm.nih.gov/geo/). R package DESeq2 was used to analyze gene expression levels between different samples, such as NASH vs. health, NAFLD vs. health, and obesity vs. health. Genes showing |log₂ fold-change | >1 and false

discovery rate <0.05 were considered to present differential expression.

### Statistics
Statistical analyses were performed using GraphPad Prism 6.0 software (GraphPad, Inc., La Jolla, CA, USA). Data were represented as the mean ± standard error of the mean unless otherwise stated. Significant differences between the means of two groups were determined using Student's $t$-test. One-way analysis of variance was used to determine statistical significance for experiments with more than two groups followed by Tukey's post hoc test. For correlation analysis, Spearman's correlation coefficient was used. $p < 0.05$ was considered to indicate statistically significant results.

### Reporting summary
Further information on research design is available in the Nature Portfolio Reporting Summary linked to this article.

## Data availability
The RNAseq data produced in this study were deposited to the public database [Accession GSE162211]. The previously published data sets re-analysed in this study were obtained from [Gene Expression Omnibus (GEO)], through the accession code [Accession GSE126848], [Accession GSE89632], and [Accession GSE135251][16,76,77]. The untargeted metabolomic data reported in this article have been deposited in the OMIX, China National Center for Bioinformation / Beijing Institute of Genomics, Chinese Academy of Sciences (https://ngdc.cncb.ac.cn/omix: accession no. OMIX003062). All other data generated or analyzed in this study are available within the article and its supplementary information files. Source data are provided with this paper.

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

## Acknowledgements

We would like to thank Dr. T.-C He (University of Chicago, USA) and Prof. Ding Xue (Tsinghua University, China) for providing the pAdEasy and CRISPR/Cas9 system, respectively. We thank Prof. Youde Cao and Yalan Wang (Chongqing Medical University, China) for providing samples and pathological analysis support. We thank Prof. Yongjun Dang, Hui Zhou and Luyi Huang (Chongqing Medical University, China) for providing the strong technical support for measuring intracellular GTP levels. This work was supported by the National Natural Science Foundation of China (grant no. U20A20392, 82073251, 82273238, 82072286, 82272975), the 111 Project (No. D20028), the Natural Science Foundation Project of Chongqing (cstc2018jcyjAX0254, cstc2019jscx-dxwtBX0019), the Science and Technology Research Program of Chongqing Municipal Education Commission (HZ2021006, KJZD-M202000401), the Future Medical Youth Innovation Team of Chongqing Medical University (W0036, W0101), the Kuanren talents program of the second affiliated hospital of Chongqing Medical University, the Scientific Research Innovation Project for Postgraduate in Chongqing (CYB19168, CYS19193), and the Youth talent fund of Sichuan Provincial People's Hospital in 2020 (No. 2020QN03).

## Author contributions

N.T., A.H., and K.Wang conceived and designed the study. Q.Y., Y.L., G.Z. and L.T. performed most experiments and analyzed the data. H.D., C.C. and K.Wu conducted bioinformatics analysis. X.P. assisted with mice experiments. X.W., J.F., and Q.P. provided human NAFLD/NASH samples. Q.Y., K.Wang, and N.T. wrote the manuscript with all authors providing feedback. The order of the co-first authors was assigned on the basis of their relative contributions to the study.

## Competing interests

The authors declare no competing interests.

### Ethical approval

Animal experiments were approved by Animal Experimentation Ethics Committees of Chongqing Medical University and were carried out in accordance with the Guide for the Care and Use of Laboratory Animals. Liver tissue collection was approved by the Institutional Ethics Committees of Chongqing Medical University and Xin Hua Hospital. Informed consent was obtained from all participants.
