## [Peer Review File · Nature Communications]

Deficiency of gluconeogenic enzyme PCK1 promotes metabolic- associated fatty liver disease (MAFLD) through PI3K/AKT/PDGF axis activationREVIEWER COMMENTS

Reviewer #1 (Remarks to the Author):

In this manuscript, authors showed that PCK1 deficiency exacerbates NASH features including steatosis, inflammation and fibrosis. Their mechanistic investigations suggest that activated RhoA/PI3K/AKT pathway contributes to the role of PCK1 in fibrosis. While the work is some interest with extensive phenotypic studies, the molecular mechanism is preliminary. There are several important points to be addressed.

1. PCK1 has been well-established as the first limit-rating enzyme of gluconeogenesis, a physiology very closely correlated with NASH progression. How about the influence of PCK1 deficiency on gluconeogenesis in mice treated with chow diet and NASH diet? Further, PCK1 is a key factor linking TCA metabolism and gluconeogenesis. Whether TCA cycle is disturbed when PCK1 depleted?
2. It's interesting that the hepatocyte-specific PCK1 deficiency led to more body weight gain under both chow and NASH diet. Authors also showed that the reduced PCK1 expression is observed in the liver of patients with obesity. Why and how Pck1 hepatocyte-specific depletion influences whole body metabolism should be addressed here.
3. Regarding the mechanisms, authors identified PI3K-AKT signaling underlying the function of PCK1 on HSC activation and fibrosis. What's the direct downstream effector of PCK1? Is there a direct interaction between PCK1 and PI3K? Whether the influence on PI3K-AKT signaling is dependent on the enzymatic activity of PCK1? Intracellular GTP and GDP levels should be examined and the general effect of altered GTP level should be put into more depth discussion.
4. In fig 1d and 1e, PCK1 protein levels seems markedly reduce by more than twice with a very mild mRNA level downregulation. Are there other possible pathways leading to PCK1 expression reduction?
5. Authors showed that ATF3 negatively regulated PCK1 expression. It's interesting to know whether the expression of ATF3 has a significant correlation with PCK1 in clinic samples. Correlation analysis based on public transcriptome dataset would help to clarify this issue.
6. In fig 7, the conclusion that Rhosin inhibited LX2 cells activation depends on MIHA cells is challenged. There might be a direct action of Rhosin on LX2 cells.
7. Based on authors' observations, the activation of HSC cells and NASH-related fibrosis are more likely mediated by paracrine PDGF-AA, instead of owing to PI3K-AKT pathway. The conclusion of this study should be appropriately revised.

Reviewer #2 (Remarks to the Author):

This manuscript describes a potential role for PCK1 in modulating NASH progression and fibrosis through activating the PI3K/AKT/PDGF pathway. By using hepatocyte-specific PCK1 knockout mice, the authors propose that hepatocyte PCK1 has an important role in decreasing the NASH hepatic phenotype in a NASH model. The authors propose that inhibiting the pathways downstream of PCK1, by using AKT and RhoA inhibitors, could have potential therapeutic value. This is an interesting study and could be

published in Nature Communications after attention is paid to several issues. Notably, rescue experiments and treatment of mice with fully developed NASH with the RhoA and MK2206 to determine if blocking RhoA/PI3K/AKT is of translational value in treating NASH.

Major issues:

1. The authors propose that hepatocyte PCK1 deficiency enhances NASH and fibrosis by affecting the PI3/AKT pathway. However, the authors just described separately the role of PCK1 and PI3/AKT pathway in modulating NASH. The contribution of PI3/AKT pathway to the phenotype of PCK1 deficiency needs to be examined using rescue experiments *in vivo*.
2. Line 124: "Interestingly, PCK1 mRNA and protein levels were downregulated in a dose-dependent manner during 24-hour PA stimulation (Fig. 1f, g), suggesting that transcription of PCK1 may be inhibited in response to lipid overload." This suggests that PCK1 is decreased as a RESULT of increased lipid. What is the mechanism by which PCK1 is decreased in NASH?
3. In Fig. 1F, the mRNA level decreased from 0.2 mM PA, but the cell viability did not decrease from 0.2 mM PA (Fig. S1D). From these results, we can consider that the cytotoxicity of PA at low concentrations is not related to the decreased expression of PCK1. The possibility of decreased expression of PCK1 independent of lipid overload should be discussed.
4. The authors also tried to correlate their mouse findings with human samples as well as some published datasets using bioinformatics. However, the sample size of Figure 1c and Figure 7k needs to be described. Quantitation of at least N=3-5 are needed to show the expression change of PCK1. It is suggested to describe the data in Figure 1c-e separately, instead of "PCR, immunoblotting, and immunohistochemistry (IHC) assays showing that PCK1 expression were dramatically downregulated in liver samples derived from NASH patients and 119 NASH model mice (Fig. 1c-e)."
5. The background color of Figure 1c for Healthy liver and NASH liver looks different. Also, the sample size needs to be shown in the figure legend for the mouse NASH model data. Whether PCK1 expression is changed at the early stage of NAFLD and whether hepatocyte-specific PCK1 expression was changed needs to be defined using the appropriate mouse NASH model.
6. Validation of hepatocyte-specific PCK1 knockout efficacy needs to be performed. The authors showed that under NASH diet, PCK1 is knocked out in the liver in Figure 4e. However, whether PCK1 is specifically deleted in the liver, but not in the other tissues, needs to be determined.
7. The authors need to determine whether hepatic fibrogenesis of PCK1-LKO mice change under chow diet?
8. How PCK1 modulates the RhoA/PI3K/AKT *in vitro* and at the early stage of NASH *in vivo* was not examined. The possibility that the PCK1 knockout affected this pathway as a result of enhanced NASH could not be excluded.
9. The authors used MIHA and HSC (LX-2) co-culture assays. LX-2 is a human HSC cell line, while MIHA is also human hepatocyte line. Primary mouse hepatocytes and primary mouse HSCs should be used. The authors need to make it clear when LX-2 cells are used and when primary HSCs were used and they need to specifically designate each of these two types instead of just abbreviating both as HSCs.
10. In Figure 5G: As expression of collagen is increased, the expression of MMPs should also change, but the expression of MMP2 was not affected. Why is the expression of MMPs (MMP1, MMP2 and so on) not affected? The authors should discuss this point in more detail.

11. In Figure S4, LAMA2 is not the major laminin isoform in the liver; LAMA5 is the major form. LAMA2 levels should be measured?
12. The authors used RhoA and AKT inhibitors with NASH treatment. However, whether these compounds showed an anti-NASH effect due to an on-target effect or off-target effect was not demonstrated. More gene-specific pathways with knockout mice or AAV8-delivered hepatocyte-specific gene knockout mouse models need to be used.
13. The Methods section showed that N=10 normal and N=36 NASH human liver samples were used. However, the qPCR data were not found. Only one representative IHC figure was shown with no statistical analyses in Figure 1C and Figure 7.
14. In isolation of primary HSCs, in line 493, it was stated as "HSCs were plated in tissue culture dishes in DMEM." Detailed information of the media need to be provided - 10% FBS or other supplements? Could primary HSCs attach to the tissue culture dishes? Usually coated plates are needed for seeding primary HSCs.
15. As noted above, whether the proposed pathway is a result or a causal factor for the phenotype was not established. Are the proposed pathways changed during the early stage of NASH diet feeding compared to chow diet? Could PCK1 knockout still promote NASH when the pathway is blocked? Does hepatocyte-specific PCK1 knockout affect the PI3K/AKT pathway in vitro?
16. In Fig. 2 and Fig. 3, the qPCR of inflammation markers and fibrosis markers need to be shown.
17. The levels of serum PDGF-AA in the NASH model in WT and LKO mice need to be determined?
18. Hepatocyte PCK1 downregulation is correlated with enhanced NASH progression. Does forced expression of PCK1 in liver reverse the chow-fed and NASH diet-induced phenotype of the PCK1 L-KO mice?
19. Wild-type mice on the NASH diet with a fully developed NASH liver phenotype should be treated with Rhosin and MK2206 to determine if blocking RhoA/PI3K/AKT is of translational value in treating NASH.
20. Figure 8: AKT inhibitor has been shown to improve the pathophysiology of NASH. However, it is known that inhibition of AKT not only inhibits fibrosis via HSC activation, but also affects TGF-beta activation. This means that the model is not suitable for focusing only on the inhibition of HSC activation. Therefore, if the authors are going to evaluate AKT inhibitors, they should simultaneously discuss the effect of changes in other factors, such as TGF-beta, on NASH as well as HSC activation.

Minor points:

1. What is the background of the mouse lines used in this study?
2. Line 63: "NAFLD may progress to NASH, a more serious form.." may not be exactly correct. NAFLD includes simple fatty liver and NASH. It could be "nonalcoholic fatty liver (NAFL) could progress to nonalcoholic steatohepatitis (NASH)."
3. In Figure. 1H, BSA is in the culture medium and is also present in the PA-treated group, so the author should change the wording in the figure panel.

Reviewer #3 (Remarks to the Author):

This is a well performed study which examines the role of Pck1 in the NASH. It builds on earlier work from 2010 which shows that low PCK1 levels result in significant dysregulation of lipid metabolism and hepatic lipid content. This manuscript has expanded on this by using hepatocyte specific PCK1 KO mice, and demonstrating activation of RhoA/P13K/AKT pathway which contributes to PDGF-AA secretion by hepatocytes and activation of HSC. Overall the work is incremental, as the general effect of PCK1 was known, and also because other studies have shown that disruption of gluconeogenesis leads to severe hepatic steatosis. The connection with HSC activation is somewhat peripheral because we know that all forms of hepatocyte injury results in HSC activation. Minor point when L-KO mice are introduced (page 8 of PDF) please explain what they are. The reader must go into the methods to find this information.

Responses to reviewers

General issues

We appreciate the reviewer's insightful and critical comments, as well as the positive comments on our study. In response to these comments, we performed additional experiments and revised the manuscript accordingly, which clarified important aspects of studied pathways, substantially strengthened our initial conclusions, and improved our work overall. Below, please find our detailed responses to specific comments, as well as relevant changes made to the figures and manuscript text.

Reviewer #1 (Remarks to the Author):

In this manuscript, authors showed that PCK1 deficiency exacerbates NASH features including steatosis, inflammation and fibrosis. Their mechanistic investigations suggest that activated RhoA/PI3K/AKT pathway contributes to the role of PCK1 in fibrosis. While the work is some interest with extensive phenotypic studies, the molecular mechanism is preliminary. There are several important points to be addressed.

Author response: Thank you for recognizing the importance of our study and providing numerous constructive suggestions. We have conducted additional experiments and analyses to better support our main conclusions. We have also revised the manuscript according to specific comments, as described below.

1. PCK1 has been well-established as the first limit-rating enzyme of gluconeogenesis, a physiology very closely correlated with NASH progression. How about the influence of PCK1 deficiency on gluconeogenesis in mice treated with chow diet and NASH diet? Further, PCK1 is a key factor linking TCA metabolism and gluconeogenesis. Whether TCA cycle is disturbed when PCK1 depleted?

Author response: Thank you for your comments.

1) We found that when WT and L-KO mice fed a chow diet were starved for 16 h for the glucose tolerance test (GTT), the blood glucose level at 0 min was significantly lower in L-KO mice than that in WT mice, suggesting impaired gluconeogenesis for L-KO mice fed the chow diet (**Figure 2c, middle panel**). Similarly, we found that blood glucose levels at 0 min in the GTT test were significantly lower in L-KO mice fed the NASH diet than those in WT mice after 16 h of starvation (**Supplementary Figure 2a**). In addition, western blot analysis revealed that key gluconeogenetic enzymes, such as G6PC and FBP1, slightly decreased in L-KO mice (**Rebuttal Figure 1a**). These results suggest that L-KO mice fed the NASH diet also suffered from impaired gluconeogenesis.

2) To investigate the changes in TCA metabolism after PCK1 depletion, the liver tissues of WT and PCK1-KO mice fed the chow diet or NASH diet were examined using targeted LC-MS/MS. Target metabolite analysis showed that compared with WT mice, several key intermediary metabolites in the TCA cycle, including fumarate, succinate, and citrate, increased in the liver of L-KO mice. These TCA cycle intermediates were more obviously increased in mice fed the NASH diet than in those fed a normal diet (**Rebuttal Figure 1b-c**). These results indicate that PCK1 depletion disturbed the TCA cycle.

In addition to its well-known gluconeogenic role, PCK1 plays an important role in regulating TCA cycle flux¹. Consistent with our results, a previous study reported that the liver-specific deletion of PCK1 impaired hepatic gluconeogenesis from lactate and pyruvate and limited TCA cycle function, resulting in the accumulation of lipids and cytosolic malate². Multiple previous studies have demonstrated that the concentrations of hepatic TCA cycle intermediates are dramatically high in PCK1-null livers^{3,4}. Moreover, gluconeogenesis was nearly absent from PCK1-deficient livers during a short fasting period (4 h), and 2-fold lower glucose production and oxygen consumption were observed in PCK1-deficient livers when fasting was extended to 24 h.

The clinical relevance of PCK1-regulated TCA metabolism and gluconeogenesis was recently clarified. PCK1 deficiency due to the homozygous PCK1 variant is associated with childhood-onset hypoglycemia with a recognizable pattern of abnormal urine organic acids. The patients showed abnormal urine organic acid profiles with increased TCA cycle intermediates and inadequate ketone body production⁵. Also, homozygous 12 bp deletion in PCK1 led to the accumulation of TCA cycle metabolites in urine organic acid analysis⁶. Thus, numerous studies have shown that PCK1 plays an important role in regulating TCA cycle flux and gluconeogenesis. We have revised the text and included new data to support the conclusion of our work (page 22-23, lines 425–431).

Rebuttal Figure 1. Metabolites detected using targeted LC-MS/MS. **a** Protein levels of PCK1, FBP1, and G6PC in the livers of WT and L-KO mice fed the NASH diet for 24 weeks. **b** and **c** Concentrations of key intermediate metabolites in the TCA cycle in the liver tissues of mice (a, chow diet; b, NASH diet).

2. It's interesting that the hepatocyte-specific PCK1 deficiency led to more body weight gain under both chow and NASH diet. Authors also showed that the reduced PCK1 expression is observed in the liver of patients with obesity. Why and how Pck1 hepatocyte-specific depletion influences whole body metabolism should be addressed here.

Author response: Thank you for raising this interesting point. To respond to various metabolic demands, metabolic events occurring in one tissue greatly affect metabolism in other distant tissues; however, inter-organ communication is quite complex and not well-understood^{7,8}. Below, we explain why and how *Pck1* hepatocyte-specific depletion influences whole-body metabolism.

First, there is a close mutual interdependence between different metabolic pathways; therefore, these phenotypes may result not only from gluconeogenesis but also from TCA cataplerosis, another metabolic process regulated by PCK1 that maintains metabolic flux through the Krebs cycle by removing excess oxaloacetate^{9,10}. Previous studies demonstrated that the loss

of PCK1 in the liver disturbed hepatic cataplerosis and led to the accumulation of TCA cycle intermediates. The slowed TCA cycle impaired fatty acid oxidation, resulting in fat accumulation in the liver³.

Second, we demonstrated that liver-specific PCK1 deficiency disturbed TCA metabolism and gluconeogenesis. Additionally, the depletion of PCK1 increased the expression of fatty acid transporters, such as *Cd36* and *Slc27a1*, and upregulated the G3P pathway, thus leading to intracellular lipid accumulation. Excessive fatty acids are secreted by the liver in very-low density lipoprotein particles and delivered to peripheral tissues such as white adipose tissue¹¹. Free fatty acids are converted to triglyceride in adipocytes¹². In fact, we also observed increased abdominal visceral fat in L-KO mice (**Rebuttal Figure 2**). Although we did not measure the weight or volume of the adipose tissues, we observed elevated plasma triglycerides and free fatty acids in L-KO mice (**Figure 2f, Supplementary Figure 2d**). Therefore, elevated plasma triglycerides and abdominal fat accumulation may have contributed to body weight gain in L-KO mice.

Our results also indicated that L-KO mice developed glucose intolerance and insulin resistance when fed the NASH diet (**Supplementary Figure 2a–b**), playing a key role in obesity and whole-body metabolism¹³. In the current study, the loss of PCK1 upregulated PDGF-AA expression (**Figure 6a–h**). PDGF-AA has been reported to be associated with an increased risk of type 2 diabetes, hyperinsulinemia, insulin resistance, and steatohepatitis¹⁴. Mechanistically, PDGF-AA has been shown to promote insulin resistance by decreasing INSR and IRS1 and by activating PKC θ and PKC ϵ ¹⁴. In addition, PCK1 deficiency may affect insulin resistance by upregulating CD36 (**Figure 4d, e**), as hepatic CD36 is significantly associated with increased insulin resistance, hyperinsulinemia, and steatosis in patients with NASH¹⁵. Interestingly, cataplerosis by mitochondrial PEPCK (PCK2) mediates insulin secretion by altering mitochondrial GTP as a signaling factor¹⁶.

In addition to the original function of PCK1, whole-genome scanning has revealed that PCK1 is located on human chromosome 20q, and genes in this region are mostly associated with type 2 diabetes, suggesting that PCK1 affects whole-body metabolism by influencing the co-expression of its neighboring gene^{17,18}.

Thus, *Pck1* hepatocyte-specific depletion may influence whole-body metabolism through multiple mechanisms, which should be further investigated. We hope the reviewer finds our responses satisfactory. We have revised the Discussion to include this point (page 22-23, lines 425–431).

Mice (NASH Diet)

Rebuttal Figure 2. L-KO mice fed the NASH diet had increased abdominal fat.

3. Regarding the mechanisms, authors identified PI3K-AKT signaling underlying the function of PCK1 on HSC activation and fibrosis. What's the direct downstream effector of PCK1? Is there a direct interaction between PCK1 and PI3K? Whether the influence on PI3K-AKT signaling is dependent on the enzymatic activity of PCK1? Intracellular GTP and GDP levels should be examined and the general effect of altered GTP level should be put into more depth discussion.

Author response: We appreciate these helpful suggestions and have performed the recommended analyses.

1) To determine whether PCK1 directly interacts PI3K, we conducted immunoprecipitation coupled with tandem mass spectrometry to screen for proteins that may interact with PCK1 (**Rebuttal Table 1**). However, we did not detect a direct physical interaction between PCK1 and proteins in PI3K pathways. In addition, proteins co-precipitated with PCK1 and identified by mass spectrometry were reported in another study, but these did not include proteins associated with the PI3K pathway, suggesting that PI3K does not interact with PCK1 directly¹⁹. Furthermore, co-immunoprecipitation experiments showed that PCK1 did not interact with P85, a regulatory subunit of PI3K (**Rebuttal Figure 3**). Therefore, the activation of the PI3K/AKT signaling pathway caused by PCK1 depletion may be mediated by other mechanisms. Rho GTPases, which cycle between active GTP-bound and inactive GDP-bound conformations, are known to activate the PI3K/AKT pathway²⁰⁻²². Considering that PCK1 catalyzes the conversion of oxaloacetate and GTP to phosphoenolpyruvate, GDP, and CO₂, we predicted that PCK1 deficiency alters intracellular GTP homeostasis and ultimately results in the

activation of the PI3K/AKT pathway. To verify this, we conducted HPLC analysis to determine intracellular GTP levels in MIHA cells. The results showed that intracellular GTP levels increased in PCK1-knockout cells but decreased in PCK1-overexpressing cells (**Supplementary Figure 5a, b**). Most Rho GTPases switch between an active GTP-bound form and an inactive GDP-bound form, and increased intracellular GTP levels activate Rho GTPases²³, whereas decreased intracellular GTP levels have the opposite effect²⁴. Therefore, we performed initial screening using western blotting (**Figure 7a**), and found that the loss of PCK1 led to the activation of RhoA (**Figure 7b–e**). These data suggest that PCK1 depletion activates RhoA by increasing intracellular GTP levels, further activating PI3K/AKT signaling. We have revised the Discussion to include this point (page 24-25, lines 461–470). **Rebuttal Table 1. Proteins co-precipitated with Flag-PCK1 identified using mass spectrometry.**

Reference	PepCount	Unique-PepCount	CoverPercent	MW	Accession
KIF11	9	9	6.82%	119,157.62	P52732
KV224	10	2	10.00%	130,78.87	A0A0C4DH68
KV229	9	2	16.67%	13,084.9	A2NJV5
scFv	7	2	4.17%	25,569.19	Q65ZC9
RS18	2	2	11.18%	17,718.45	P62269
TBA1C	2	2	3.79%	49,894.78	Q9BQE3
ACTB	2	2	4.00%	41,736.29	P60709
TRAJ56	10	1	38.10%	2110.41	A0A075B6Z2
DC2L1	7	1	1.42%	39,624.16	Q8TCX1
LORF2	5	1	0.39%	149,009.84	O00370
SH321	4	1	0.78%	70,518.32	A4FU49
CSF2RB	4	1	7.41%	11,645.44	L0R5A1
TRPA1	3	1	0.54%	127,499.59	O75762
SORTILIN	3	1	30.00%	2414.65	A0A0J9YY30
IL16	2	1	0.60%	141,750.72	Q14005
PTHD3	2	1	1.30%	86,870.71	Q3KNS1
PLSL	2	1	1.44%	70,287.59	P13796
GANP	1	1	0.35%	218,402.15	O60318
CAND2	1	1	0.49%	135,254.88	O75155
TXD12	1	1	4.07%	19,205.55	O95881
RO52	1	1	1.47%	54,169.15	P19474
SMBP2	1	1	0.60%	109,147.8	P38935
H4	1	1	7.77%	11,367.2	P62805
GOGA3	1	1	0.47%	167,352.73	Q08378
EF1A3	1	1	1.52%	50,184.44	Q5VTE0
H3C	1	1	6.67%	15,213.56	Q6NXT2
HORN	1	1	0.25%	282,386.64	Q86YZ3

Z280C	1	1	1.09%	83,095.12	Q8ND82
TP4AP	1	1	0.63%	90851.04	Q8TEL6
ELMO1	1	1	1.65%	83828.54	Q92556
P63	1	1	0.88%	76784.64	Q9H3D4
KCTD5	1	1	2.99%	26092.26	Q9NXV2
MYOF	1	1	0.44%	234705.9	Q9NZM1
ZN229	1	1	1.33%	93,706.39	Q9UJW7
ERH	1	1	5.77%	12,258.81	P84090
U17LD	1	1	1.32%	59,693.1	C9JLJ4
SRSF3	1	1	4.27%	19,329.38	P84103
RIMS2	1	1	0.35%	160,401.07	Q9UQ26
CQ058	1	1	8.25%	11,219	Q2M2W7
CCNF	1	1	0.64%	87,638.85	P41002
CXA10	1	1	1.10%	61,871.1	Q969M2
LYSC	1	1	6.08%	16,536.86	P61626
TELO2	1	1	0.84%	91,745.96	Q9Y4R8
HSP7C	1	1	1.08%	70,897.24	P11142
SHRM3	1	1	0.45%	216,854.92	Q8TF72
HS902	1	1	3.50%	39,364.33	Q14568
RL6	1	1	3.13%	32,727.51	Q02878
ANM5	1	1	0.94%	72,683.02	O14744
MTOR	1	1	0.59%	288,888.4	P42345
GPR15	1	1	1.39%	40,786.68	P49685
CDK5RAP2	1	1	0.33%	206,317.4	B7ZLE9
INSULIN	1	1	6.36%	11,980.79	P01308
1433B	1	1	2.85%	28,082.08	P31946
IF2B1	1	1	1.39%	63,479.84	Q9NZI8
LYRM7	1	1	5.77%	11,954.74	Q5U5X0
MARH7	1	1	0.85%	78,050.14	Q9H992
RLA0	1	1	2.21%	34,273.11	P05388
THIO	1	1	8.57%	11,737.38	P10599

Rebuttal Figure 3. Interactions between PCK1 and PI3K-P85 in MIHA cells

were determined by co-immunoprecipitation.

2) To determine whether the activation of PI3K-AKT signaling depends on the enzyme activity of PCK1, recombinant adenoviral G309R (PCK1 mutation 925G>A), an enzymatically deficient mutant of PCK1²⁵, was constructed using the AdEasy system as described previously²⁶. Interestingly, the catalytically inactive G309R mutant of PCK1 did not reduce p-AKT(S473) and p-AKT(T308) levels in MIHA cells under the same culture conditions (**Rebuttal Figure 4**), suggesting that the enzyme activity of PCK1 plays an essential role in regulating the PI3K/AKT signaling pathway.

Rebuttal Figure 4. Representative western blot analysis of the indicated proteins in MIHA cells infected with adenovirus overexpressing GFP (AdGFP), AdPCK1, or AdG309R (an enzymatically deficient mutant of PCK1) with or without 0.2 mM PA treatment.

3) In summary, several lines of evidence suggest that PCK1 deletion increased intracellular GTP levels, thus promoting the activation of RhoA and further activating the PI3K/AKT pathway. New data related to intracellular GTP levels have been included in **Supplementary Figure 5a–b** and described in the revised manuscript (page 16, lines 294–299).

4. In fig 1d and 1e, PCK1 protein levels seems markedly reduce by more than twice with a very mild mRNA level downregulation. Are there other possible pathways leading to PCK1 expression reduction?

Author response: Thank you for pointing this out. According to previous reports, PCK1 expression is regulated not only at the transcriptional and post-transcriptional levels but also at the post-translational level. Acetylation, ubiquitination, and phosphorylation modulate the biological functions of PCK1^{19,27,28}. In this study, mice were fed a high-fat diet with drinking water containing a high concentration of glucose; high glucose levels reportedly destabilize PCK1 by stimulating its acetylation, which promotes its ubiquitinylation and subsequent degradation¹⁹. Moreover, the GSK3β-induced phosphorylation of PCK1 promotes its self-ubiquitination and degradation

under high-glucose conditions²⁸. These post-transcriptional modifications may be the underlying mechanisms by which PCK1 protein levels decreased more than the mRNA levels.

5. Authors showed that ATF3 negatively regulated PCK1 expression. It's interesting to know whether the expression of ATF3 has a significant correlation with PCK1 in clinic samples. Correlation analysis based on public transcriptome dataset would help to clarify this issue.

Author response: We appreciate this constructive suggestion. We analyzed the relationship between *PCK1* and *ATF3* using a public transcriptome dataset. Correlation analysis revealed a significant negative correlation between *ATF3* and *PCK1* expression in clinical specimens (GEO: GSE135251). This new analysis of human GEO data strongly supports the correlation between ATF3 and PCK1 expression at the clinical level. This result has been included in **Figure 1m (Rebuttal Figure 5)** and described in the Results section (page 8, lines 125–127).

Rebuttal Figure 5. Correlation analysis of *ATF3* mRNA level with *PCK1* based on public transcriptome dataset (GSE135251) (n = 206).

6. In fig 7, the conclusion that Rhosin inhibited LX2 cells activation depends on MIHA cells is challenged. There might be a direct action of Rhosin on LX2 cells.

Author response: Thank you for this comment; we apologize for our unclear description in the Methods section. LX-2 was pre-cultured in the lower chamber, and MIHA cells were seeded into the transwell inserts. Rhosin treatment of MIHA cells was conducted only before co-culture, as the Rhosin-containing medium was removed and MIHA cells were washed before loading into the LX-2-containing wells. Thus, there was no direct effect of Rhosin on LX-2 in our *in vitro* system. To confirm the role of RhoA in MIHA, two lentivirus-carrying RhoA short hairpin RNAs were used to knock down RhoA expression in MIHA cells. The results showed that the decreased activation of hepatic stellate cells (LX-2) co-cultured with PCK1-knockout hepatocytes was indeed caused by RhoA silencing. These data are described on page 16-17,

lines 312–317. The corresponding images have been included in **Supplementary Figure 5d-f (Rebuttal Figure 6)** in the revised manuscript).

Rebuttal Figure 6. RhoA knockdown reversed the activation of the PI3K/AKT/PDGF-AA axis. **a** PDGF-AA levels in the supernatant of PCK1-KO MIHA cells infected with either control lentivirus (shControl) or shRhoA lentivirus treated with 0.2 mM PA. **b** Relative mRNA expression of *ACTA2*, *COL1A1*, and *COL3A1* in LX-2 cells co-cultured with PCK1-KO MIHA cells infected with either shControl or shRhoA. **c** Immunoblot analysis of indicated proteins in PCK1-KO MIHA cells infected with either shControl or shRhoA.

7. Based on authors' observations, the activation of HSC cells and NASH-related fibrosis are more likely mediated by paracrine PDGF-AA, instead of owing to PI3K-AKT pathway. The conclusion of this study should be appropriately revised.

Author response: Thank you for these helpful suggestions. We agree that the activation of HSC cells and NASH-related fibrosis is likely mediated by paracrine PDGF-AA; we have revised this in the Conclusions of this study (page 3, lines 43–44; page 20, line 368).

Reviewer #2 (Remarks to the Author):

This manuscript describes a potential role for PCK1 in modulating NASH progression and fibrosis through activating the PI3K/AKT/PDGF pathway. By using hepatocyte-specific PCK1 knockout mice, the authors propose that hepatocyte PCK1 has an important role in decreasing the NASH hepatic phenotype in a NASH model. The authors propose that inhibiting the pathways downstream of PCK1, by using AKT and RhoA inhibitors, could have potential therapeutic value. This is an interesting study and could be published in Nature Communications after attention is paid to several issues. Notably, rescue experiments and treatment of mice with fully developed NASH with the Rhosin and MK2206 to determine if blocking RhoA/PI3K/AKT is of translational value in treating NASH.

Author response: We appreciate the reviewer's positive comments on our work and hope that the extensive new data and improved presentation of the revised manuscript clarify our message and the conceptual advances of our results.

Major issues:

1. The authors propose that hepatocyte PCK1 deficiency enhances NASH and fibrosis by affecting the PI3/AKT pathway. However, the authors just described separately the role of PCK1 and PI3/AKT pathway in modulating NASH. The contribution of PI3/AKT pathway to the phenotype of PCK1 deficiency needs to be examined using rescue experiments *in vivo*.

Author response: We appreciate the reviewer's constructive suggestion. In the original manuscript, we described that pharmacological inhibition of AKT1 or RhoA reversed NASH in L-KO mice fed the NASH diet (See **Figure 8** and **Supplementary Figure 5** in the original manuscript). In the revised manuscript, we infected L-KO mice with a pSECC lentivirus-based system that combines both the CRISPR system and Cre recombinase²⁹, which expressed validated sgRhoA or sgAKT1 to silence RhoA or AKT1 in L-KO mice, respectively. RhoA and AKT1 were silenced in the liver tissues, and the silencing of RhoA or AKT1 reversed NASH and fibrosis progression induced by the NASH diet in L-KO mice (**Figure 8f–i** and **Supplementary Figure 7**).

In addition, as the reviewer suggested, we used adeno-associated virus 8 (AAV8)-TBG viral vectors to restore PCK1 expression in hepatocytes *in vivo*, which indicated that the forced expression of PCK1 in the liver of L-KO mice reduced NASH and liver fibrosis induced by the chow and NASH diets (**Figure 3h–l** and **Supplementary Figure 2h–k**). These rescue experiments confirm our original conclusions. We have reorganized and edited all sections of the main text to incorporate the new data supporting our conclusions.

2. Line 124: “Interestingly, PCK1 mRNA and protein levels were downregulated in a dose-dependent manner during 24-hour PA stimulation (Fig. 1f, g), suggesting that transcription of PCK1 may be inhibited in response to lipid overload.” This suggest that PCK1 is decreased as a RESULT of increased lipid. What is the mechanism by which PCK1 is decreased in NASH?

Author response: Thank you for this comment. We agree that this is an interesting point. PCK1 expression is regulated at the transcriptional, post-transcriptional, and post-translational levels. Transcriptional regulation is considered to be the major modality of PCK1 regulation. In normal gluconeogenic tissues, glucagon and glucocorticoids activate *PCK1* transcription, whereas insulin inhibits PCK1 expression. Obesity, type 2 diabetes, and NASH are strongly associated with insulin resistance, and elevated insulin levels can lead to decreased PCK1 expression¹⁰. Moreover, under high-glucose conditions, insulin inhibits Forkhead box protein O1 and PCK1 expression via the PI3K-PIP3-AKT axis^{30,31}.

Furthermore, it has been reported that under a high-fat diet, palmitoleic acid decreases PCK1 expression by inhibiting SIRT3 expression³². Additionally, PCK1 expression is regulated at the post-transcriptional level. Acetylation, ubiquitination, and phosphorylation modulate the expression and biological functions of PCK1. High glucose levels can reportedly destabilize PCK1 by stimulating its acetylation, which promotes its ubiquitinylation and subsequent degradation¹⁹. Moreover, the GSK3 β -induced phosphorylation of PCK1 promotes its self-ubiquitination and degradation under high-glucose conditions²⁸. Our results showed that lipid overload with palmitic acid led to decreased PCK1 expression by upregulating the transcriptional repressor ATF3.

Therefore, the downregulation of PCK1 in NASH may be caused by multiple complex mechanisms. Hormone- and stress signaling-mediated transcription and protein modification contribute to PCK1 downregulation in NASH. In turn, decreased PCK1 levels cause intracellular lipid accumulation by enhancing the G3P pathway and promoting the expression of fatty acid transporters, such as *Cd36* and *Slc27a1*. Eventually, increased lipid accumulation in liver cells further triggers inflammation and fibrosis, likely contributing to NASH progression. Together, positive feedback may occur, that is, an increase in lipid deposition and decrease in PCK1 expression may be mutually causal factors in NASH.

3. In Fig. 1F, the mRNA level decreased from 0.2 mM PA, but the cell viability did not decrease from 0.2 mM PA (Fig. S1D). From these results, we can consider that the cytotoxicity of PA at low concentrations is not related to the decreased expression of PCK1. The possibility of decreased expression of

PCK1 independent of lipid overload should be discussed.

Author response: We appreciate the reviewer's discussion of the possibility of PCK1 downregulation. Please refer to the reply to Reviewer #2's Comment #2. We have revised the Discussion to include this point (page 21, line 395–400).

4. The authors also tried to correlate their mouse findings with human samples as well as some published datasets using bioinformatics. However, the sample size of Figure 1c and Figure 7k needs to be described. Quantitation of at least N=3-5 are needed to show the expression change of PCK1. It is suggested to describe the data in Figure 1c-e separately, instead of "PCR, immunoblotting, and immunohistochemistry (IHC) assays showing that PCK1 expression were dramatically downregulated in liver samples derived from NASH patients and 119 NASH model mice (Fig. 1c-e)."

Author response: Thank you for this comment.

1) We performed immunohistochemical quantification, as requested. The new data are presented in **Figure 1c**, **Figure 1j**, and **Figure 7l**.

2) As suggested, we have described the results of PCR, immunoblotting, and immunohistochemistry (IHC) experiments separately. Please see page 7, lines 105–107 of the revised manuscript.

5. The background color of Figure 1c for Healthy liver and NASH liver looks different. Also, the sample size needs to be shown in the figure legend for the mouse NASH model data. Whether PCK1 expression is changed at the early stage of NAFLD and whether hepatocyte-specific PCK1 expression was changed needs to be defined using the appropriate mouse NASH model.

Author response:

1) Thank you for pointing out this shortcoming. We have revised the images to have similar background colors (**Figure 1c**).

2) According to the reviewer's suggestion, we have described the sample size in the figure legend (page 45, line 953).

3) We agree that the choice of experimental models is important and must be well-justified. Animal models of NASH are typically based on different diets, such as high-fat, high-glucose, sucrose, fructose, methionine and choline-deficient, choline-deficient L-amino-defined, high-cholesterol, and cholesterol and cholate diets^{33,34}. The methionine and choline-deficient diet is commonly used to produce the most severe phenotype of NASH in the shortest time. However, the associated weight loss and lack of systemic insulin resistance make it quite different from human NASH. Importantly, there is poor concordance between the differentially expressed genes in this model and in human NASH³⁵. In this study, we aimed to induce a NASH-like phenotype by

mimicking certain dietary patterns (high-fat, high-glucose) of the disease. Thus, WT mice were fed a chow diet and NASH diet (high fat diet with drinking water containing 23.1 g/L fructose and 18.9 g/L glucose) for 8 weeks (n = 5 per group). The results showed that PCK1 expression decreased (**Rebuttal Figure 7**) after 8 weeks of NASH diet feeding. In addition, we examined PCK1 expression in the liver tissues of Alb-Cre *Pten*^{fllox/fllox} (*Pten*-KO) mice, a genetic models of NAFLD³⁶. The results showed that PCK1 expression decreased in *Pten*-KO mice. We agree that detecting PCK1 expression in the early stage of NAFLD is important. Given the limited number of animal models and lack of human clinical samples derived from early stage NAFLD, our results should be confirmed in future studies.

Rebuttal Figure 7. PCK1 expression in mouse model. **a, b** Immunohistochemistry (**a**) and western blot (**b**) assay were used to detect PCK1 expression in the liver tissues of WT mice fed the chow diet and NASH diet for 8 weeks (n = 5). **c, d** Immunohistochemistry (**c**) and western blot (**d**) assay were used to detect PCK1 expression in the liver tissues of WT and *Pten*-KO mice (n = 3).

6. Validation of hepatocyte-specific PCK1 knockout efficacy needs to be performed. The authors showed that under NASH diet, PCK1 is knocked out in the liver in Figure 4e. However, whether PCK1 is specifically deleted in the liver, but not in the other tissues, needs to be determined.

Author response: As suggested, specific deletion of PCK1 in the mouse liver tissue, but not in other tissues, was determined using western blotting. The new data are presented in **Figure 2b (Rebuttal Figure 8)** of the revised manuscript.

Rebuttal Figure 8. PCK1 protein expression in WT and L-KO mouse organs involving the heart, liver, spleen, cecum, and kidney were confirmed by immunoblotting.

7. The authors need to determine whether hepatic fibrogenesis of PCK1-LKO mice change under chow diet?

Author response: Thank you for this suggestion. Histochemistry and ELISA analysis showed that L-KO mice had prominent hepatic steatosis and increased inflammatory infiltration; however, hepatic fibrosis was not observed in mice fed the chow diet (**Rebuttal Figure 9a, b**). Additionally, PCK1 deficiency significantly increased the mRNA levels of fatty acid transport- and inflammation-related genes, such as *Cd36*, *Slc27a1*, and *Ccl2*. There were no significant changes in fibrosis-related genes, such as *Col1a1*, *Col3a1*, and *Timp-1*, in the liver of L-KO mice fed the chow diet (**Rebuttal Figure 9c**). These new data have been included in **Figure 2h–i** and **k** of the revised manuscript.

Rebuttal Figure 9. L-KO mice fed chow diet exhibit a distinct hepatic steatosis phenotype without fibrosis. **A** Paraffin-embedded liver sections were stained with H&E, Oil Red O, Sirius Red, and F4/80. Frozen sections stained with Oil Red O. Scale bars: 50 μ m. **B** Quantification of liver sections of WT and

L-KO mice fed the chow diet. **c** mRNA expression in the livers of mice fed the chow diet was analyzed using qPCR.

8. How PCK1 modulates the RhoA/PI3K/AKT in vitro and at the early stage of NASH in vivo was not examined. The possibility that the PCK1 knockout affected this pathway as a result of enhanced NASH could not be excluded.

Author response: We agree that it is important to determine the role of PCK1 knockout in the early stage of NASH.

1) *In vitro*, a human hepatic cell line (MIHA) and human hepatic stellate cell line (LX-2) were used to explore the effects of the overexpression or knockout of PCK1 on the RhoA/PI3K/AKT signaling pathway. These results are shown in **Figure 5e–i, l and m, Figure 6g–k,m-o**, and **Figure 7d–j**. Our data suggest that PCK1 knockout activates the RhoA/PI3K pathway by increasing intracellular GTP levels.

2) As described in our response to Reviewer #2's Comment #5, PCK1 expression was slightly downregulated in the early stage of NAFLD (**Rebuttal Figure 7**). The PI3K/AKT pathway was slightly activated after 8 weeks of NASH diet feeding, suggesting that the PI3K/AKT pathway was activated prior to liver fibrosis (**Rebuttal Figure 10a**). In addition, we found that PCK1 loss slightly activated the PI3K/AKT pathway after 8 weeks of NASH diet feeding (**Rebuttal Figure 10b**). Collectively, these results suggest that PCK1 knockout activates the PI3K/AKT pathway before the occurrence of NASH, but is not concomitant with the results of NASH.

Rebuttal Figure 10. Immunoblotting analysis of indicated proteins in mouse liver tissues. **a** Levels of indicated proteins in the livers of WT mice fed the chow diet or NASH diet for 8 weeks. **b** Levels of indicated proteins in the livers of WT and L-KO mice fed the NASH diet for 8 weeks.

9. The authors used MIHA and HSC (LX-2) co-culture assays. LX-2 is a human HSC cell line, while MIHA is also human hepatocyte line. Primary mouse

hepatocytes and primary mouse HSCs should be used. The authors need to make it clear when LX-2 cells are used and when primary HSCs were used and they need to specifically designate each of these two types instead of just abbreviating both as HSCs.

Author response:

1) We apologize for the lack of clarity in the original manuscript on this point. We have corrected the description in the revised manuscript (page 13, lines 244–246).

2) Thank you for this constructive suggestion. We have added new data on primary hepatocytes and primary stellate cells to the revised manuscript. The results showed that the RhoA and PI3K/AKT signaling pathway was activated in the primary hepatocytes of L-KO mice fed the NASH diet (**Rebuttal Figure 11a, b**). The activation of co-cultured primary hepatic stellate cells was partially eliminated by inhibiting the expression of AKT or RhoA in primary hepatocytes (**Rebuttal Figure 11c**). These new data are presented in **Figure 5d** and **Supplementary Figure 5c, g** in the revised manuscript.

Rebuttal Figure 11. Immunoblot analysis of indicated proteins in primary hepatocytes or primary stellate cells of mice fed the NASH diet. a, b Activation of PI3K/AKT signaling pathway (**a**) and RhoA (**b**) in primary hepatocytes isolated from WT and L-KO mice fed the NASH diet. **C** Isolated mouse primary hepatocytes from WT and L-KO mice fed the NASH diet for 24 weeks were treated with MK2206, Rhosin, or DMSO vehicle; co-cultured with primary hepatic stellate cells isolated from WT mice fed the chow diet; and evaluated to determine protein levels.

10. In Figure 5G: As expression of collagen is increased, the expression of MMPs should also change, but the expression of MMP2 was not affected. Why is the expression of MMPs (MMP1, MMP2 and so on) not affected? The authors should discuss this point in more detail.

Author response: Thank you for this comment. Extracellular matrices (ECMs) are maintained by tightly coupled processes of continuous synthesis and degradation. The degradation of ECMs is mediated by a family of proteolytic enzymes known as matrix metalloproteinases (MMPs). MMPs are released into the ECM as inactive or latent pro-enzymes and require the proteolytic removal of a small pro-peptides for activation. Once activated, these MMPs can degrade most ECM components. The activity of these enzymes is

regulated by the binding of tissue inhibitors of MMPs (TIMPs) to the catalytic site³⁷. Therefore, the metalloproteinase system may include pro-MMPs, activated-MMPs, and TIMPs. It is currently thought that the ratio of TIMPs/MMPs determines the relative rate of its degradation³⁷⁻³⁹. Thus, elevated ratios may compromise degradation, leading to the accumulation of abnormal ECM material, whereas diminished ratios are thought to lead to excessive ECM degradation. In this study, although the mRNA level of *MMP2* did not change significantly, the *TIMP-1* level significantly increased (**Figure 5h, i**), indicating the increased TIMP/MMP ratio, therefore inhibiting the enzyme activity. Moreover, we performed qPCR assays to detect other MMPs, including *MMP1* and *MMP9* (**Figure 5h, i**), and found that *MMP1* expression decreased in LX-2 cells co-cultured with PCK1-deficient hepatocytes. Therefore, *MMP2* expression was not affected or downregulated overall, whereas *TIMP-1* expression increased, contributing to the accumulation of abnormal ECM materials.

11. In Figure S4, LAMA2 is not the major laminin isoform in the liver; LAMA5 is the major form. LAMA2 levels should be measured?

Author response: We appreciate the reviewer's suggestion. We have examined the mRNA levels of *LAMA2* and *LAMA5* in the liver tissues of mice fed the NASH diet. The results showed that the expression of *LAMA2* and *LAMA5* significantly increased in the liver tissue of L-KO mice. The new data are presented in **Supplementary Figure 4a**.

12. The authors used RhoA and AKT inhibitors with NASH treatment. However, whether these compounds showed an anti-NASH effect due to an on-target effect or off-target effect was not demonstrated. More gene-specific pathways with knockout mice or AAV8-delivered hepatocyte-specific gene knockout mouse models need to be used.

Author response: We agree that this is an important point. As requested by the reviewer, we conducted a series of *in vivo* experiments by infecting L-KO mice fed the NASH diet with the pSECC lentivirus-based system, which combines both the CRISPR system and Cre recombinase²⁹, and expressed sgRhoA or sgAKT1 to silence RhoA or AKT1 in L-KO mice. The results showed that RhoA and AKT1 were significantly knocked down in the liver tissues. Silencing of RhoA and AKT1 reversed the NASH and fibrosis phenotypes induced by the NASH diet in L-KO mice. These new data are shown in **Figure 8f-i** and **Supplementary Figure 7**.

13. The Methods section showed that N=10 normal and N=36 NASH human liver samples were used. However, the qPCR data were not found. Only one representative IHC figure was shown with no statistical analyses in Figure 1C and Figure 7.

Author response: We apologize for the lack of clarity on this point. The n = 10

normal and n = 36 NASH human liver samples described in the Methods section were paraffin sections rather than fresh liver tissues. In clinical practice, fresh liver tissue from patients with NASH is difficult to obtain, and thus we did not conduct q-PCR to detect *PCK1* mRNA expression levels in the liver tissue of patients with NASH. Paraffin sections from patients with NASH were retrospectively collected, and only immunohistochemical staining was performed. We have revised this description in the revised manuscript (page 27, line 524).

14. In isolation of primary HSCs, in line 493, it was stated as “HSCs were plated in tissue culture dishes in DMEM.” Detailed information of the media need to be provided - 10% FBS or other supplements? Could primary HSCs attach to the tissue culture dishes? Usually coated plates are needed for seeding primary HSCs.

Author response: We apologize that the information was incomplete. After purification via Nycodenz gradient centrifugation, the isolated HSC cells were seeded on collagen-coated plates and stabilized in DMEM supplemented with 10% FBS and 1% penicillin-streptomycin as previously described⁴⁰. More detailed information has been included in the revised manuscript (page 30, lines 580–582).

15. As noted above, whether the proposed pathway is a result or a causal factor for the phenotype was not established. Are the proposed pathways changed during the early stage of NASH diet feeding compared to chow diet? Could *PCK1* knockout still promote NASH when the pathway is blocked? Does hepatocyte-specific *PCK1* knockout affect the PI3K/AKT pathway in vitro?

Author response: We appreciate the reviewer’s comments on the role of *PCK1* knockout in the early stage of NASH and agree that it is important to distinguish whether the proposed pathway is a resultant or causal factor.

1) In response to Reviewer #2’s Comment #8, to explore whether the proposed pathways changed during the early stages of NASH, WT mice were fed chow and NASH diets for 8 weeks (n = 5 per group). After 8 weeks, western blotting was performed to detect the activation of the PI3K/AKT pathway in the mouse liver tissue. The result showed that the PI3K/AKT signaling pathway was slightly activated in the early stage of NAFLD, possibly before NASH occurred (**Rebuttal Figure 10a**).

2) According to the results in Figure 8, genetic and pharmacological disruption of RhoA and AKT1 partially reversed, but did not completely inhibit, *PCK1* deficiency-mediated NASH progression. This may be because *PCK1* knockout promotes the progression of NASH through multiple mechanisms. For example, genes involved in lipid metabolism, such as *Cd36* and *Cidec*, were highly expressed in L-KO mice, both of which are reportedly associated with the progression of steatohepatitis^{41,42}. Therefore, the inhibition of RhoA and

AKT1 only partially eliminated fibrosis, inflammation, and lipid droplet infiltration in NASH.

3) Further, the human hepatic cell line MIHA and human hepatic stellate cell line (LX-2) were used to examine the effects of overexpression or knockout of PCK1 on the RhoA/PI3K/AKT signaling pathway *in vitro*. These results are shown in **Figure 5e–i, l, and m**; **Figure 6g–k, m–o**; and **Figure 7d–j**. Collectively, our data suggest that hepatocyte-specific PCK1 knockout activates the RhoA/PI3K pathway by increasing intracellular GTP levels.

16. In Fig. 2 and Fig. 3, the qPCR of inflammation markers and fibrosis markers need to be shown.

Author response: Thank you for pointing out this issue. As described in our response to Reviewer #2's Comment #7, we examined the expression of genes related to inflammation and fibrosis in the liver tissue of mice fed the NASH diet and found that the expression of inflammatory factors, lipogenic enzymes, and fibrogenesis-associated genes was upregulated in L-KO mice (**Supplementary Figure 2d–f** in the original manuscript). We also examined the expression of inflammation and fibrosis markers in L-KO mice fed the chow diet, which revealed no significant changes in fibrosis-related genes in the liver of L-KO mice. Interestingly, the expression of inflammation-related genes *Ccl2* and *Il-1b* was upregulated in the liver tissues of L-KO mice. The activation of inflammatory markers may be due to excessive lipid accumulation in the liver. We have included these new data in **Figure 2h–i and k** and restructured Supplementary Figure 2 to clarify these points. (page 9, lines 145–149)

17. The levels of serum PDGF-AA in the NASH model in WT and L-KO mice need to be determined?

Author response: Thank you for these constructive suggestions. As the reviewer suggested, serum PDGF-AA levels in WT and L-KO mice fed the NASH diet were determined using ELISA. The results showed that serum PDGF-AA levels were increased in L-KO mice fed the NASH diet. The new data have been included in **Figure 6f** of the revised manuscript (page 15, line 267).

18. Hepatocyte PCK1 downregulation is correlated with enhanced NASH progression. Does forced expression of PCK1 in liver reverse the chow-fed and NASH diet-induced phenotype of the PCK1 L-KO mice?

Author response: Thank you for this constructive suggestion. We conducted a series of additional *in vivo* experiments. Using AAV8-TBG viral vectors to restore PCK1 expression in hepatocytes, the forced expression of PCK1 in the liver of L-KO mice reversed the phenotype induced by the chow and NASH diets. Mice with PCK1 re-expression had a lower liver weight, body weight,

serum liver enzymes. Moreover, lipid deposition, inflammation, and fibrosis significantly improved in L-KO mice injected with AAV-*Pck1*. We have described these experiments in the revised manuscript (**Figure 3h–l** and **Supplementary Figure 2h–k**, page 10-11, lines 174–191).

19. Wild-type mice on the NASH diet with a fully developed NASH liver phenotype should be treated with Rhosin and MK2206 to determine if blocking RhoA/PI3K/AKT is of translational value in treating NASH.

Author response: Thank you for this suggestion. According this advice, we treated WT mice fed the NASH diet with Rhosin and MK2206 for 24 weeks; the results are shown below (**Rebuttal Figure 12**). Rhosin and MK2206 treatment decreased the weight, liver weight, and lipid accumulation (TG and FFA levels) in the liver and improved the inflammation and fibrosis phenotype in WT mice, suggesting that treatment with Rhosin and MK2206 in WT mice fed the NASH diet partially alleviated the progression of NASH.

Rebuttal Figure 12. Rhosin and MK2206 treatments inhibited NASH progression in WT mice fed the NASH diet. WT mice were fed the NASH diet for 24 weeks, and therapeutic treatment with AKT or RhoA inhibitor was initiated at 16 weeks (n = 5). **a** Representative whole body, gross liver morphology, and liver weight. **b** Serum ALT, AST, TG, TC, and FFA levels were

measured. **C** Paraffin-embedded liver sections were stained with H&E, Sirius Red, and F4/80. Frozen sections were stained with Oil Red O. Scale bars: 50 μ m. **d** Quantification of Oil red O, Sirius red, and IHC staining. **e** NAS scores of each group.

20. Figure 8: AKT inhibitor has been shown to improve the pathophysiology of NASH. However, it is known that inhibition of AKT not only inhibits fibrosis via HSC activation, but also affects TGF-beta activation. This means that the model is not suitable for focusing only on the inhibition of HSC activation. Therefore, if the authors are going to evaluate AKT inhibitors, they should simultaneously discuss the effect of changes in other factors, such as TGF-beta, on NASH as well as HSC activation.

Author response: Thank you for pointing out this issue. We agree that AKT inhibitors may affect other factors as well, and we did not exclude the possibility that the AKT inhibitor affects TGF- β activation in our model. It has been proposed that the PI3K/AKT signaling pathway is a master regulator of fibrosis, particularly in idiopathic pulmonary fibrosis⁴³. In this study, we proposed that the activated PI3K/AKT signaling pathway promotes fibrosis by increasing paracrine PDGF-AA. In fact, because of the important role of PI3K/AKT in regulating receptor-mediated signal transduction, AKT inhibition may not only affect TGF- β ^{44,45} but also participate in fibrosis by regulating its downstream targets, such as NF- κ B⁴⁶, Hippo⁴⁷⁻⁴⁹, and Notch signaling pathways, all of which are reportedly associated with fibrosis or NASH^{50,51}. We have described these points in detail in the revised Discussion (page 25, lines 472-474).

Minor points:

1. What is the background of the mouse lines used in this study?

Author response: We have described the genetic background of the mice in the Methods section (page 26, lines 485-491). *Pck1*^{loxp/loxp} mice had a 129S6/SvEv background. Transgenic mice carrying Cre-recombinase driven by an albumin promoter (*Alb-Cre*) had a C57/BL6 background. *Pck1*^{loxp/loxp} mice were bred with *Alb-Cre* mice for five generations prior to the experiments. *Pck1*^{loxp/loxp} mice from the same breed were used as controls (wild-type, WT).

2. Line 63: "NAFLD may progress to NASH, a more serious form." may not be exactly correct. NAFLD includes simple fatty liver and NASH. It could be "nonalcoholic fatty liver (NAFL) could progress to nonalcoholic steatohepatitis (NASH)."

Author response: We apologize for the inaccurate description. We have altered this information in the revised manuscript (page 4, lines 51).

3. In Figure. 1H, BSA is in the culture medium and is also present in the

PA-treated group, so the author should change the wording in the figure panel.
Author response: Thank you for pointing this out. We have revised the figure panel (**Figure 1h**) and figure legends (page 45, lines 956).

Reviewer #3 (Remarks to the Author):

This is a well performed study which examines the role of Pck1 in the NASH. It builds on earlier work from 2010 which shows that low PCK1 levels result in significant dysregulation of lipid metabolism and hepatic lipid content. This manuscript has expanded on this by using hepatocyte specific PCK1 KO mice, and demonstrating activation of RhoA/P13K/AKT pathway which contributes to PDGF-AA secretion by hepatocytes and activation of HSC.

Author response: We appreciate the reviewer's positive comments on our study. We hope that the extensive new data and improved presentation of the revised manuscript provide strong support for our findings.

Overall the work is incremental, as the general effect of PCK1 was known, and also because other studies have shown that disruption of gluconeogenesis leads to severe hepatic steatosis. The connection with HSC activation is somewhat peripheral because we know that all forms of hepatocyte injury results in HSC activation.

Author response: We thank the reviewer for evaluating our manuscript and providing critical comments. The previous version of our manuscript did not adequately emphasize the most important findings of our work, and additional data were needed to confirm our conclusions. We have addressed these points in the revised manuscript.

(i) Our data show that PCK1 is downregulated in patients with NASH and mouse models of NASH. Furthermore, the identification of a key transcription factor leading to the downregulation of PCK1 in the context of NASH is novel and can be observed at the clinical level (e.g., we found a significant negative correlation between *ATF3* and *PCK1* expression based on a public transcriptome dataset (GEO: GSE135251)). In addition, AAV-mediated restoration of hepatic PCK1 alleviates the NASH phenotype in Pck1-deficient mice, which has important clinical implications.

(ii) Unlike previous studies showing that the disruption of gluconeogenesis leads to severe hepatic steatosis through *de novo* lipogenesis, our data revealed that the depletion of PCK1 promoted the transportation of fatty acids by upregulating translocators, such as *Cd36* and *Slc27a1*, and enhanced the G3P pathway, thus leading to intracellular lipid accumulation. Notably, although we confirmed some previous observations, there is no conceptual overlap between these studies and the main finding of our paper.

(iii) Our data expand the current knowledge of NASH pathology-associated

mechanisms (e.g., hepatic loss of Pck1 promotes inflammation and fibrogenesis, in addition to hepatic steatosis). Our work also revealed that Pck1-deficiency enhances the secretion of paracrine PDGF-AA, promoting the activation of hepatic stellate cells in hepatic fibrosis progression. Furthermore, we examined whether PCK1 deficiency increases intracellular GTP levels, further activating the RhoA/PI3K pathway. We showed that PCK1 deficiency upregulated intracellular GTP levels, further activated the RhoA/PI3K/PDGF-AA axis, and led to fibrosis. In addition, genetic and pharmacological disruption of RhoA and AKT1 in L-KO mice effectively mitigated the NASH phenotypes, providing new concepts and methods for clinical treatment.

Rebuttal Table 2. New discoveries reported in this manuscript.

What was known	Our findings
Nothing known the role of PCK1 in NASH	 - PCK1 is downregulated in patients with NASH and mouse models of NASH - Palmitic acids inhibited PCK1 transcription by upregulating ATF3
Pck1 hepatocyte-specific depletion leads to steatosis	 - Hepatic loss of Pck1 promotes inflammation and fibrogenesis in mice fed a NASH diet - AAV-mediated restoration of hepatic PCK1 alleviates NASH phenotype in Pck1-deficient mice
Limited experiments suggested that loss of PCK1 affects the expression of enzymes related to de novo lipogenesis, such as ACC and FAS	 - Transcriptomic and metabolomics analyses confirmed that the depletion of PCK1 promoted the expression of fatty acid transporters, such as Cd36 and Slc27a1, and upregulated the G3P pathway, leading to intracellular lipid accumulation
Nothing known regarding the link between PCK1 deficiency and hepatic stellate cell activation	 - HPLC analysis and co-culture experiments demonstrated that PCK1 loss activated the RhoA/PI3K/AKT axis, increased the secretion of paracrine PDGF-AA, and promoted the activation of hepatic stellate cells, leading to liver fibrosis. - The genetic and pharmacological

Minor point when L-KO mice are introduced (page 8 of PDF) please explain what they are. The reader must go into the methods to find this information.

Author response: Thank you for pointing this out. We apologize for the lack of clarity in our manuscript, and we have explained that L-KO is the abbreviation for liver-specific gene *Pck1* knockout mice at the first instance where it is mentioned in the Results section (page 8, lines 133–134).

References

1. Burgess, S. C. *et al.* Cytosolic Phosphoenolpyruvate Carboxykinase Does Not Solely Control the Rate of Hepatic Gluconeogenesis in the Intact Mouse Liver. *Cell Metabolism* **5**, 313–320 (2007).
2. Méndez-Lucas, A. *et al.* PEPCK-M expression in mouse liver potentiates, not replaces, PEPCK-C mediated gluconeogenesis. *Journal of Hepatology* **59**, 105–113 (2013).
3. Burgess, S. C. *et al.* Impaired Tricarboxylic Acid Cycle Activity in Mouse Livers Lacking Cytosolic Phosphoenolpyruvate Carboxykinase. *Journal of Biological Chemistry* **279**, 48941–48949 (2004).
4. She, P. *et al.* Phosphoenolpyruvate Carboxykinase Is Necessary for the Integration of Hepatic Energy Metabolism. *MOL. CELL. BIOL.* **10**.
5. Vieira, P. *et al.* Cytosolic phosphoenolpyruvate carboxykinase deficiency: Expanding the clinical phenotype and novel laboratory findings. *J of Inher Metab Disea* **45**, 223–234 (2022).
6. Santra, S. *et al.* Cytosolic phosphoenolpyruvate carboxykinase deficiency presenting with acute liver failure following gastroenteritis. *Molecular Genetics and Metabolism* **118**, 21–27 (2016).

7. Castillo-Armengol, J., Fajas, L. & Lopez-Mejia, I. C. Inter-organ communication: a gatekeeper for metabolic health. *EMBO Rep* **20**, (2019).
8. Priest, C. & Tontonoz, P. Inter-organ cross-talk in metabolic syndrome. *Nat Metab* **1**, 1177–1188 (2019).
9. Beale, E. G., Harvey, B. J. & Forest, C. PCK1 and PCK2 as candidate diabetes and obesity genes. *Cell Biochem Biophys* **48**, 89–95 (2007).
10. Beale, E., Hammer, R., Antoine, B. & Forest, C. Disregulated glyceroneogenesis: PCK1 as a candidate diabetes and obesity gene. *Trends in Endocrinology and Metabolism* **15**, 129–135 (2004).
11. Heeren, J. & Scheja, L. Metabolic-associated fatty liver disease and lipoprotein metabolism. *Molecular Metabolism* **50**, 101238 (2021).
12. Ebbert, J. & Jensen, M. Fat Depots, Free Fatty Acids, and Dyslipidemia. *Nutrients* **5**, 498–508 (2013).
13. Kahn, B. B. & Flier, J. S. Obesity and insulin resistance. *J. Clin. Invest.* **106**, 473–481 (2000).
14. Abderrahmani, A. *et al.* Increased Hepatic PDGF-AA Signaling Mediates Liver Insulin Resistance in Obesity-Associated Type 2 Diabetes. *Diabetes* **67**, 1310–1321 (2018).
15. Miquilena-Colina, M. E. *et al.* Hepatic fatty acid translocase CD36 upregulation is associated with insulin resistance, hyperinsulinaemia and increased steatosis in non-alcoholic steatohepatitis and chronic hepatitis C. *Gut* **60**, 1394–1402 (2011).
16. Jesinkey, S. R. *et al.* Mitochondrial GTP Links Nutrient Sensing to β Cell Health, Mitochondrial Morphology, and Insulin Secretion Independent of OxPhos. *Cell Reports* **28**, 759-772.e10 (2019).

17. Zouali, H. A susceptibility locus for early-onset non-insulin dependent (type 2) diabetes mellitus maps to chromosome 20q, proximal to the phosphoenolpyruvate carboxykinase gene. *Human Molecular Genetics* **6**, 1401–1408 (1997).
18. Hani EH, Zouali H, Philippi A, et al. Indication for genetic linkage of the phosphoenolpyruvate carboxykinase (PCK1) gene region on chromosome 20q to non-insulin-dependent diabetes mellitus. *Diabetes Metab.* 1996;22(6):451-454.
19. Jiang, W. *et al.* Acetylation Regulates Gluconeogenesis by Promoting PEPCK1 Degradation via Recruiting the UBR5 Ubiquitin Ligase. *Molecular Cell* **43**, 33–44 (2011).
20. Higuchi, M., Masuyama, N., Fukui, Y., Suzuki, A. & Gotoh, Y. Akt mediates Rac/Cdc42-regulated cell motility in growth factor-stimulated cells and in invasive PTEN knockout cells. *Current Biology* **11**, 1958–1962 (2001).
21. Dou, C. *et al.* P300 Acetyltransferase Mediates Stiffness-Induced Activation of Hepatic Stellate Cells Into Tumor-Promoting Myofibroblasts. *Gastroenterology* **154**, 2209-2221.e14 (2018).
22. Calvayrac, O. *et al.* The RAS -related GTP ase RHOB confers resistance to EGFR -tyrosine kinase inhibitors in non-small-cell lung cancer via an AKT -dependent mechanism. *EMBO Mol Med* **9**, 238–250 (2017).
23. Heasman, S. J. & Ridley, A. J. Mammalian Rho GTPases: new insights into their functions from in vivo studies. *Nat Rev Mol Cell Biol* **9**, 690–701 (2008).
24. Syed, I., Kyathanahalli, C. N. & Kowluru, A. Phagocyte-like NADPH oxidase generates ROS in INS 832/13 cells and rat islets: role of protein prenylation. *American Journal of Physiology-Regulatory, Integrative and Comparative Physiology* **300**, R756–R762 (2011).

25. Vieira, P. *et al.* Novel homozygous PCK1 mutation causing cytosolic phosphoenolpyruvate carboxykinase deficiency presenting as childhood hypoglycemia, an abnormal pattern of urine metabolites and liver dysfunction. *Molecular Genetics and Metabolism* **120**, 337–341 (2017).
26. Xiang, J. *et al.* Gluconeogenic enzyme PCK1 deficiency promotes CHK2 O-GlcNAcylation and hepatocellular carcinoma growth upon glucose deprivation. *Journal of Clinical Investigation* **131**, e144703 (2021).
27. Grasmann, G., Smolle, E., Olschewski, H. & Leithner, K. Gluconeogenesis in cancer cells – Repurposing of a starvation-induced metabolic pathway? *Biochimica et Biophysica Acta (BBA) - Reviews on Cancer* **1872**, 24–36 (2019).
28. Latorre-Muro, P. *et al.* Dynamic Acetylation of Phosphoenolpyruvate Carboxykinase Toggles Enzyme Activity between Gluconeogenic and Anaplerotic Reactions. *Molecular Cell* **71**, 718-732.e9 (2018).
29. Sánchez-Rivera, F. J. *et al.* Rapid modelling of cooperating genetic events in cancer through somatic genome editing. *Nature* **516**, 428–431 (2014).
30. Hall, R. K. *et al.* Regulation of phosphoenolpyruvate carboxykinase and insulin-like growth factor-binding protein-1 gene expression by insulin. The role of winged helix/forkhead proteins. *Journal of Biological Chemistry* **276**, 23212 (2001).
31. Hatting, M., Tavares, C. D. J., Sharabi, K., Rines, A. K. & Puigserver, P. Insulin regulation of gluconeogenesis: Insulin regulation of gluconeogenesis. *Ann. N.Y. Acad. Sci.* **1411**, 21–35 (2018).
32. Guo, X. *et al.* The Role of Palmitoleic Acid in Regulating Hepatic Gluconeogenesis through SIRT3 in Obese Mice. *Nutrients* **14**, 1482 (2022).

33. Asgharpour, A. *et al.* A diet-induced animal model of non-alcoholic fatty liver disease and hepatocellular cancer. *Journal of Hepatology* **65**, 579–588 (2016).
34. Clapper, J. R. *et al.* Diet-induced mouse model of fatty liver disease and nonalcoholic steatohepatitis reflecting clinical disease progression and methods of assessment. *American Journal of Physiology-Gastrointestinal and Liver Physiology* **305**, G483–G495 (2013).
35. Santhekadur, P. K., Kumar, D. P. & Sanyal, A. J. Preclinical models of non-alcoholic fatty liver disease. *Journal of Hepatology* **68**, 230–237 (2018).
36. Febbraio, M. A. *et al.* Preclinical Models for Studying NASH-Driven HCC: How Useful Are They? *Cell Metabolism* **29**, 18–26 (2019).
37. Clark, I., Swingler, T., Sampieri, C. & Edwards, D. The regulation of matrix metalloproteinases and their inhibitors. *The International Journal of Biochemistry & Cell Biology* **40**, 1362–1378 (2008).
38. Hussain, A. A., Lee, Y. & Marshall, J. Understanding the complexity of the matrix metalloproteinase system and its relevance to age-related diseases: Age-related macular degeneration and Alzheimer’s disease. *Progress in Retinal and Eye Research* **74**, 100775 (2020).
39. Duarte, S., Baber, J., Fujii, T. & Coito, A. J. Matrix metalloproteinases in liver injury, repair and fibrosis. *Matrix Biology* **44–46**, 147–156 (2015).
40. Choi, W.-M. *et al.* Glutamate Signaling in Hepatic Stellate Cells Drives Alcoholic Steatosis. *Cell Metabolism* **30**, 877–889.e7 (2019).
41. Zhao, L. *et al.* CD36 palmitoylation disrupts free fatty acid metabolism and promotes tissue inflammation in non-alcoholic steatohepatitis. *Journal of Hepatology* **69**, 705–717 (2018).

42. Xu, M.-J. *et al.* Fat-Specific Protein 27/CIDEA Promotes Development of Alcoholic Steatohepatitis in Mice and Humans. *Gastroenterology* **149**, 1030-1041.e6 (2015).
43. Wang, J. *et al.* Targeting PI3K/AKT signaling for treatment of idiopathic pulmonary fibrosis. *Acta Pharmaceutica Sinica B* **12**, 18–32 (2022).
44. Korfhagen, T. R. *et al.* Rapamycin Prevents Transforming Growth Factor- α -Induced Pulmonary Fibrosis. *Am J Respir Cell Mol Biol* **41**, 562–572 (2009).
45. Le Cras, T. D. *et al.* Inhibition of PI3K by PX-866 Prevents Transforming Growth Factor- α -Induced Pulmonary Fibrosis. *The American Journal of Pathology* **176**, 679–686 (2010).
46. Han, M. H. *et al.* Flavonoids Isolated from Flowers of *Lonicera japonica* Thunb. Inhibit Inflammatory Responses in BV2 Microglial Cells by Suppressing TNF- α and IL- β Through PI3K/Akt/NF- κ B Signaling Pathways: *Lonicera Japonica* Thunb. Inhibit Inflammatory Responses. *Phytother. Res.* **30**, 1824–1832 (2016).
47. Fan, R., Kim, N.-G. & Gumbiner, B. M. Regulation of Hippo pathway by mitogenic growth factors via phosphoinositide 3-kinase and phosphoinositide-dependent kinase-1. *Proc. Natl. Acad. Sci. U.S.A.* **110**, 2569–2574 (2013).
48. Zhao, Y. *et al.* PI3K Positively Regulates YAP and TAZ in Mammary Tumorigenesis Through Multiple Signaling Pathways. *Molecular Cancer Research* **16**, 1046–1058 (2018).
49. Jeong, S.-H. & Lim, D.-S. Insulin receptor substrate 2: a bridge between Hippo and AKT pathways. *BMB Rep.* **51**, 209–210 (2018).
50. Schwabe, R. F., Tabas, I. & Pajvani, U. B. Mechanisms of Fibrosis Development in Nonalcoholic Steatohepatitis. *Gastroenterology* **158**, 1913–1928 (2020).

51. Zhu, C., Tabas, I., Schwabe, R. F. & Pajvani, U. B. Maladaptive regeneration — the reawakening of developmental pathways in NASH and fibrosis. *Nat Rev Gastroenterol Hepatol* **18**, 131–142 (2021).

REVIEWER COMMENTS

Reviewer #1 (Remarks to the Author):

The authors have addressed my concerns and there is no more comments.

Reviewer #2 (Remarks to the Author):

The authors have positively responded to most of the points raised in the review. However, there are a few other issues that need attention before this study can be published in Nature Communications.

1. Figure 8: Sirius red staining data need to be provided for Rhosin and MK2206-treated mice. Serum cytokine levels by ELISA is more important than showing the liver cytokine levels using ELISA. The color of livers for Figure 8f and Figure 8b look different; Figure 8b looks pale while Figure 8f looks more normal. Based on the liver pictures in this response letter and Figure 8f, the other liver pictures that show pale and white livers are confusing. The authors should address this discrepancy or delete these low-quality liver pictures. The Sirius red staining in Figure S7 is of low quality indicating low fibrosis in this model, with only the area around the vein showing significant staining, which is inconsistent with the Sirius red staining in Figure 3e. In addition, Figure 3e shows no lipid droplets in the Sirius red staining as well as α -SMA and F4/80 staining, inconsistent with the H&E histology in this Figure 3e. Higher amplification of the Sirius red staining as well as the α -SMA and F4/80 on Figure 3e is needed. The shape of Sirius red staining of Figure 3e is inconsistent with Figure 3k. At least the Figure 3k shows lipid drops around the vein area.

2. In Figure S3h: Measure Pck1 mRNA and PCK1 protein levels in the liver.

3. Author response to Point 2 should be added to the Discussion. Pck1 mRNA data or the luciferase reporter assays would be more informative to show the role of ATF3 in Pck1 transactivation beyond the western blot of PCK1 in MIHA in Figure 1I, when ChIP data is used to demonstrate the ATF3 engagement on the Pck1 promoter. Primary mouse hepatocytes would be preferred instead of the MIHA cell line. PA 0.2 mM is a relatively low dose compared to the general use of PA 0.4 mM and PA 0.5 mM in the in vitro model. Whether this dose of 0.2 mM PA induces hepatocyte steatosis and inflammation needs to be verified in primary hepatocytes. Does the PCK1 KO exhibit PA-induced hepatocyte steatosis and inflammation in vitro?

4. Pck1 mRNA is needed for the common tissues including liver, intestine, white adipose, brown adipose, kidney, spleen. N=3 is expected to be found for liver western blot. Did the authors use GAPDH or ACTB for normalizing the Western blot? For GAPDH, the MW is 37 kD, while ACTB has a MW of 42 kD instead of 37 kD as shown above. The authors need to correct this for all ACTB Mr in this manuscript, not only limited to this figure. The uncut blot data for all western blot need to be provided in supplemental material.

5. The mRNA nomenclature needs to be corrected throughout the manuscript and figures (<https://www.ncbi.nlm.nih.gov/gene>). For examples, Tnf- α needs to be Tnfa. Timp-1 needs to be Timp1. Il-10 needs to be Il10. Il-1b needs to be Il1b. Il-6 needs to be Il6.

6. What cells contribute to RhoA/PI3K/AKT expression change in the liver, hepatocytes or HSC? The direct effect of hepatocyte PCK1 knockout in RhoA/PI3K/AKT modulation should be examined with primary hepatocytes when the investigated PCK1 is located in hepatocytes. It is meaningless to show the effect of PCK1 overexpression in the RhoA/PI3K/AKT in hepatic stellate cells when the current study is trying to study the role of hepatocyte PCK1 function gain and loss. Could increasing GTP levels in hepatocytes modulate RhoA/PI3K/AKT pathway?

7. "As described in our response to Reviewer #2's Comment #5, PCK1 expression was slightly downregulated in the early stage of NAFLD (Rebuttal Figure 7). The PI3K/AKT pathway was slightly activated after 8 weeks of NASH diet feeding, suggesting that the PI3K/AKT pathway was activated prior to liver fibrosis (Rebuttal Figure 10a). In addition, we found that PCK1 loss slightly activated the PI3K/AKT pathway after 8 weeks of NASH diet feeding (Rebuttal Figure 10b). Collectively, these results suggest that PCK1 knockout activates the PI3K/AKT pathway before the occurrence of NASH, but is not concomitant with the results of NASH."

This response answered how PCK1 is changed at the early stage of NASH, but did not address how hepatocyte PCK1 modulates the proposed mechanistic pathway under NASH and chow diet. What is the phenotype after 8-week NASH feeding? It is expected that the LKO mice should show a robust enhanced change of NASH-related phenotype compared to WT mice since LKO mice could show a phenotype under chow diet. Thus, it is not suitable to conclude that "these results suggest that PCK1 knockout activates the PI3K/AKT pathway before the occurrence of NASH, but is not concomitant with the results of NASH." What is the RhoA/PI3K/AKT signaling under chow diet and the early stage such as 1-2-week NASH diet-feeding in LKO mice and WT mice before LKO showed an enhanced NASH phenotype?

8. For co-culture experiments, isolation of primary hepatocytes at the early stage of NASH diet-fed LKO mice to coculture with primary HSC is needed if the authors claim this as a causal factor and it is more important to test the levels of PDGF-AA signaling and the secretion in the supernatant for co-culture experiments. Could Rho/AKT/PI3K signaling activation or other mechanisms modulated by hepatocyte PCK1 lead to the increased secretion of PDGF-AA?

9. The sgRhoA and SgAkt1 add evidence of target-specific effects. Since the authors did sgRhoA and SgAkt1 experiments separately from the inhibitor treatment, the off-target effect of those inhibitors could not be excluded. This needs to be discussed as a limitation of showing data of these inhibitors unless the authors could show that the inhibitors rescue the phenotype depending on the presence of RhoA or Akt1 at least in vitro PA-treated primary hepatocyte model.

10. The information for human samples is incomplete. Only "clinical informations (should be

information) of patients with NASH” was provided in Supplemental Table 1, which is unacceptable. The information for normal human (n=10) needs to be provided. All details for individual information need to be provided to generate the statistics and averages, but not only described the average levels plus SEM.

11. The Rebuttal Figure 10a only investigates a role of the NASH diet at 8 weeks (relatively early stage) in modulation of PCK1 expression. It does not address how hepatocyte PCK1 modulates the proposed PI3K/AKT signaling at the early stage so that this proposed mechanism could be a causal factor to explain the phenotype. Many pathways could be disturbed at the end of NASH diet feeding as a result of the enhanced NASH phenotype, but not all changes could be a causal factor. It is important to show whether hepatocyte PCK1 upregulates the proposed mechanistic pathway before the hepatocyte PCK1 enhanced the NASH phenotype or under basic conditions or at early stages if the enhanced phenotype occurs at the beginning stage.

12. Serum PDGF-AA levels at the early stage of NASH needs to be determined and the ELISA of the supernatant for PDFG-AA levels in primary hepatocytes from LKO and WT mice needs to be determined.

13. “We conducted a series of additional in vivo experiments. Using AAV8-TBG viral vectors to restore PCK1 expression in hepatocytes, the forced expression of PCK1 in the liver of L-KO mice reversed the phenotype induced by the chow and NASH diets. Mice with PCK1 re-expression had a lower liver weight, body weight, serum liver enzymes. Moreover, lipid deposition, inflammation, and fibrosis significantly improved in L-KO mice injected with AAV-Pck1. We have described these experiments in the revised manuscript (Figure 3h–l and Supplementary Figure 2h–k, page 10-11, lines 174–191).”

Hepatocyte PCK1 needs to be quantified to determine whether the forced expression worked. The source of the materials listed in Table S2 needs to be clarified. It is not acceptable to show the source of material as “lab stock”. The detailed information for all new materials used in the revision needs to be provided and updated in Supplemental Tables.

14. “Thank you for this suggestion. According this advice, we treated WT mice fed the NASH diet with Rhosin and MK2206 for 24 weeks; the results are shown below (Rebuttal Figure 12). Rhosin and MK2206 treatment decreased the weight, liver weight, and lipid accumulation (TG and FFA levels) in the liver and improved the inflammation and fibrosis phenotype in WT mice, suggesting that treatment with Rhosin and MK2206 in WT mice fed the NASH diet partially alleviated the progression of NASH.”

This experiment did not address the therapeutic effect, but still the preventive effect? It is expected that the mice need to be fed a NASH diet first to establish NASH model and then treated with inhibitors. The body weight of WT mice after 24-week HFD feeding is only 32 g and the histology data showed a very mild and weak steatosis as well as liver fibrosis, indicating that this NASH model is a very mild NASH model. In addition, the NASH-inducing diet is too mild in the current model, which was also evidenced by 1.2 times of liver weight increase as well as liver lipid increase in the 24-week NASH diet-fed WT mice in Figure 2c compared to Figure 3e. Did this NASH diet really induce a substantially increase of NASH,

fatty liver, inflammation and fibrosis? The NASH diet D0910031 (Research diets) would be more potent and suitable to induce NASH and fibrosis than the current diet, which is too mild and weak to induce NASH. The Sv129 background of the LKO mice and WT mice may render the mouse insensitive to the NASH induction. The brown coat color for these mice does not indicate that they were backcrossed with C57BL/6J mice for 5 generations. C57BL/6J mice are preferred for metabolic studies. This disadvantage needs to be addressed in the Discussion.

15. The authors described that the "... prevents NASH development in vivo." in the Figure 8 legend. Is this preventive effect or therapeutic effect?

Point-by-point response to the reviewer's comments.

Reviewer #1 (Remarks to the Author):

The authors have addressed my concerns and there is no more comments.

Author response: We appreciate your constructive comments to improve our manuscript's quality.

Reviewer #2 (Remarks to the Author):

The authors have positively responded to most of the points raised in the review. However, there are a few other issues that need attention before this study can be published in Nature Communications.

Author response: We thank you for your positive remarks, and we are happy to address your concerns in point-by-point responses herein.

1. Figure 8: Sirius red staining data need to be provided for Rhosin and MK2206-treated mice. Serum cytokine levels by ELISA is more important than showing the liver cytokine levels using ELISA. The color of livers for Figure 8f and Figure 8b look different; Figure 8b looks pale while Figure 8f looks more normal. Based on the liver pictures in this response letter and Figure 8f, the other liver pictures that show pale and white livers are confusing. The authors should address this discrepancy or delete these low-quality liver pictures. The Sirius red staining in Figure S7 is of low quality indicating low fibrosis in this model, with only the area around the vein showing significant staining, which is inconsistent with the Sirius red staining in Figure 3e. In addition, Figure 3e shows no lipid droplets in the Sirius red staining as well as a-SMA and F4/80 staining, inconsistent with the H&E histology in this Figure 3e. Higher amplification of the Sirius red staining as well as the a-SMA and F4/80 on Figure 3e is needed. The shape of Sirius red staining of Figure 3e is inconsistent with Figure 3k. At least the Figure 3k shows lipid drops around the vein area.

Author response: Thank you for your constructive suggestions.

1. As suggested by the reviewer, we have provided Sirius Red staining data for Rhosin and MK2206-treated mice in **Figure. 8d (Rebuttal Figure 1a)**.
2. Serum cytokine levels were detected by ELISA as recommended. Rhosin and MK2206 treatment decreased serum levels of proinflammatory cytokines, such as IL-6 and TNF- α , in L-KO mice, suggesting that treatment with Rhosin and MK2206 in L-KO mice fed the NASH diet partially alleviated the progression of NASH (**Figure 8c, Rebuttal Figure 1b**).
3. We have removed all low-quality liver images throughout the manuscript, as

suggested by the reviewer.

4. As requested, we have replaced the Sirius Red staining in **Supplementary Figure 7b (Rebuttal Figure 1c)** and **Figure 3e (Rebuttal Figure 1d)**, as well as added higher-resolution images of the Sirius Red, α -SMA, and F4/80 staining in **Figure 3e**. As expected, liver sections from L-KO mice displayed increased fat droplets, more severe fibrosis, and greater macrophage infiltration (**Figure 3e**). We have updated the H&E histology in **Figure 3e** and **3k**. The genetic inhibition of AKT or RhoA significantly alleviated hepatic inflammation and fibrosis in L-KO mice (**Supplementary Figure 7b**).

Rebuttal Figure 1. **a** Paraffin-embedded liver sections were stained with hematoxylin and eosin (H&E), Sirius Red, or immunostained for F4/80, COL3A1, and α -SMA. Frozen sections were stained with Oil Red O. Scale bars: 50 μ m. **b** TNF- α and IL-6 levels in serum were detected using enzyme-linked immunosorbent assay (ELISA). **c** Representative images of H&E, Sirius Red, Oil Red O, and immunohistochemical staining of liver tissues from L-KO mice

infected with the control pSECC-sgCtrl lentivirus, the pSECC-sgAKT1, or the pSECC-sgRhoA lentivirus. Scale bars: 50 μ m. **d** Paraffin-embedded liver sections were stained with H&E, Sirius Red, α -SMA immunostaining, and F4/80 immunostaining. Frozen sections stained with Oil Red O. Scale bars: 50 μ m.

2. In Figure S3h: Measure Pck1 mRNA and PCK1 protein levels in the liver.

Author response: Thank you for pointing this out. In the revised manuscript, liver *Pck1* mRNA and PCK1 protein levels were measured and have been added to **Supplementary Figures 2h, I (Rebuttal Figures 2a-b)**.

Rebuttal Figure 2. (a) RT-qPCR and **(b)** western blot analysis of exogenous PCK1 mRNA and protein levels in the livers of AAV-injected mice.

3. Author response to Point 2 should be added to the Discussion. Pck1 mRNA data or the luciferase reporter assays would be more informative to show the role of ATF3 in Pck1 transactivation beyond the western blot of PCK1 in MIHA in Figure 1I, when ChIP data is used to demonstrate the ATF3 engagement on the Pck1 promoter. Primary mouse hepatocytes would be preferred instead of the MIHA cell line. PA 0.2 mM is a relatively low dose compared to the general use of PA 0.4 mM and PA 0.5 mM in the in vitro model. Whether this dose of 0.2 mM PA induces hepatocyte steatosis and inflammation needs to be verified in primary hepatocytes. Does the PCK1 KO exhibit PA-induced hepatocyte steatosis and inflammation in vitro?

Author response: We thank you for these constructive suggestions.

1. We have revised the *Discussion* to include the point response to Point 2 (page 21, lines 394-403).
2. Per the reviewer's suggestion, we have conducted the luciferase reporter assays to explore the role of ATF3 in *Pck1* transcription. The assays demonstrated that ATF3 knockdown significantly promotes transcription of the *PCK1* promoter luciferase reporter in MIHA cells (**Supplementary**

Figure 1i, Rebuttal Figure 3a). In addition, ATF3 knockdown restored PCK1 expression in primary mouse hepatocytes (**Rebuttal Figure 3b**). Taken together, these data confirm that ATF3 targets the *PCK1* promoter and represses *PCK1* transcription. We have included the luciferase reporter data in the revised manuscript (page 8, lines 121-122).

3. Although 0.2 mM is considered a relatively low dose of PA *in vitro*, treatment with 0.2 mM PA significantly reduced PCK1 expression in our study (**Figure 1g**). Moreover, 0.2 mM PA was chosen as experimental condition to induce hepatocyte steatosis and inflammation in primary hepatocytes in previous studies¹⁻³. Besides, palmitic acid is clearly lipotoxic and induced a dramatic, dose-dependent increase in apoptotic cell death^{4,5}. Based on these previous studies and our own data, we chose 0.2 mM PA as the experimental condition in the *in vitro* study.
4. To examine the effect of 0.2 mM PA on hepatocyte steatosis and inflammation, primary hepatocytes were treated with 0.2 mM PA, a dose that caused significant increases of lipid droplets visualized by BODIPY staining (**Rebuttal Figure 3c**) and *Il6* and *Tnfa* production, confirmed by RT- qPCR analysis (**Rebuttal Figure 3d**). Interestingly, previous findings have demonstrated that exposure to 0.2 mM PA results in intracellular lipid accumulation² and production of proinflammatory cytokine interleukin-8 from hepatocytes⁵. Therefore, these results suggested that 0.2 mM PA can induce hepatocyte steatosis and inflammation in primary hepatocytes.
5. In **Supplementary Fig 3e, f**, we have shown that PCK1-KO facilitates lipid accumulation in MIHA cells. In addition, a qPCR assay has been included to investigate whether PCK1-KO induced hepatocyte inflammation *in vitro*. The results show that genes related to inflammation, including *IL-6* and *TNF- α* , were highly expressed in L-KO cells (**Rebuttal Figure 3e**); therefore, it is clear that PCK1 KO exhibits PA-induced hepatic steatosis and inflammation *in vitro*.

Rebuttal Figure 3. a PCK1 promoter luciferase reporter activity in MIHA cells co-transfected with shControl or shATF3 plasmid. Results were obtained as relative luciferase activity against the activity of pGL3-Basic. Each figure represents at least three independent experiments. *** $p < 0.001$. **b** Protein levels of PCK1 in primary hepatocytes infected with either shControl or shATF3 and then treated with 0.2 mM PA. **c** Representative BODIPY staining in primary hepatocytes treated with BSA or PA-BSA. Scale bars: 25 μm. **d** mRNA levels of *Il6* and *Tnfa* in primary hepatocytes treated with BSA or PA-BSA (n=3). **e** mRNA levels of *IL-6* and *TNF-α* in PCK1-KO MIHA cells (n=3).

4. *Pck1* mRNA is needed for the common tissues including liver, intestine, white adipose, brown adipose, kidney, spleen. N=3 is expected to be found for liver western blot. Did the authors use GAPDH or ACTB for normalizing the Western blot? For GAPDH, the MW is 37 kD, while ACTB has a MW of 42 kD instead of 37 kD as shown above. The authors need to correct this for all ACTB Mr in this manuscript, not only limited to this figure. The uncut blot data for all western blot need to be provided in supplemental material.

Author response: Thank you for your valuable comments.

1. The PCK1 mRNA and PCK1 protein from common tissues have been examined in **Rebuttal Figure 4a** and **Figure 2b (Rebuttal Figure 4b)**. As seen in Rebuttal Figure 4a, *Pck1* mRNA is similarly expressed in multiple tissues both in wild-type (WT) and liver-specific *Pck1*-knockout (L-KO) mice, excluding liver tissues. The hepatic *Pck1* mRNA expression was specifically lost in L-KO mice. However, PCK1 protein was highly expressed in the liver, kidney, and brown tissue in WT mice. As expected, PCK1 expression was specifically lost in the liver tissue but not in the kidney or brown tissues of L-

KO mice. Thus, PCK1 distribution is tissue-specific and PCK1 protein expression is specifically lost in liver tissues of L-KO mice.

- We apologize for the unclear description in the manuscript. We used ACTB for the western blot normalization. The 37 kD in all figures represents Prestained Protein Marker. The uncut blot data for all western blots are available in Source Data.

Rebuttal Figure 4. *Pck1* mRNA (**a**) and PCK1 protein (**b**) expression in WT and L-KO mouse intestine, liver, spleen, kidney, white adipose, and brown adipose tissues.

- The mRNA nomenclature needs to be corrected throughout the manuscript and figures (<https://www.ncbi.nlm.nih.gov/gene>). For examples, Tnf- α needs to be Tnfa. Timp-1 needs to be Timp1. Il-10 needs to be Il10. Il-1b needs to be Il1b. Il-6 needs to be Il6.

Author response: We thank you for this suggestion. The mRNA nomenclature has been corrected throughout the manuscript and figures accordingly.

- What cells contribute to RhoA/PI3K/AKT expression change in the liver, hepatocytes or HSC? The direct effect of hepatocyte PCK1 knockout in RhoA/PI3K/AKT modulation should be examined with primary hepatocytes when the investigated PCK1 is located in hepatocytes. It is meaningless to show the effect of PCK1 overexpression in the RhoA/PI3K/AKT in hepatic stellate cells when the current study is trying to study the role of hepatocyte PCK1 function gain and loss. Could increasing GTP levels in hepatocytes modulate RhoA/PI3K/AKT pathway?

Author response: Thank you for this valuable comment.

1. In this study, we found that loss of hepatic PCK1 activated the RhoA/PI3K/AKT pathway, which contributed to increased secretion of PDGF-AA and promoted the activation of hepatic stellate cells. Our present study indicated that hepatocytes contribute to the changes in RhoA/PI3K/AKT expression in the liver.
2. We apologize for this imprecise description. Actually, we did not show the effect of PCK1 overexpression in the RhoA/PI3K/AKT in hepatic stellate cells. We have examined the direct effect of hepatocyte PCK1 knockout in RhoA/PI3K/AKT modulation in primary hepatocytes (**Figures 5d and 6c and Supplementary Figure 5c**) and found that the loss of PCK1 in hepatocytes induces HSC activation and ECM formation by activating the PI3K/AKT pathway.
3. As suggested, we performed experiments in the presence of 5'-GTP, 2Na⁺ (salt of guanosine triphosphate) or mycophenolic acid (MPA), an inhibitor of *de novo* guanine nucleotide synthesis to verify whether the increased GTP levels modulates the RhoA/PI3K/AKT pathway or not. The results revealed that the addition of 5'-GTP, 2Na⁺ can activate the RhoA/PI3K/AKT pathway in primary hepatocytes (**Supplementary Figure 5d, Rebuttal Figure 5a**). Interestingly, previous studies have also shown that the PI3K/AKT pathway can be activated by supplementation with guanosine, the substrate of GTP^{6,7}. On the other hand, MPA, an inhibitor of *de novo* guanine nucleotide synthesis, can robustly inhibit the RhoA/PI3K/AKT pathway by decreasing the extent of phosphorylated AKT (**Supplementary Figure 5e, Rebuttal Figure 5b**). These results suggest that increasing GTP levels activates the RhoA/PI3K/AKT pathway in hepatocytes. Consistent with our results, other studies have reported that MPA suppressed RhoA/PI3K/AKT pathway^{8,9}.

Rebuttal Figure 5. Immunoblot analysis of AKT, p-AKT (S473 or T308), PDGF-AA

AA, p-RhoA (S188), and RhoA in primary hepatocytes treated with 100 μM 5'-GTP, 2Na⁺ for 120 min **(a)** or 10 μM MPA for 48 h **(b)** in NASH fed mice.

7. "As described in our response to Reviewer #2's Comment #5, PCK1 expression was slightly downregulated in the early stage of NAFLD (Rebuttal Figure 7). The PI3K/AKT pathway was slightly activated after 8 weeks of NASH diet feeding, suggesting that the PI3K/AKT pathway was activated prior to liver fibrosis (Rebuttal Figure 10a). In addition, we found that PCK1 loss slightly activated the PI3K/AKT pathway after 8 weeks of NASH diet feeding (Rebuttal Figure 10b). Collectively, these results suggest that PCK1 knockout activates the PI3K/AKT pathway before the occurrence of NASH, but is not concomitant with the results of NASH."

This response answered how PCK1 is changed at the early stage of NASH, but did not address how hepatocyte PCK1 modulates the proposed mechanistic pathway under NASH and chow diet. What is the phenotype after 8-week NASH feeding? It is expected that the LKO mice should show a robust enhanced change of NASH-related phenotype compared to WT mice since LKO mice could show a phenotype under chow diet. Thus, it is not suitable to conclude that "these results suggest that PCK1 knockout activates the PI3K/AKT pathway before the occurrence of NASH, but is not concomitant with the results of NASH." What is the RhoA/PI3K/AKT signaling under chow diet and the early stage such as 1-2-week NASH diet-feeding in LKO mice and WT mice before LKO showed an enhanced NASH phenotype?

Author response: Thank you for your insightful comments.

1. We have performed additional experiments to observe the phenotype after 8 weeks of NASH feeding. Compared with the WT mice, the L-KO mice exhibited more severe hepatic steatosis indicated by Oil Red O staining after 8-week NASH feeding (**Rebuttal Figure 6a**). The results suggested that the L-KO mice showed an enhanced change of NASH-related phenotype compared with the WT mice.
2. To explore how PCK1 regulates RhoA/PI3K/AKT at the early stages of NASH *in vivo*, we established an early NASH animal model by feeding WT and L-KO mice with chow diet or NASH diet for two weeks (n=5 per group). We found that RhoA/PI3K/AKT pathway was slightly up-regulated in the livers of L-KO mice fed a NASH diet for 2 weeks compared with that in WT mice (**Rebuttal Figure 6b**), while the PI3K/AKT pathway was not obviously altered in WT mice under the NASH diet (**Rebuttal Figure 6c**). In addition, we observed that PCK1 expression is slightly downregulated in the early stages of NASH (**Rebuttal Figure 6c**). We have also performed additional experiments to observe the phenotype of hepatic steatosis after 2 weeks of

NASH feeding. Compared with the WT mice, the L-KO mice only exhibited a slight hepatic steatosis indicated by Oil Red O staining after 2-week chow feeding (**Rebuttal Figure 6d**). On the other hand, there was no significant difference in hepatic steatosis of WT mice feeding with chow diet or NASH diet for two weeks (**Rebuttal Figure 6e**). These results suggest that PCK1 knockout activates the PI3K/AKT pathway in the early stages of NASH.

Rebuttal Figure 6. **a** Representative H&E, Sirius Red, Oil Red O, and immunohistochemical staining of liver tissues from WT and L-KO mice fed the NASH diet for 2 weeks. Scale bars: 50 μ m. **b** Levels of indicated proteins in the

livers of WT and L-KO mice fed the NASH diet for 2 weeks. **c** Levels of indicated proteins in the livers of WT mice fed the Chow diet or the NASH diet for 2 weeks. **d-e** Paraffin-embedded liver sections were stained with H&E, Sirius Red, and F4/80. Frozen sections were stained with Oil Red O. Scale bars: 50 μ m.

8. For co-culture experiments, isolation of primary hepatocytes at the early stage of NASH diet-fed LKO mice to coculture with primary HSC is needed if the authors claim this as a causal factor and it is more important to test the levels of PDGF-AA signaling and the secretion in the supernatant for co-culture experiments. Could Rho/AKT/PI3K signaling activation or other mechanisms modulated by hepatocyte PCK1 lead to the increased secretion of PDGF-AA?

Author response: Based on your valuable suggestion, we performed additional experiments to examine whether the activation of primary HSC is dependent on PDGF-AA signaling from primary hepatocytes at the early stage of NASH diet-fed L-KO mice. Compared with those from the supernatant of primary hepatocytes from NASH diet-fed WT mice, the levels of PDGF-AA were higher in that of L-KO mice by ELISA analysis (**Rebuttal Figure 7a**). Upon exploring the role of the AKT signaling pathway in this process, we found that the AKT inhibitor MK2206 significantly blocked the increase in PDGF-AA levels in the supernatants of primary hepatocytes of L-KO mice (**Rebuttal Figure 7b**). Taken together, these data confirmed that PCK1 deficiency promoted PDGF-AA expression through the PI3K/AKT pathway and activated HSCs through hepatocyte-HSC crosstalk.

Rebuttal Figure 7. a Secreted PDGF-AA levels in conditioned medium with primary hepatocytes from WT and L-KO mice fed the NASH diet for 2 weeks. **b** Secreted PDGF-AA levels in conditioned medium with primary hepatocytes treated with AKT inhibitor MK2206 (10 μ M) from L-KO mice fed the NASH diet for 2 weeks.

9. The sgRhoA and SgAkt1 add evidence of target-specific effects. Since the authors did sgRhoA and SgAkt1 experiments separately from the inhibitor treatment, the off-target effect of those inhibitors could not be excluded. This needs to be discussed as a limitation of showing data of these inhibitors unless the authors could show that the inhibitors rescue the phenotype depending on the presence of RhoA or Akt1 at least in vitro PA-treated primary hepatocyte model.

Author response: We thank you for this presentation and agree that off-target effects from those inhibitors cannot be excluded.

The PI3K pathway has been shown to have more crosstalk with many other pathways. This characteristic feature may induce more off-target effects in PI3K/Akt/mTOR inhibitors. Since both types of interventions were done separately, we have now re-enforced this limitation in the *Discussion* (page 25, lines 479-482).

10. The information for human samples is incomplete. Only “clinical informations (should be information) of patients with NASH” was provided in Supplemental Table 1, which is unacceptable. The information for normal human (n=10) needs to be provided. All details for individual information need to be provided to generate the statistics and averages, but not only described the average levels plus SEM.

Author response: We thank you for this valuable comment. We have now added the information for normal human (n=10) and the statistics and averages information into the Supplemental material (**Rebuttal Table 1**). And all details for individual information have been provided in Source Data.

Rebuttal Table 1. Clinical characteristics of subjects with control and NASH patients.

	Control	NASH	P value
N (male/female)	10(5/5)	36(18/18)	NS
Age (years)	43.7 ± 4.94	44.64 ± 2.81	NS
BMI (kg/m ²)	21.34 ± 0.92	27.54 ± 0.76	<0.0001
ALT (U/L)	21.5 ± 4.54	136.2 ± 11.47	<0.0001
AST (U/L)	23.4 ± 1.56	74 ± 6.52	0.0002
γ-GT (U/L)	18.6 ± 1.69	83.17 ± 10.49	0.0025

TC (mmol/L)	2.30 ± 0.20	4.56 ± 0.20	<0.0001
TG (mmol/L)	1.09 ± 0.15	1.41 ± 0.12	NS
Glucose (mmol/L)	4.71 ± 0.17	7.79 ± 0.38	0.0001

Clinical and biochemical characteristics of NASH patients and controls. Data are expressed as mean ± SEM. Differences were analyzed by the unpaired t test. BMI: body mass index; ALT: alanine transaminase; AST: aspartate transaminase; γ-GT: γ-glutamyl-transferase; TC: total cholesterol; TG: triglyceride.

11. The Rebuttal Figure 10a only investigates a role of the NASH diet at 8 weeks (relatively early stage) in modulation of PCK1 expression. It does not address how hepatocyte PCK1 modulates the proposed PI3K/AKT signaling at the early stage so that this proposed mechanism could be a causal factor to explain the phenotype. Many pathways could be disturbed at the end of NASH diet feeding as a result of the enhanced NASH phenotype, but not all changes could be a causal factor. It is important to show whether hepatocyte PCK1 upregulates the proposed mechanistic pathway before the hepatocyte PCK1 enhanced the NASH phenotype or under basic conditions or at early stages if the enhanced phenotype occurs at the beginning stage.

Author response: We thank you for pointing this out.

As described in our response to your comment # 7, PCK1 knockout activates the PI3K/AKT pathway in the early stages of NASH. The results of this study confirmed a previous finding that AKT phosphorylation was also enhanced in the liver of PCK1-silenced mice¹⁰. Meanwhile, additional experiments were performed to observe the phenotype of hepatic steatosis after 2 weeks of NASH feeding. The results showed that the phenotype of hepatic steatosis was not obviously induced in both WT and L-KO mice under the NASH diet. Hence, our present study indicated that the loss of PCK1 activates PI3K/AKT pathway in the early stage of NASH, which is not the result of the enhanced NASH phenotype.

12. Serum PDGF-AA levels at the early stage of NASH needs to be determined and the ELISA of the supernatant for PDFG-AA levels in primary hepatocytes from LKO and WT mice needs to be determined.

Author response: We thank you for this valuable comment. As suggested, serum PDGF-AA levels at the early stage of NASH have been determined in **Rebuttal Figure 8**. The ELISA assay of the supernatant for PDFG-AA levels in

primary hepatocytes from L-KO and WT mice has also been performed as seen in **Rebuttal Figure 7a**. The ELISA assay showed higher levels of PDGF-AA in the supernatants of primary hepatocytes or serum from L-KO mice than from WT mice fed the NASH diet for 2 weeks. These results suggested that PCK1 deficiency promoted PDGF-AA expression through the PI3K/AKT pathway.

Rebuttal Figure 8 PDGF-AA levels in serum from WT and L-KO mice fed the NASH diet for 2 weeks detected using enzyme-linked immunosorbent assay (ELISA).

13. “We conducted a series of additional in vivo experiments. Using AAV8-TBG viral vectors to restore PCK1 expression in hepatocytes, the forced expression of PCK1 in the liver of L-KO mice reversed the phenotype induced by the chow and NASH diets. Mice with PCK1 re-expression had a lower liver weight, body weight, serum liver enzymes. Moreover, lipid deposition, inflammation, and fibrosis significantly improved in L-KO mice injected with AAV-Pck1. We have described these experiments in the revised manuscript (Figure 3h–l and Supplementary Figure 2h–k, page 10-11, lines 174–191).”

Hepatocyte PCK1 needs to be quantified to determine whether the forced expression worked. The source of the materials listed in Table S2 needs to be clarified. It is not acceptable to show the source of material as “lab stock”. The detailed information for all new materials used in the revision needs to be provided and updated in Supplemental Tables.

Author response: We thank you for highlighting this point. Hepatocyte PCK1 has been quantified in **Supplementary Figure. 2h, I (Rebuttal Figure 2a-b)**. The Supplemental Table S2 has also been updated in the revised version of our manuscript accordingly.

14. “Thank you for this suggestion. According this advice, we treated WT mice fed the NASH diet with Rhosin and MK2206 for 24 weeks; the results are shown below (Rebuttal Figure 12). Rhosin and MK2206 treatment decreased the

weight, liver weight, and lipid accumulation (TG and FFA levels) in the liver and improved the inflammation and fibrosis phenotype in WT mice, suggesting that treatment with Rhosin and MK2206 in WT mice fed the NASH diet partially alleviated the progression of NASH.”

This experiment did not address the therapeutic effect, but still the preventive effect? It is expected that the mice need to be fed a NASH diet first to establish NASH model and then treated with inhibitors. The body weight of WT mice after 24-week HFD feeding is only 32 g and the histology data showed a very mild and weak steatosis as well as liver fibrosis, indicating that this NASH model is a very mild NASH model. In addition, the NASH-inducing diet is too mild in the current model, which was also evidenced by 1.2 times of liver weight increase as well as liver lipid increase in the 24-week NASH diet-fed WT mice in Figure 2c compared to Figure 3e. Did this NASH diet really induce a substantially increase of NASH, fatty liver, inflammation and fibrosis? The NASH diet D0910031 (Research diets) would be more potent and suitable to induce NASH and fibrosis than the current diet, which is too mild and weak to induce NASH. The Sv129 background of the LKO mice and WT mice may render the mouse insensitive to the NASH induction. The brown coat color for these mice does not indicate that they were backcrossed with C57BL/6J mice for 5 generations. C57BL/6J mice are preferred for metabolic studies. This disadvantage needs to be addressed in the Discussion.

Author response: We thank you for your constructive input and help in making this paper more conclusive and focused. Compared with that of D0910030, the content of D12492 (60% kcal from fat similar in composition to an American fast-food diet) is higher in fat, which is more likely to induce fatty liver. D12492 is one of the widely used HFDs for obesity research, which induced visceral obesity and hepatic steatosis characterized by significantly increased plasma FFA and TG concentrations and plasma alanine aminotransferase (ALT) in C57BL/6 mice fed for 8 weeks¹¹. In another study, C57BL/6 mice fed diet D12492 for 16 weeks exhibited increased body and adipose tissue weights, widespread hepatic steatosis indicated by Oil Red O staining and increased hepatic TGs, mild fibrosis, and adipose tissue inflammation¹². When fed chronically over one year, an HFD (D12492) induced NASH phenotype characterized by inflammation along with excess body weight, hyperinsulinemia, and hypercholesterolemia^{13–16}. Even a slightly lower-fat (45 kcal% fat, D12451) containing HFD induced steatosis and steatohepatitis after 6 months in C57BL/6 mice¹⁷. Also, it is well documented that the NASH diet we adopted can induce a NASH phenotype^{18–20}. In addition to the HFDs, 23.1 g/L fructose and 18.9 g/L glucose were added to the drinking water of our NASH diet to promote liver inflammation. Fructose excessive intake causes intestinal barrier deterioration and endotoxemia, further aggravating liver inflammation^{21,22}. In summary, our dietary strategy can indeed establish a NASH model.

Pck1^{loxp/loxp} mice have a 129S6/SvEv background. Indeed, the brown coat color for the L-KO mice indicates that the mouse background conversion was not entirely completed. Considering these factors, we have adjusted the conclusions and added the limitations of the NASH model in the *Discussion* (page 22, lines 419-426).

15. The authors described that the "... prevents NASH development in vivo." in the Figure 8 legend. Is this preventive effect or therapeutic effect?

Author response: We thank you for raising this point.

We apologize for this imprecise description. As such, we have revised this in the Figure 8 legend of the revised manuscript (page 51, lines 1095) as follows: "Fig. 8 Pharmacological inhibition or genetic silencing of AKT1 or RhoA alleviates NASH development *in vivo*."

References

1. Kim, S. H. *et al.* Ezetimibe ameliorates steatohepatitis via AMP activated protein kinase-TFEB-mediated activation of autophagy and NLRP3 inflammasome inhibition. *Autophagy* **13**, 1767–1781 (2017).
2. Kaur *et al.* Increased Expression of RUNX1 in Liver Correlates with NASH Activity Score in Patients with Non-Alcoholic Steatohepatitis (NASH). *Cells* **8**, 1277 (2019).
3. Fan, Z. *et al.* The histone methyltransferase Suv39h2 contributes to nonalcoholic steatohepatitis in mice. *Hepatology* **65**, 1904–1919 (2017).
4. Malhi, H., Bronk, S. F., Werneburg, N. W. & Gores, G. J. Free Fatty Acids Induce JNK-dependent Hepatocyte Lipoapoptosis. *Journal of Biological Chemistry* **281**, 12093–12101 (2006).
5. Joshi-Barve, S. *et al.* Palmitic acid induces production of proinflammatory cytokine interleukin-8 from hepatocytes. *Hepatology* **46**, 823–830 (2007).
6. Di Iorio, P. *et al.* The antiapoptotic effect of guanosine is mediated by the activation of the PI 3-kinase/AKT/PKB pathway in cultured rat astrocytes. *Glia* **46**, 356–368 (2004).
7. Dal-Cim, T. *et al.* Guanosine protects human neuroblastoma SH-SY5Y cells against mitochondrial oxidative stress by inducing heme oxygenase-1 via PI3K/Akt/GSK-3 β pathway. *Neurochemistry International* **61**, 397–404 (2012).
8. Aghazadeh, S. & Yazdanparast, R. Mycophenolic acid potentiates HER2-overexpressing SKBR3 breast cancer cell line to induce apoptosis: involvement of AKT/FOXO1 and JAK2/STAT3 pathways. *Apoptosis* **21**, 1302–1314 (2016).
9. Gu, C. *et al.* Inhibiting the PI3K/Akt pathway reversed progesterin resistance in endometrial cancer. *Cancer Science* **102**, 557–564 (2011).
10. Gómez-Valadés, A. G. *et al.* *Pck1* Gene Silencing in the Liver Improves Glycemia Control, Insulin Sensitivity, and Dyslipidemia in *db/db* Mice. *Diabetes* **57**, 2199–2210 (2008).
11. Kirpich, I. A. *et al.* Integrated hepatic transcriptome and proteome analysis of mice with high-fat diet-induced nonalcoholic fatty liver disease. *The Journal of Nutritional Biochemistry* **22**, 38–45 (2011).
12. Flores-Costa, R. *et al.* The soluble guanylate cyclase stimulator IW-1973 prevents inflammation

- and fibrosis in experimental non-alcoholic steatohepatitis: Selective stimulation of sGC prevents NASH. *British Journal of Pharmacology* **175**, 953–967 (2018).
13. Liu, X. *et al.* S100A9: A Potential Biomarker for the Progression of Non-Alcoholic Fatty Liver Disease and the Diagnosis of Non-Alcoholic Steatohepatitis. *PLoS ONE* **10**, e0127352 (2015).
 14. Xu, F. *et al.* Annexin A5 regulates hepatic macrophage polarization via directly targeting PKM2 and ameliorates NASH. *Redox Biology* **36**, 101634 (2020).
 15. Sasaki, G. Y. *et al.* Green Tea Extract Treatment in Obese Mice with Nonalcoholic Steatohepatitis Restores the Hepatic Metabolome in Association with Limiting Endotoxemia-TLR4-NFκB-Mediated Inflammation. *Mol. Nutr. Food Res.* **63**, 1900811 (2019).
 16. Zhu, C. *et al.* Notch activity characterizes a common hepatocellular carcinoma subtype with unique molecular and clinicopathologic features. *Journal of Hepatology* **74**, 613–626 (2021).
 17. van der Heijden, R. A. *et al.* High-fat diet induced obesity primes inflammation in adipose tissue prior to liver in C57BL/6j mice. *Aging* **7**, 256–268 (2015).
 18. Zhou, R. *et al.* Intestinal α1-2-Fucosylation Contributes to Obesity and Steatohepatitis in Mice. *Cellular and Molecular Gastroenterology and Hepatology* **12**, 293–320 (2021).
 19. Tsuchida, T. *et al.* A simple diet- and chemical-induced murine NASH model with rapid progression of steatohepatitis, fibrosis and liver cancer. *Journal of Hepatology* **69**, 385–395 (2018).
 20. Liu, X.-J. *et al.* Characterization of a murine nonalcoholic steatohepatitis model induced by high fat high calorie diet plus fructose and glucose in drinking water. *Lab Invest* **98**, 1184–1199 (2018).
 21. Jones, N. *et al.* Fructose reprogrammes glutamine-dependent oxidative metabolism to support LPS-induced inflammation. *Nat Commun* **12**, 1209 (2021).
 22. Todoric, J. *et al.* Fructose stimulated de novo lipogenesis is promoted by inflammation. *Nat Metab* **2**, 1034–1045 (2020).

REVIEWER COMMENTS

Reviewer #2 (Remarks to the Author):

The authors have positively responded to most of the comments. However, the major issue for this manuscript the NASH model employed, was not adequately addressed. They use D12492 in combination with fructose and glucose, which they aim to induce NASH. This model induces a very mild NASH phenotype as shown here and in other publications but fails to generate significant NASH endpoints. Liver weights of wild-type (WT) mice fed a chow diet for 24 weeks are 1.9 g in Figure 2d, while in Figure 3c, the liver weight is 2.1 g; this is quite a weak model to show only 0.3 g increase of liver weight induced by their NASH diet. Liver triglycerides it is about 27 mg/g liver in chow-fed WT mice, while also only 30 mg/g in the liver after NASH diet feeding for 24 weeks, as long as chow diet feeding, as shown in Figure S2e. These data show that their NASH diet fails to generate significantly higher hepatic lipid accumulation. No cholesterol is presented in this model and the fibrosis is low. As the authors noted in their rebuttal to Comment #14, D12492 is more widely used for obesity research. A recent publication (PMID: 34651580), showed that the high-fat/high cholesterol (HFHC) diet induces a phenotype more resembling human NASH, and D12492 resembles the standard high-fat diet HFD. Considering the poor effect of D12492 in inducing NASH, the authors need to add another NASH diet, such as D0910031 (Research diets), to induce NASH in this study.

D12492 diet is called NASH diet in the whole manuscript as well as in Figures. This diet is more accurately to be nominated as “HFD”.

Gene/mRNA nomenclature for *Tnfa*, *Tgfb*, etc, should be corrected as *Tnfa*, *Tgfb* (italics and no Greek symbols) protein name in the figures and manuscripts should be all caps, no italics.

Ethical approval protocol numbers for human sample collection and use needs to be provided. In addition, for human NASH samples, it only includes the exclusion criteria in the Methods in Line537-545. The inclusion criteria of NASH and non-NASH patients were not mentioned. In other words, what is the criteria for including normal and NASH human liver samples in this study? How to judge the NASH presence in the patients in this study and how to determine which livers could be normal control liver, both of which need to be defined in methods.

Efficacy for shATF3 in the MIHA cells were not shown when they showed that shATF3 could regulate the luciferase reporter activity. Did overexpressing ATF3 decrease the Pck1 luciferase reporter activity?

In Figure 3e, space for pictures of Sirius Red staining between 2 groups could be increased to match and be consistent with other pictures in this figure panel.

Author Rebuttals to Second Revision

Reviewer #2 (Remarks to the Author):

The authors have positively responded to most of the comments. However, the major issue for this manuscript the NASH model employed, was not adequately addressed. They use D12492 in combination with fructose and glucose, which they aim to induce NASH. This model induces a very mild NASH phenotype as shown here and in other publications but fails to generate significant NASH endpoints. Liver weights of wild-type (WT) mice fed a chow diet for 24 weeks are 1.9 g in Figure 2d, while in Figure 3c, the liver weight is 2.1 g; this is quite a week model to show only 0.3 g increase of liver weight induced by their NASH diet. Liver triglycerides it is about 27 mg/g liver in chow-fed WT mice, while also only 30 mg/g in the liver after NASH diet feeding for 24 weeks, as long as chow diet feeding, as shown in Figure S2e. These data show that their NASH diet fails to generate significantly higher hepatic lipid accumulation. No cholesterol is presented in this model and the fibrosis is low. As the authors noted in their rebuttal to Comment #14, D12492 is more widely used for obesity research. A recent publication (PMID: 34651580), showed that the high-fat/high cholesterol (HFHC) diet induces a phenotype more resembling human NASH, and D12492 resembles the standard high-fat diet HFD. Considering the poor effect of D12492 in inducing NASH, the authors need to add another NASH diet, such as D0910031 (Research diets), to induce NASH in this study.

D12492 diet is called NASH diet in the whole manuscript as well as in Figures. This diet is more accurately to be nominated as "HFD".

Author response: We thank Referee for insights and constructive comments.

We really appreciate your recommendations regarding the NASH diet. We agree that the choice of research diets is important. However, due to the time cost of performing another dietary model (D0910031 diet) and the COVID-19 pandemic, we are very sorry that we will not be able to complete all the experiments with another NASH diet in three months. We also agree that our model induces a mild NASH phenotype and still have some limitations, but our dietary strategy that D12492 in combination with fructose and glucose can indeed establish a NASH model as reported in previously studies¹⁻³. The remarkable increase of liver weights and TG were not obviously in our model. In addition, liver cholesterol has been presented in **Figures 2f-g and 8g and Supplementary Figure. 2d-e and 6d-e**. As the reviewer's previous Comment #14, we speculate that the background of the mice may contribute to lack of sensitive to the NASH induction and we have included the limitations in the *Discussion of revised manuscript* (page 22, lines 428-430).

To make up for this, we used a genetic model with hepatic deficiency in phosphatase and tensin homolog (PTEN) to induce NASH. We generated a mouse model with biallelic deletion of *Pck1* and *Pten* in the liver. The breeding scheme for the generation of the following two groups of mouse cohorts include the following: (1) single homozygous knockout of *Pten* with *Alb-Cre* (*cPten^{ff}*); (2) homozygous knockout of *Pten* and *Pck1* with *Alb-Cre* (*cPten^{ff}Pck1^{ff}*). PCR performed with genomic DNA validated that the various alleles were excised accordingly (**rebuttal figure 1a**). Western blot analysis of liver lysates isolated from 6-month-old mice showed that PTEN knockout was efficient in *cPten^{ff}* and *cPten^{ff}Pck1^{ff}* mice and PCK1 knockout was efficient in *cPten^{ff}Pck1^{ff}* mice (**rebuttal figure 1b**). At 5-6 months, *cPten^{ff}Pck1^{ff}* mice had higher liver/body weight ratios compared with *cPten^{ff}* mice (**rebuttal figure 1c**). AST, ALT, IL-6, TNF- α , TC, TG, and FFAs increased in the serum and liver tissues of *cPten^{ff}Pck1^{ff}* mice, suggesting more serious liver injury and lipid metabolism disorder in *cPten^{ff}Pck1^{ff}* mice (**rebuttal figure 1d-e**). Analysis of *cPten^{ff}Pck1^{ff}* liver sections revealed increased fat droplets, more severe fibrosis, and greater macrophage infiltration (**rebuttal figure 1f-g**). Additionally, elevated expression levels of genes related to inflammation and fibrosis (**rebuttal figure 1h**) and α -SMA, COL1A1, COL3A1, PDGF-AA, and p-AKT (S473, T308) (**rebuttal figure 1i**) were observed in *cPten^{ff}Pck1^{ff}* mice. In summary, hepatic *Pck1* depletion showed substantial liver steatosis, inflammation and fibrosis in PTEN-null livers. These extensive new data support the conclusions of our work.

As suggested, the NASH diet is more accurately to be nominated as “HFCD-HF/G”³, we have made the changes accordingly in the revised manuscript. As all reviewers have agreed, we have already tried our best to carry out extensive experiments to address all issues raised by the reviewers, so we hope the reviewer will also agree that our work in its current state is more than sufficient for the publication in the Nature Communications.

Rebuttal Figure 1. **a** Representative PCR genotyping results from WT, *cPten^{ff}* and *cPten^{ff}Pck1^{ff}* mice. **b** Western blot analysis of PCK1 and PTEN levels in livers from mice of the indicated genotypes, with β -actin serving as the loading control. **c** Liver weight to body weight (percentage) in three types of mice (n = 5). **d** ALT, AST, IL-6 and TNF- α levels in serum were detected using enzyme-linked immunosorbent assay (ELISA). **e** Plasma levels of total triglycerides (TG), total cholesterol (TC), and free fatty acids (FFA). **f-g** Paraffin-embedded liver sections were stained with hematoxylin and eosin (H&E), Sirius Red, or immunostained for F4/80. Frozen sections were stained with Oil Red O. Scale bars: 50 μ m. **h** Quantitative PCR analysis of liver mRNA expression. Data are expressed as the mean \pm SEM; **p* < 0.05, ***p* < 0.01, ****p* < 0.001; n.s., not significant. *p* values obtained via two-tailed unpaired Student's *t* tests. **i** Levels of the indicated proteins, with β -actin serving as the loading control.

Gene/mRNA nomenclature for *Tnfa*, *Tgfb*, etc, should be corrected as *Tnfa*,

Tgfb (*italics* and no Greek symbols) protein name in the figures and manuscripts should be all caps, no italics.

Author response: We thank you for pointing this out. The mRNA nomenclature has been corrected throughout the manuscript and figures accordingly.

Ethical approval protocol numbers for human sample collection and use needs to be provided. In addition, for human NASH samples, it only includes the exclusion criteria in the Methods in Line537-545. The inclusion criteria of NASH and non-NASH patients were not mentioned. In other words, what is the criteria for including normal and NASH human liver samples in this study? How to judge the NASH presence in the patients in this study and how to determine which livers could be normal control liver, both of which need to be defined in methods.

Author response: We thank you for this valuable comment.

1. Ethical approval protocol number has been provided in the *Methods* (page 28, lines 550).
2. The inclusion criteria of NASH and non-NASH patients were provided in the *Methods* (page 27-28, lines 536-543).

Efficacy for shATF3 in the MIHA cells were not shown when they showed that shATF3 could regulate the luciferase reporter activity. Did overexpressing ATF3 decrease the Pck1 luciferase reporter activity?

Author response: Thank you for your valuable comments. Efficacy for shATF3 in the MIHA cells has been validated in **Figure 1i**. In addition, we revalidated the efficacy of shATF3 in the MIHA cells in **rebuttal figure 2a**.

As suggested, we have conducted the luciferase reporter assays and found that ATF3 overexpression significantly repressed the transcription of human *PCK1* promoter as shown by the decrease of luciferase activity in MIHA cells (**Supplemental figure 1j and rebuttal figure 2b**), and we validated the efficacy of ATF3 overexpression in the MIHA cells in **rebuttal figure 2c**.

Rebuttal Figure 2. **a** Protein levels of ATF3 in MIHA infected with either shControl or shATF3 and then treated with 0.2 mM PA. **b** *PCK1* promoter luciferase reporter activity in MIHA cells co-transfected with either empty vector or ATF3 overexpressing plasmid. Results were obtained as relative luciferase

activity against the activity of pGL3-Basic. Each figure represents at least three independent experiments. * $p < 0.05$. **c** Protein levels of ATF3 in MIHA transfected with either vector or ATF3 plasmid.

In Figure 3e, space for pictures of Sirius Red staining between 2 groups could be increased to match and be consistent with other pictures in this figure panel.

Author response: Thank you for raising this point. Space for pictures of Sirius Red staining between 2 groups has been increased in **Figure 3e**.

Reference

1. Tsuchida, T. *et al.* A simple diet- and chemical-induced murine NASH model with rapid progression of steatohepatitis, fibrosis and liver cancer. *Journal of Hepatology* **69**, 385–395 (2018).
2. Zhou, R. *et al.* Intestinal α 1-2-Fucosylation Contributes to Obesity and Steatohepatitis in Mice. *Cellular and Molecular Gastroenterology and Hepatology* **12**, 293–320 (2021).
3. Liu, X.-J. *et al.* Characterization of a murine nonalcoholic steatohepatitis model induced by high fat high calorie diet plus fructose and glucose in drinking water. *Lab Invest* **98**, 1184–1199 (2018).

REVIEWERS' COMMENTS

Reviewer #2 (Remarks to the Author):

1. Figure 2d: The authors should remove this low-quality data. Similarly, the picture for mouse shape/size in Figure 2c, 2i, 8e could be removed as the authors already showed body weight data. These figures should be made with a ruler if shown.

2. Data using cPten transgenic mice in Rebuttal Figure 1 supports the conclusion of the major finding. These data should be shown in the manuscript to support the phenotype of the current finding.

3. As the authors clearly realize, the increase of liver weights and TG as well as the NAFLD model was mild in their model. Quantitation of Sirius red staining of WT chow-fed mice showed 1% in Figure 2i, while quantitation in Figure 3I also showed about 1% in WT HFCD-HF/G-fed mice, thus indicating that no significant fibrosis was induced by this model. The authors provided some data from other publications supporting that their dietary regimen could induce NASH, however the fibrosis, liver weight and liver TG were only marginally increased in this study. The authors need to revise the description of their dietary model to NAFLD or the more recently accepted nomenclature metabolic-associated fatty liver disease (MAFLD), instead of NASH. The essence of the message of this work does not change.

I suggest a change of the title to: Deficiency of gluconeogenic enzyme PCK1 promotes metabolic-associated fatty liver disease (MAFLD) through PI3K/AKT/PDGF axis activation

Author Rebuttals to The Third Revision

Reviewer #2 (Remarks to the Author):

1. Figure 2d: The authors should remove this low-quality data. Similarly, the picture for mouse shape/size in Figure 2c, 2i, 8e could be removed as the authors already showed body weight data. These figures should be made with a ruler if shown.

Author response: We thank you for this valuable comment.

As suggested by the reviewer, we have removed the low-quality picture for mouse shape/size in **Figure 2d, 3c, 3i, 8e**.

2. Data using cPten transgenic mice in Rebuttal Figure 1 supports the conclusion of the major finding. These data should be shown in the manuscript to support the phenotype of the current finding.

Author response: We appreciate your constructive suggestions to improve our manuscript.

Per the reviewer's suggestion, we have included the data using the genetic model (*cPten^{ff}* mice) in **Supplementary Figure. 3** and we also have included the results in the revised manuscript (page 10-11, lines 179-192).

3. As the authors clearly realize, the increase of liver weights and TG as well as the NAFLD model was mild in their model. Quantitation of Sirius red staining of WT chow-fed mice showed 1% in Figure 2i, while quantitation in Figure 3I also showed about 1% in WT HFCD-HF/G-fed mice, thus indicating that no significant fibrosis was induced by this model. The authors provided some data from other publications supporting that their dietary regimen could induce NASH, however the fibrosis, liver weight and liver TG were only marginally increased in this study. The authors need to revise the description of their dietary model to NAFLD or the more recently accepted nomenclature metabolic-associated fatty liver disease (MAFLD), instead of NASH. The essence of the message of this work does not change.

I suggest a change of the title to: Deficiency of gluconeogenic enzyme PCK1 promotes metabolic-associated fatty liver disease (MAFLD) through PI3K/AKT/PDGF axis activation

Author response: We thank you for the constructive suggestion.

We really appreciate your recommendations regarding the NASH model. We

agree with the reviewer about revising the description of our dietary model to metabolic-associated fatty liver disease (MAFLD), instead of NASH.